# Continual Learning With Participation Privacy: An Auditable Buffering-Aggregation Recipe

**T-H. Hubert Chan** [1]  **Elaine Shi** [2]  **Mengshi Zhao** [1]  **Mingxun Zhou** [3]

## Abstract

Modern federated and streaming learning systems often release intermediate models, so privacy must hold for the full trajectory under adaptive interaction. Motivated by participation privacy, we study single-edit neighboring user streams, where one insertion/deletion shifts all subsequent updates and defeats standard Hamming-neighbor continual-release analyses. We give an auditable modular recipe. A randomized buffering wrapper emits bins of size $[U, 2U]$, reducing single-edit streams to a Hamming-style per-bin update stream with explicit backlog/delay guarantees, where $U$ is calibrated by the privacy parameters $(\varepsilon, \delta)$. We then prove a certification theorem for independently decomposable (prefix-causal, fresh-noise) continual mechanisms: any non-adaptive Hamming-neighbor DP proof lifts to adaptive inputs. Together, these ingredients yield trajectory-level $(\varepsilon, \delta)$-DP for single-edit streams using standard primitives (e.g., tree prefix sums), with an explicit privacy–latency link via $U$. Streaming DP-SGD experiments validate the privacy-utility-latency tradeoffs and the induced delay distributions.

## 1. Introduction

Modern federated and streaming learning systems release intermediate model snapshots throughout training, so an adversary may observe the entire trajectory $(w^{(t)})_{t \geq 0}$ rather than only a final model. A canonical instance is streaming stochastic gradient descent (SGD): at step $t$, a user contributes a private loss function $f_t$ and the learner updates using a gradient step based on the current model (Thakurta & Smith, 2013; Kairouz et al., 2021). Beyond protecting the data within each $f_t$, some deployments also require *participation privacy*: a user may wish to ensure *plausible deniability* of whether they participated at all. This motivates *single-edit* (edit-distance) neighboring streams, where two user-update streams are neighbors if one can be obtained from the other by inserting or deleting a single user event.

Continual release also makes the learning process intrinsically interactive: the released model $w^{(t)}$ influences what happens next, so the update stream can be adaptively generated in response to prior outputs. While classical continual-release primitives such as tree-based private prefix sums (Dwork et al., 2010a; Chan et al., 2011) were originally analyzed under statically chosen (Hamming-neighbor) streams, recent work formalizes differential privacy against such adaptive interaction (Jain et al., 2023). Our goal is to make this adaptive viewpoint compatible with the *single-edit* participation model above.

The difficulty is that Hamming-style neighboring streams are not the right abstraction for participation privacy on a stream. Most continual DP analyses change the value at a single time index $t_0$ while keeping all other positions aligned (Dwork, 2006). In contrast, a single insertion/deletion shifts the alignment of all subsequent positions, so an edit-neighbor pair can be far from Hamming-neighboring. This also undermines deterministic batching: partitioning the stream into fixed contiguous blocks can cause one edit to shift many later batch boundaries, so standard Hamming-neighbor continual DP guarantees do not directly transfer. Appendix G.1 gives an explicit boundary-shift attack on deterministic batching under edit adjacency, and we corroborate this failure empirically in our experiments.

To address this mismatch, we use a simple modular recipe: randomized buffering and an adaptive-safety certification for downstream continual primitives. We apply a *randomized buffering* wrapper (Chan et al., 2022; Zhou et al., 2023) that introduces random delay and releases user updates in *bins* of controlled size (e.g., in $[U, 2U]$); here $U$ is a

Authors are listed in alphabetical order. [1]The University of Hong Kong, Hong Kong, China [2]Carnegie Mellon University, Pittsburgh, USA [3]The Hong Kong University of Science and Technology, Hong Kong, China. Correspondence to: T-H. Hubert Chan <hubert@cs.hku.hk>.

*Proceedings of the 43rd International Conference on Machine Learning*, Seoul, South Korea. PMLR 306, 2026. Copyright 2026 by the author(s).

*privacy-implied systems cost*, calibrated by the target $(\varepsilon, \delta)$ and inducing delay. We then invoke standard continual DP primitives (e.g., tree/prefix-sum) to privately aggregate the resulting vector updates, and prove a *certification theorem* showing that a broad class of independently decomposable continual mechanisms are *adaptive-safe*: any non-adaptive privacy proof lifts automatically to the feedback/adaptive setting.

**Contributions.**

- **Edit-neighbor continual learning under feedback.** We formalize participation privacy via single-edit neighboring user-update streams in a continual learning setting with intermediate releases.

- **Modular pipeline with an explicit privacy to latency link.** We combine randomized buffering with standard continual DP primitives to obtain trajectory-level privacy for edit-neighbor streams; the target $(\varepsilon, \delta)$ determines the buffering level $U$, making induced delay explicit.

- **Certification theorem for adaptive safety of continual DP primitives.** Under a checkable structural condition (independent decomposability), any non-adaptive privacy analysis carries over to the feedback/adaptive setting, certifying safe reuse of canonical primitives such as tree/prefix-sum mechanisms.

- **Empirical validation (streaming SGD).** We report privacy–utility–latency tradeoffs under $(\varepsilon, \delta)$-DP (fixing $\delta$ and sweeping $\varepsilon$), including delay distributions implied by $U(\varepsilon, \delta)$; further instantiations are deferred to the appendix.

**Results and organization.** Our end-to-end guarantee is trajectory-level $(\varepsilon, \delta)$-DP for *single-edit* neighboring participation streams under feedback, obtained by composing randomized buffering with a continual-release DP aggregator (Theorem 3.1); the privacy target explicitly induces a buffering level $U(\varepsilon, \delta)$ and hence delay/staleness. A second ingredient is a reusable *certification* theorem showing that for independently decomposable continual mechanisms, a standard non-adaptive privacy analysis lifts to the adaptive/feedback setting (Theorem 4.3). Section 2 formalizes the interactive model, edit adjacency, and metrics; Section 3 gives a high-level view of the pipeline and states the main theorem; Section 4 develops the two auditable ingredients (randomized buffering and certification) and combines them via modular composition; Section 5 instantiates the recipe with continual DP-SGD (tree/prefix-sum aggregation), and experiments appear in Section 6. Technical proofs and additional instantiations are deferred to the appendix.

## 1.1. Related Work

**Continual-Release DP.** Canonical continual-release primitives for streaming aggregates include tree-based private prefix sums (Dwork et al., 2010a; Chan et al., 2011) and closely related lower-triangular (matrix-style) mechanisms (Denisov et al., 2022). These mechanisms are typically analyzed for *Hamming-neighboring* update streams, where only one time index differs. In contrast, our participation model uses *single-edit* (insertion/deletion) neighboring streams, where alignment shifts after the edit; this gap is mild for static datasets (Birrell et al., 2024) but is fundamental for streams.

Sliding-window variants of private stream release have also been studied for window aggregate queries and graph streams (Cao et al., 2013; Upadhyay et al., 2021). These works target recent-window utility, whereas our setting focuses on single-edit participation privacy, where one insertion/deletion shifts all subsequent stream positions and requires an edit-to-Hamming shielding interface.

**Adaptive Streams and Feedback.** Continual learning is inherently interactive: released outputs can influence future updates. Recent work formalizes differential privacy against such adaptive interaction (e.g., left-or-right style games and verification viewpoints) (Denisov et al., 2022; Jain et al., 2023). Our *certification* result is complementary: it gives a checkable structural condition (independent decomposability / prefix-causality with fresh randomness) under which a standard non-adaptive Hamming-DP analysis of a continual primitive lifts directly to the adaptive setting.

**Privacy-Induced Delayed Reactions.** Cohen et al. (2024) give a conceptually related delayed-reaction phenomenon for continual observation and online threshold queries. Their Appendix B delays threshold-type outputs on a directly observed online bit stream. Our delay has a different technical role: RandBin waits for future arrivals to form bins in $[U, 2U]$ so that a single insertion/deletion, which appears only in the neighboring-stream analysis, becomes a bounded Hamming-style perturbation while the full transcript, including timing and $\perp$ outputs, remains protected.

**Randomized Buffering, RSC, and Obliviousness Lineage.** Randomized buffering has appeared as infrastructure in oblivious data structures and related privacy modularity work (Chan et al., 2022; Zhou et al., 2023). A closely related random-partition idea also appears in the Reorder-Slice-Compute (RSC) paradigm of Cohen et al. (2023), where random slicing is used to synchronize neighboring executions and avoid a one-edit domino effect. Our setting differs in that the continual streaming transcript exposes release timing itself: the adversary observes whether each step emits a real block or the dummy symbol $\perp$. Thus, in

our use, RandBin is an interactive interface that converts single-edit participation streams into a sparse, Hamming-style per-bin update stream while also providing explicit backlog/delay guarantees calibrated by $(\varepsilon, \delta)$.

**Composition and Concurrency.** Beyond classical $(\varepsilon, \delta)$-DP (Dwork, 2006), alternative formalisms (e.g., Rényi divergence and tradeoff-based views) can streamline composition reasoning (Mironov, 2017; Dong et al., 2022; Vadhan & Zhang, 2023; Zhou et al., 2024), recovering familiar advanced composition bounds (Dwork et al., 2010b; Kairouz et al., 2015). Concurrent composition in interactive settings has also been studied, including adaptively chosen privacy parameters (Haney et al., 2023; Henzinger et al., 2026). These works typically assume each mechanism's neighboring relation is defined on a single underlying (static or dynamic) database; in our pipeline, we additionally need to reason about *neighbor-preserving transformations* between components (edit-to-Hamming shielding), which motivates our modular composition viewpoint. Our modular composition theorem can be viewed as combining the refinement-pair reduction of NPDP to standard DP in (Zhou et al., 2024) with the interactive/concurrent composition framework of (Henzinger et al., 2026); for completeness and to match our power-function accounting, we give a self-contained proof in Appendix C.

**Conflict of Interest Disclosure.** The authors declare no financial or other substantive conflicts of interest relevant to this work.

## 2. Setting and Preliminaries

We now fix the interactive streaming-learning model underlying continual release: the adversary observes both the released model trajectory and the release timing, and may influence future user events. We also define edit-style (participation) adjacency and the utility–latency metrics used throughout. With these conventions in place, Section 3 summarizes our buffering–aggregation pipeline and main guarantee.

**Streaming Learning Loop and Transcript.** At each discrete step $t = 1, 2, \ldots$, an adaptive environment produces a *user event* $f_t$ (e.g., a private loss or update). The server maintains a model $w^{(t)} \in \mathbb{R}^d$ and may release intermediate snapshots. To unify per-step and buffered releases, we include an observable timing bit $v_t \in \{0, 1\}$: if $v_t = 1$ the server incorporates available (buffered) events to form an update $g_t$ and publishes the new snapshot; if $v_t = 0$ it holds $w^{(t)} = w^{(t-1)}$. The adversary's view up to horizon $T$ is the transcript $\mathrm{tr}_{\leq T} := \left( (v_t, w^{(t)}) \right)_{t=0}^{T}$, and our privacy notion is defined with respect to this *joint* transcript (including timing). All privacy guarantees are required to

hold for every horizon $T$, and thus apply to the full trajectory.

**Feedback/Adaptive Interaction.** The environment (and thus the event stream) may depend on the past transcript: formally, $f_t$ may be chosen as an arbitrary (possibly randomized) function of $\mathrm{tr}_{<t}$. All guarantees quantify over such adaptive interaction. When we refer to a *non-adaptive* privacy analysis for a continual primitive, we mean its input stream is fixed in advance (independent of past releases).

**Participation Privacy via Single-Edit Adjacency.** Let $\mathcal{M}$ denote the full interactive mechanism (e.g., buffering $\rightarrow$ aggregation $\rightarrow$ releases) mapping an event stream $F = (f_1, f_2, \ldots)$ to a transcript. We use *single-edit* adjacency: $F \sim_{\mathrm{edit}} F'$ if $F'$ is obtained from $F$ by inserting or deleting exactly one event, shifting subsequent indices; equivalently, for some $t_0$, either (i) $f_t' = f_t$ for $t < t_0$ and $f_t' = f_{t+1}$ for all $t \geq t_0$ (deletion), or (ii) the reverse (insertion). The *unit of privacy* is one participation event; other units (e.g., a session spanning multiple events) reduce via grouping/composition (formal composition tools appear in Appendix A).

**Differential Privacy (Under Adaptive Interaction).** We say $\mathcal{M}$ is $(\varepsilon, \delta)$-DP w.r.t. $\sim_{\mathrm{edit}}$ if for all $F \sim_{\mathrm{edit}} F'$ and all measurable transcript-sets $S$,

$$\mathbf{Pr}\left[\mathcal{M}(F) \in S\right] \leq e^{\varepsilon} \mathbf{Pr}\left[\mathcal{M}(F') \in S\right] + \delta,$$

where the probability is over $\mathcal{M}$'s internal randomness; the quantification over event-generation strategies is captured by the adaptive-interaction model above. Our appendix sometimes uses the equivalent tradeoff-/power-function viewpoint to streamline interactive composition; the main text states guarantees in $(\varepsilon, \delta)$-DP; see Appendix A.

**Metrics.** *Utility:* test accuracy / loss of the final released model by time $T$ (e.g., $w^{(T)}$ or the last release by $T$). *Systems:* backlog $Q_t$ (pending, not-yet-incorporated events after time $t$) and inclusion delay $D(i) := \min\{t \geq i : f_i \text{ is incorporated by time } t\} - i$, measured in event arrivals.

**Extension: From One Edit to $k$ Edits.** If neighbors may differ by up to $k$ insertions/deletions, standard group-privacy/composition yields a $k$-dependent degradation of $(\varepsilon, \delta)$; in our pipeline this correspondingly increases the required privacy budgets and thus the induced buffering/latency. Full statements are deferred to the appendix.

## 3. Overview

This section summarizes our buffering–aggregation pipeline (Fig. 1) and the resulting end-to-end guarantee:

continual-release $(\varepsilon, \delta)$-DP for single-edit participation streams under feedback, where the adversary observes the released trajectory (and timing). The required buffering level $U(\varepsilon, \delta)$ is privacy-implied and explicitly governs delay.

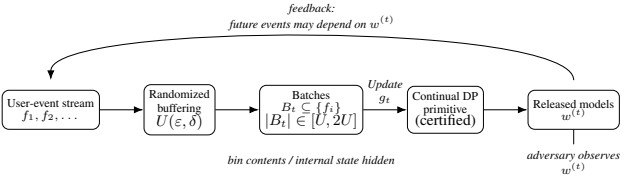

*Figure 1.* **Pipeline overview.** Randomized buffering (parameter $U(\varepsilon, \delta)$) converts a single-edit stream into bins that are consumed (in the sense defined in Appendix B) by a certified continual-release DP primitive, producing intermediate model releases. The adversary observes the released models; bin contents and internal states are hidden.

We next give the recipe and state the end-to-end guarantee.

**Recipe (buffering → aggregation → releases).** Given user-event stream $(f_t)$, at each step $t \geq 1$, buffering outputs $B_t$ with $|B_t| \in [U, 2U]$ or $\perp$. Let $\tau(1) < \tau(2) < \cdots$ be the (random) steps with $B_{\tau(k)} \neq \perp$.

1. **Buffer.** Run randomized buffering with $U(\varepsilon, \delta)$ to obtain $(B_t)_{t \geq 1}$.

2. **Update.** For each $k \geq 1$, form a clipped per-bin update $g_k := \frac{1}{|B_{\tau(k)}|} \sum_{f \in B_{\tau(k)}} \mathsf{Clip}_G\left(\nabla f(w^{(\tau(k)-1)})\right)$, where $\mathsf{Clip}_G(x) := x \cdot \min\left\{1, \frac{G}{\|x\|_2}\right\}$.

3. **Aggregate + release.** Feed $(g_k)_{k \geq 1}$ to a certified continual-release DP primitive and update only at bin times: output $w^{(\tau(k))}$ after processing $g_k$, and hold $w^{(t)} = w^{(\tau(k))}$ for $\tau(k) \leq t < \tau(k+1)$ (feedback allowed).

Theorem 3.1 states the resulting end-to-end privacy.

**Theorem 3.1** (End-to-end continual-release DP for single-edit streams)**.** *Run the above recipe with privacy target $(\varepsilon, \delta)$, allocating $(\varepsilon_b, \delta_b)$ to buffering and $(\varepsilon_a, \delta_a)$ to aggregation. Set $U = \Theta\left(\frac{1}{\varepsilon_b} \log \frac{1}{\delta_b}\right)$, and plug in any continual-release DP primitive that is adaptively $(\varepsilon_a, \delta_a)$-DP for Hamming-neighboring update streams. Then the released model trajectory $(w^{(t)})_{t \geq 0}$ is $(\varepsilon, \delta)$-DP with respect to single-edit neighboring event streams, even when future events are generated adaptively from past releases (feedback).*

*Proof idea.* Randomized buffering turns a single insertion/deletion on the event stream into a bounded Hamming-style perturbation on the emitted update stream; we then

apply a certified continual-release DP primitive whose non-adaptive analysis remains valid under feedback, and conclude via modular composition (Appendix C).

**Certification preview (adaptive safety of continual primitives).** We use a reusable *certification* for feedback: writing $x_t$ for the per-round input and $x_{1:t}$ for its prefix, a continual-release primitive is *independently decomposable* if at each release time $t$ it samples fresh independent randomness $R_t$ and outputs a deterministic function of $(x_{1:t}, R_t)$. For such primitives, standard *non-adaptive* $(\varepsilon, \delta)$-DP analyses remain valid under feedback. This covers tree/prefix-sum (Appendix F) and more general lower-triangular matrix mechanisms (Section 4.2).

**SGD instantiation and what we measure.** We instantiate the pipeline with streaming SGD: each emitted bin $B_{\tau(k)}$ triggers one clipped mini-batch gradient update, privatized by a continual-release DP aggregator (tree/prefix-sum in our experiments). The target $(\varepsilon, \delta)$ fixes $U(\varepsilon, \delta)$, yielding random batch sizes in $[U, 2U]$ and a backlog bound of order $O\left(\frac{1}{\varepsilon_b} \log t \left(\log t + \log \frac{1}{\delta_b}\right)\right)$ (Lemma 4.2). We report accuracy versus $(\varepsilon, \delta)$ together with empirical delay distributions implied by $U(\varepsilon, \delta)$. The theoretical convergence of various SGD variants under DP noise has already been extensively analyzed; we refer the reader to standard works (Jain et al., 2012; Song et al., 2013; Thakurta & Smith, 2013; Kairouz et al., 2021; Denisov et al., 2022).

**Deferred details.** Sections 4.1 and 4.2 formalize buffering and certification; the end-to-end proof of Theorem 3.1 (via modular composition) and additional instantiations/experiments appear in the appendix.

## 4. Auditable Ingredients: Randomized Buffering and Adaptive-Safe Continual-Release DP

This section isolates the two ingredients that make the pipeline in Fig. 1 auditable. First, we treat randomized buffering (RandBin) as an interface: it converts a *single-edit* event stream into a sparse per-bin update stream that admits a standard Hamming-neighbor DP analysis, while providing an explicit backlog guarantee. Second, we give a reusable *certification* theorem for adaptive interaction (output-dependent updates): for independently decomposable (prefix-causal) continual-release mechanisms, a non-adaptive DP proof remains valid even when updates are chosen adaptively as a function of past releases.

**Separation of Roles.** Randomized buffering is not needed for the certification theorem itself: Theorem 4.3 is a reusable adaptivity lift for independently decomposable

continual mechanisms. In our pipeline, buffering serves a different role: it shields the downstream primitive from single-edit adjacency, where one insertion or deletion can shift many later batch boundaries. Thus, our modular proof separates an upstream edit-to-Hamming interface from a downstream adaptive-safety certificate. We do not claim that randomized buffering is the only possible route; alternatives would require either another edit-shielding interface or a direct native proof under edit adjacency.

### 4.1. Randomized Buffering Wrapper

**RandBin Interface.** We use RandBin as a lightweight wrapper that turns an incoming event stream into well-sized bins, while keeping the buffering backlog under control and enabling our edit-to-Hamming reduction downstream. The input is a stream of user events $f_1, f_2, \ldots$ (losses, gradients, or updates). At each step $t \geq 1$, RandBin outputs a symbol $B_t$: either a concrete bin $B_t$ containing some pending events among those arrived by time $t$, or $\bot$ (no emission). Let $\tau(1) < \tau(2) < \cdots$ be the (random) emission times with $B_{\tau(k)} \neq \bot$. When $B_t = \bot$, nothing is sent to the downstream aggregator and the released model remains unchanged.

We rely on three interface properties:

**(P1) Bin size.** Whenever $B_t \neq \bot$, the bin size is controlled: $|B_t| \in [U, 2U]$.

**(P2) Backlog control.** The number of pending (unbinned) events $Q_t$ remains small for all $t$ (formal bound below), yielding explicit delay guarantees.

**(P3) Edit-to-Hamming wrapper: neighbor-preserving (NP) DP.** A single insertion/deletion in the input stream induces only a bounded Hamming-style perturbation in the emitted-bin (and hence per-bin update) stream, in the sense needed to compose with a Hamming-DP continual primitive. The details are deferred to Appendix D.

**Input-Identifiability Convention.** Lemma 4.1 uses the standard convention that events are identifiable, e.g., by unique IDs or metadata tags, so that after the first mismatch in an edit-neighbor pair the simulation can determine which stream contains the extra event. Anonymous streams with duplicate values require an extra disambiguation assumption; see Appendix D.

**Systems Metrics: Backlog and Delay.** The buffering level $U(\varepsilon_b, \delta_b)$ controls how aggressively RandBin batches the stream and thus the latency regime seen by learning. We track the **backlog** $Q_t$, the number of pending (unbinned) events after time $t$. For an event $f_i$, its **inclusion delay** $D(i)$ is the number of subsequent arrivals until it is placed

---

**Algorithm 1** RandBin (Interface-Level Skeleton)
___
1: **Input:** $(\varepsilon_b, \delta_b)$; stream $(f_t)_{t \geq 1}$
2: **Output:** $B_t \in \{\bot\} \cup \{\text{bins of size in } [U, 2U]\}$
3: Set $U = \Theta(\varepsilon_b^{-1} \log(1/\delta_b))$ and a distribution $\mathcal{D}$ supported on $[U, 2U]$
4: Maintain a FIFO buffer buf and a private randomized emission scheduler
5: **for** $t = 1, 2, \ldots$ **do**
6:     Append $f_t$ to buf
7:     **if** there is no scheduled emission **then**
8:         Output $\bot$
9:     **else**
10:         Sample $C \in [U, 2U]$ from $\mathcal{D}$
11:         Pop $C$ items from buf as $B_t$
12:         Output $B_t$; refresh scheduler
13:     **end if**
14: **end for**
___

in an emitted bin. In particular, if $Q_t \leq B(t)$ for all $t \in [i, T]$, then $D(i) \leq B(T)$ (and sharper bounds follow from the trajectory of $(Q_t)$).

**Interface Guarantees for Modular Composition.** We will use two interface properties of RandBin; proofs and constants appear in Appendix D.

**Lemma 4.1** (Edit-to-Hamming Wrapper (NPDP Contract)). *For any single-edit neighboring input streams,* RandBin *admits a neighbor-preserving paired simulation (with refinement) such that the coupled emitted-bin/update streams differ in a bounded Hamming-style way at the interface required for modular composition with a Hamming-DP continual primitive.*

*Intuition:* randomized boundaries prevent one insertion/deletion from cascading into many downstream bin shifts; the paired simulation formalizes this stability under interaction.

**Lemma 4.2** (Backlog Bound (Privacy-Implied Delay)). *With $U = \Theta(\varepsilon_b^{-1} \log(1/\delta_b))$, for every time $t$ the backlog satisfies $Q_t = O\left(\frac{1}{\varepsilon_b} \log t \cdot \left(\log t + \log \frac{1}{\delta_b}\right)\right)$.*

*This bound is obtained by implementing* RandBin*'s private emission scheduler via the streaming DP prefix-sum mechanism (Theorem F.1) and translating its additive-error guarantee into a worst-case queue bound. As a consequence, inclusion delay is controlled at the same order (up to constants) for events arriving by time $t$.*

*Intuition:* the noisy scheduling forces sufficiently frequent emissions to prevent sustained queue growth, while keeping bin sizes within $[U, 2U]$.

**On the Horizon Dependence.** The $\log t$ dependence in Lemma 4.2 is not merely an artifact of loose algebra. In our implementation, the emission scheduler is a continual private prefix-sum mechanism, and its additive error is

translated directly into backlog. On a unit-rate stream, if $M_t$ is the cumulative number of emitted items by time $t$, then $Q_t = t - M_t$. Thus, within this scheduler family, improving the backlog bound would amount to improving the corresponding continual private counting error. Known continual-counting lower bounds suggest that logarithmic horizon dependence is intrinsic to this proof route (Dwork et al., 2010a; Cohen et al., 2024), although we do not claim a minimax lower bound for all possible NPDP wrappers.

**Scope of the Approximate-DP Regime.** The bounded-delay guarantee in Lemma 4.2 is tied to the approximate-DP implementation of RandBin: the proof uses bounded-support buffering noise and an always-valid prefix-sum error bound. For RandBin-style shift masking, pure DP would require unbounded-support noise, and therefore cannot give the same probability-one finite-delay guarantee. We discuss this scope limitation further in Appendix I.

**Positioning.** Randomized buffering is often used in oblivious data structures to hide access patterns (Chan et al., 2022; Zhou et al., 2023). Our use is different in emphasis: RandBin serves as an *infrastructure interface* that (i) converts single-edit participation streams into a sparse, Hamming-style stream of per-bin updates and (ii) exposes explicit backlog guarantees. Once Lemmas 4.1–4.2 hold, the downstream component becomes plug-and-play: any continual primitive that passes our certification in Section 4.2 can be safely reused. Full pseudocode, the NPDP security game (paired simulation + refinement), and sharper tail bounds are deferred to Appendix D.

## 4.2. Certifying Feedback-Safe Continual-Release Primitives

**Why Certification?** Our pipeline treats the continual-release DP primitive as a plug-in component. To make this modularity *auditable*, we give a simple certification criterion under which a standard *non-adaptive* Hamming-neighbor DP analysis remains valid even when per-round updates are chosen *adaptively* as a function of past releases.

**Independently Decomposable Mechanisms.** Write $x_t$ for the input update at round $t$ and $x_{1:t} := (x_1, \ldots, x_t)$. A continual-release mechanism $\mathcal{M}$ is *independently decomposable* if, at each round $t$, it samples fresh independent randomness $R_t$ and releases $y_t = \mathcal{M}_t(x_{1:t}; R_t)$ for some deterministic map $\mathcal{M}_t$. Equivalently, $y_t$ depends on the input history only through the prefix $x_{1:t}$, and all randomness used for the $t$-th release is contained in the fresh seed $R_t$ (no shared randomness across rounds).

*Importantly,* after producing the output $y_{1:t}$ at time $t$, the mechanism may apply *arbitrary* post-processing to $(y_1, \ldots, y_t)$ and output any derived value at time $t$; by clo-

sure under post-processing, this does not weaken the privacy guarantee proved for the underlying releases.

---

**Checklist 1 (Certification Conditions).** A continual-release primitive $\mathcal{M}$ passes certification if:

1. **Prefix dependence:** the release at time $t$ is a function of $x_{1:t}$ (not future inputs).

2. **Fresh randomness:** the mechanism samples independent $R_t$ at each $t$ (no reuse/correlation across rounds).

3. **Deterministic release map:** given $(x_{1:t}, R_t)$, the released output $y_t$ is deterministic.

4. **Arbitrary post-processing:** after producing $y_t$, the mechanism may output any (possibly randomized) function of the transcript $y_{1:t}$ at time $t$.

*Consequence:* any non-adaptive $(\varepsilon, \delta)$-DP proof for Hamming-neighboring update streams remains valid under adaptive interaction.

---

**Certification Theorem.** The next theorem formalizes the consequence above and serves as an audit tool for reusing continual DP primitives inside interactive learning loops.

**Theorem 4.3** (Certification: Non-Adaptive DP Implies Feedback-Safe DP)**.** *Consider Hamming-style neighboring update streams (two streams differ in at most one position). Let $\mathcal{M}$ be an independently decomposable continual-release mechanism. If $\mathcal{M}$ is non-adaptively $(\varepsilon, \delta)$-DP for Hamming-neighboring update streams, then $\mathcal{M}$ is also $(\varepsilon, \delta)$-DP against adaptive interaction, where each $x_t$ may be chosen as an arbitrary function of past releases $(y_1, \ldots, y_{t-1})$.*

**Proof Sketch.** (Full proof in Appendix E) Fresh per-round randomness prevents cross-round correlations induced by adaptivity. Formally, one conditions on the past transcript and applies the same one-step DP guarantee at each round; independence of $R_t$ ensures the conditional distributions match the non-adaptive analysis.

**Mini-Audit 1: Tree / Prefix-Sum.** View the tree mechanism as first sampling all node noises once, independently, and then answering each query $t$ by outputting the sum of the $O(\log t)$ noisy node values on the root-to-leaf cover of the prefix $[1..t]$. The *independently decomposable* core is the release of these noisy node values: each node's value depends only on the updates in its interval and its own fresh noise. The published prefix sum at time $t$ is then a deterministic function of the already-released noisy nodes (a post-processing), so by Theorem 4.3 the usual non-adaptive Hamming-neighbor DP analysis carries over to adaptive update choices.

**Mini-Audit 2: Lower-Triangular (Prefix) Linear Mechanisms.** Many continual primitives can be written as an *independently decomposable core* followed by linear post-processing. Concretely, suppose we first release a sequence of "core" noisy statistics $z_t = \langle b_t, x_{1:t} \rangle + \eta_t$, $\mathrm{supp}(b_t) \subseteq \{1, \ldots, t\}$, where the noises $\eta_t$ are fresh and independent across $t$. Then $z_t$ is independently decomposable with seed $R_t := \eta_t$ and deterministic map $\mathcal{Z}_t(x_{1:t}; R_t) = \langle b_t, x_{1:t} \rangle + R_t$. Any published output of the form $y_t = \sum_{s \leq t} A_{t,s} z_s$ (i.e., any lower-triangular linear combination, including the usual $y_t = \langle a_t, x_{1:t} \rangle + \eta_t$ as a special case) is a deterministic function of $(z_1, \ldots, z_t)$ and therefore pure post-processing. Hence any non-adaptive Hamming-neighbor DP bound for the core stream $(z_t)$ remains valid under adaptive interaction, and the same holds for the released stream $(y_t)$ by Theorem 4.3. For details, see Appendix F.1.

**Scope Boundary.** Theorem 4.3 is a sufficient black-box certification condition, not a characterization of adaptive privacy. If independent decomposability fails, a non-adaptive proof alone need not remain valid under adaptive inputs; for instance, reused or correlated randomness can be more vulnerable to adaptive choices (Denisov et al., 2022). Conversely, failure of Checklist 1 does not imply that adaptive privacy is false; it means that a separate interaction-aware proof is needed, as for RandBin in Appendix D.

### 4.3. Putting the Ingredients Together

We prove end-to-end privacy by a *modular composition* argument that composes two interactive mechanisms in series. Our general composition theorem (Appendix C) applies to any upstream wrapper that is $(\varepsilon_b, \delta_b)$-NPDP and any downstream continual primitive that is $(\varepsilon_a, \delta_a)$-DP for Hamming-neighboring update streams and is feedback-safe by certification. The pipeline in Fig. 1 is one instantiation, with RandBin as the wrapper and a continual DP aggregator as the downstream mechanism.

Section 4.1 supplies the wrapper side: RandBin implements an *edit-to-Hamming* reduction, i.e., a single insertion/deletion in the event stream induces only a bounded Hamming-style change in the induced (per-bin) update stream (Lemma 4.1). Section 4.2 supplies the downstream side: for primitives that pass Checklist 1, a standard non-adaptive DP proof for Hamming-neighboring inputs remains valid when updates are chosen adaptively from past releases (Theorem 4.3).

**Modular Composition (Informal).** View buffering as producing (i) a *visible* transcript (e.g., whether a bin is emitted) and (ii) a *hidden* interface stream of per-bin updates. The downstream continual primitive consumes this interface stream and releases the trajectory observed by the adversary.

The end-to-end guarantee follows from two conditions:

1. **Wrapper condition (NPDP).** The wrapper is $(\varepsilon_b, \delta_b)$-NPDP for single-edit neighboring event streams, and the induced interface streams can be aligned to be Hamming-neighboring (Lemma 4.1).

2. **Downstream condition (Hamming-DP + certification).** The downstream primitive is $(\varepsilon_a, \delta_a)$-DP for Hamming-neighboring update streams on its own input, and passes Checklist 1, hence remains DP under adaptive interaction (Theorem 4.3).

**Conclusion.** Under these conditions, the composed pipeline is $(\varepsilon_b + \varepsilon_a, \delta_b + \delta_a)$-DP with respect to single-edit neighboring event streams, even under adaptive interaction. This is the only composition step used to derive Theorem 3.1; the formal NPDP definition, alignment/refinement machinery, and the general modular composition theorem are deferred to Appendix C.

**Takeaway.** Sections 4.1–4.3 justify treating the pipeline as plug-and-play: once the wrapper and certification conditions hold, we can focus on the DP-SGD instantiation and evaluation.

## 5. Instantiation: Continual DP-SGD Pipeline

This section instantiates the buffering–aggregation recipe (Section 4) with streaming SGD (Algorithm 2). Each arriving user event $f_t$ is buffered by RandBin. When a bin is emitted, we form a clipped mini-batch gradient and pass a (stepsize-scaled) update to a continual-release DP aggregator. In our experiments, the aggregator PrivStreamSum is either (i) the standard tree-based prefix-sum mechanism (Chan et al., 2010), or (ii) the lower-triangular matrix factorization prefix-sum mechanism (Fichtenberger et al., 2023). Their formal DP and error/consistency guarantees are summarized in Appendix F. As in Fig. 1, the adversary observes the released models and the release timing (i.e., whether $B_t = \perp$ at each step), while bin contents and internal states are hidden.

**Fixed vs. Tuned Parameters.** The privacy target $(\varepsilon, \delta)$ and its allocation into $(\varepsilon_b, \delta_b)$ and $(\varepsilon_a, \delta_a)$ determine (i) the buffering level $U(\varepsilon_b, \delta_b)$ and (ii) the aggregation budget for PrivStreamSum. Given a learning task, the remaining choices follow standard DP-SGD practice: the clipping norm $G$ and stepsize schedule $(\eta_k)$ are tuned for utility. The release schedule is induced by RandBin: bins are emitted at random times, and satisfy $|B_t| \in [U, 2U]$ whenever

**Algorithm 2** Continual DP-SGD Under Single-Edit User Streams

1: **Input:** Privacy target $(\varepsilon, \delta)$; public horizon $T_{\max}$, i.e., maximum number of steps/time indices and hence an upper bound on the number of emitted bins; allocation $(\varepsilon_b, \delta_b)$ for buffering and $(\varepsilon_a, \delta_a)$ for aggregation; initial model $w^{(0)}$; clip norm $G$; stepsizes $(\eta_k)_{k \geq 1}$
2: **Output:** Released trajectory $(w^{(t)})_{t \geq 0}$ and release timing
3: Set $U = \Theta(\varepsilon_b^{-1} \log(1/\delta_b))$ and initialize RandBin with level $U$
4: Initialize the continual DP prefix-sum mechanism PrivStreamSum with budget $(\varepsilon_a, \delta_a)$
5: $k \leftarrow 0$
6: **for** $t = 1, 2, \ldots, T_{\max}$ **do**
7: $\quad$ Receive user event $f_t$
8: $\quad B_t \leftarrow \mathsf{RandBin}(f_t)$
9: $\quad$ **if** $B_t = \bot$ **then**
10: $\quad\quad w^{(t)} \leftarrow w^{(t-1)}$
11: $\quad$ **else**
12: $\quad\quad k \leftarrow k + 1$
13: $\quad\quad g_k \leftarrow \frac{1}{|B_t|} \sum_{f \in B_t} \mathsf{Clip}_G\left(\nabla f(w^{(t-1)})\right)$
14: $\quad\quad u_k \leftarrow \eta_k \, g_k$ $\qquad \triangleright$ *stepsize-scaled update*
15: $\quad\quad S_k \leftarrow \mathsf{PrivStreamSum}(u_k)$ $\quad \triangleright$ *private prefix sum*
16: $\quad\quad w^{(t)} \leftarrow w^{(0)} - S_k$ $\qquad \triangleright$ *post-processing of $S_k$*
17: $\quad$ **end if**
18: **end for**

$B_t \neq \bot$. The aggregator PrivStreamSum is used as a plug-in, subject to: (i) $(\varepsilon_a, \delta_a)$-DP for Hamming-neighboring update streams on its own input, and (ii) passing Checklist 1 so its non-adaptive DP analysis remains valid under adaptive interaction (Section 4.2).

**Privacy-Budget Split.** We use a symmetric $50/50$ split between buffering and aggregation as a simple default. Since privacy composes as $(\varepsilon, \delta) = (\varepsilon_b + \varepsilon_a, \delta_b + \delta_a)$ and latency scales as $U = \Theta(\varepsilon_b^{-1} \log(1/\delta_b))$, any fixed split $\varepsilon_b = c\varepsilon$, $\varepsilon_a = (1-c)\varepsilon$ with $c \in (0, 1)$ changes the asymptotic guarantees only by constant factors; assigning constant fractions of $\delta$ changes $\log(1/\delta_b)$ only by an additive $O(1)$ term. The best finite-sample split can be chosen empirically by sweeping $c$ and measuring the latency–accuracy tradeoff.

**Mini-Batch View.** When a bin is emitted, the gradient estimate uses the standard mini-batch average $\frac{1}{|B_t|} \sum_{f \in B_t} \nabla f(w)$. Since RandBin enforces $|B_t| \in [U, 2U]$, the batch size is controlled and varies by at most a factor of 2.

**Implementation Notes and Measured Latency.** The server maintains (a) the current released model $w^{(t)}$, (b) the FIFO buffer state of RandBin, and (c) the internal state of PrivStreamSum (e.g., the tree nodes). The model is updated only on emission times and is held fixed otherwise, yielding a piecewise-constant released trajectory. To quan-

tify privacy-implied latency, we log the backlog $Q_t$ (pending events after time $t$). For an event $f_i$, its inclusion delay is $D(i) := \min\{t \geq i : f_i \in B_t\} - i$, which can be computed directly from the same FIFO queue evolution underlying $(Q_t)$; we report summary statistics and empirical CDFs in the experiments.

**Other Optimizers.** The same template applies to streaming optimizers whose per-bin updates can be encoded as a clipped, sensitivity-bounded stream for the certified continual-release aggregator: replace $g_k = \mathsf{Clip}(\cdot)$ and $u_k$ by the desired update encoding, while keeping RandBin as the edit-to-Hamming wrapper and PrivStreamSum as the aggregation primitive. Variants with additional state, such as momentum or adaptive preconditioning, require a separate audit of Theorem 4.3's conditions; falling outside the theorem does not by itself imply failure of adaptive privacy. Optimizer-specific privacy-loss saturation results are complementary to our backlog analysis; see Appendix I.

# 6. Applications and Experiments

We report experiments to demonstrate the framework.[1] The main text focuses on deep networks trained with SGD under different aggregation mechanisms. In Appendix G.1, we give an attack showing that a Hamming-style continual release mechanism (e.g., tree-based prefix sums) is not adaptive to single-edit streams; in Appendix G.2, we validate the delay phenomenon introduced in Section 4; in Appendix G.4, we give a synthetic ADMM study that fits our setting.

## 6.1. Deep-Network Experiments

We evaluate Algorithm 2 in a deep-network setting to study the utility impact of the randomized buffering wrapper RandBin under single-edit user streams. We use MNIST, CIFAR-10, and EMNIST ByMerge, where simple networks achieve strong accuracy but the DP variants remain informative.

**Model and Algorithm.** We follow the framework of Kairouz et al. (2021), adapting its training and accounting pipeline to our continual-release setting. We use a small MLP for MNIST/EMNIST and VGG-128 (Simonyan & Zisserman, 2015) for CIFAR-10. Our algorithms instantiate Algorithm 2 with different choices of the prefix-sum mechanism PrivStreamSum, which determines how noisy gradients are aggregated over time:

(i) `sgd_nodp`: non-private baseline without noise;
(ii) `sgd_naive_dp`: i.i.d. Gaussian noise to each batch;

[1]Our source code is available at:
https://github.com/MengshiZ/ICML2026.

(iii) `sgd_tree_dp`: standard tree-based DP prefix-sum;

(iv) `sgd_matrix_dp`: DP prefix-sum based on the lower-triangular matrix mechanism of Fichtenberger et al. (2023).

We evaluate each method with and without the randomized buffering wrapper RandBin: without RandBin the stream is treated under Hamming-style neighbors, while with RandBin we obtain single-edit privacy. We append `_hamming` or `_edit` to indicate whether RandBin is disabled or enabled.

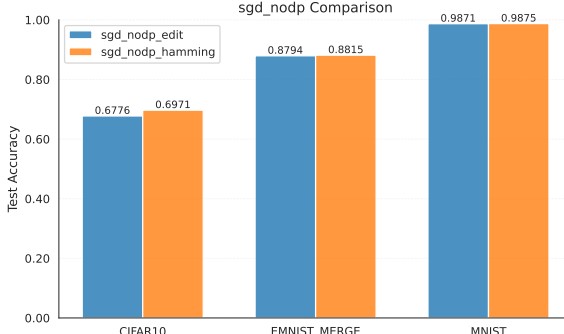

*Figure 2.* Comparison of `sgd_nodp_hamming` and `sgd_nodp_edit` baselines across MNIST, CIFAR-10, and EMNIST. This demonstrates that the RandBin wrapper incurs negligible accuracy overhead when converting from Hamming-style to single-edit privacy across all datasets.

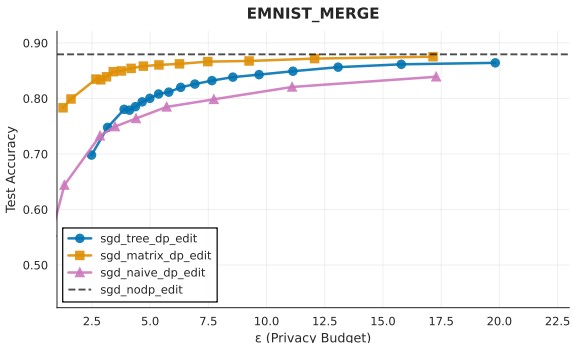

*Figure 3.* Accuracy–privacy tradeoff on EMNIST ByMerge with single-edit privacy. All algorithms use the RandBin wrapper (`sgd_naive_dp_edit`, `sgd_tree_dp_edit`, and `sgd_matrix_dp_edit`). Structured aggregation mechanisms improve over naive per-minibatch noise; MNIST and CIFAR-10 results are deferred to the appendix.

**Privacy Parameter Setup.** Instead of fixing $(\varepsilon, \delta)$ in advance, we treat the noise standard deviation as a hyperparameter and track RDP during training, then convert it to $(\varepsilon, \delta)$ at a fixed $\delta$; see Section F.1 for the details. By sweeping the noise multiplier, we obtain accuracy–privacy tradeoff curves. We fix $\delta = 10^{-5}$ for MNIST/CIFAR-10 and $\delta = 10^{-6}$ for EMNIST.

**Implementation Details.** We compare performance with and without RandBin. When RandBin is enabled, batch sizes vary in $[U, 2U]$; when it is disabled, we use a fixed batch size equal to the mean, $3U/2$, to match expected computation. For EMNIST and CIFAR-10 we set the mean batch size to 500, and for MNIST to 250.

**Single-Edit Privacy Cost.** Enabling RandBin (single-edit privacy) introduces only negligible accuracy loss relative to the corresponding Hamming-style baselines (`sgd_nodp_hamming` vs. `sgd_nodp_edit`), as shown in Figure 2 across all three datasets. This validates our core contribution: the randomized buffering approach achieves stronger privacy guarantees (single-edit versus Hamming neighbors) with nearly no utility cost.

**Discussion of Results.** Figure 3 presents the accuracy–privacy tradeoff on EMNIST ByMerge, for all three aggregation mechanisms under single-edit privacy. Similar patterns hold for MNIST and CIFAR-10, deferred to Appendix G.3. Across datasets we observe two consistent trends. First, as $\varepsilon$ increases, the tree mechanism increasingly outperforms the naive per-minibatch noise approach, consistent with findings in Kairouz et al. (2021). Second, the matrix mechanism of Fichtenberger et al. (2023) exhibits the most robust performance across the entire privacy spectrum, providing the best accuracy–privacy tradeoffs.

# 7. Conclusion

We studied continual learning under adaptive interaction when privacy is defined over *edit-style* (single-insertion/deletion) user streams, where naive continual-release analyses do not directly apply. Our main recipe composes a randomized buffering wrapper RandBin, which reduces single-edit neighbors to a Hamming-style per-bin update interface with explicit backlog guarantees, with a certified continual-release DP primitive (tree-based prefix sums in our experiments), and then invokes modular composition to obtain end-to-end $(\varepsilon, \delta)$-DP.

Empirically, the privacy target determines $U(\varepsilon_b, \delta_b)$ and thus a concrete delay regime, and we observe the resulting utility–latency tradeoff across privacy levels. Beyond streaming SGD, the same buffering–aggregation interface can instantiate other optimizers and continual primitives (e.g., ADMM-style updates), which we defer to Appendix G.4. Further discussion on limitations is given in Appendix I.

## Acknowledgements

T-H. Hubert Chan is partially supported by the Hong Kong RGC grants 17201823 and 17203725. Elaine Shi is supported in part by NSF under award numbers 2128519 and 2212746, a Packard Fellowship, and an ONR grant. Mingxun Zhou is partially supported by a startup grant from the Hong Kong University of Science and Technology.

## Impact Statement

**Scope.** This paper develops an auditable recipe for *continual-release* differential privacy in streaming/federated learning loops under *single-edit* participation changes. We focus on direct and tractable impacts arising from (i) protecting user participation over time and (ii) the explicit privacy–latency link induced by buffering.

**Potential Benefits.** Our mechanisms can reduce privacy risks in systems that repeatedly publish intermediate model updates (or derived statistics) by providing a principled trajectory-level guarantee under adaptive feedback. This may improve the safety of deploying learning systems in sensitive domains (e.g., health, finance, and user-facing personalization) and can support compliance efforts by making privacy guarantees more modular and checkable.

**Potential Risks and Limitations.** (1) *Misuse or overclaiming:* practitioners might treat the guarantee as a blanket protection against all forms of leakage (e.g., membership or attribute inference) even when deployment deviates from the assumed threat model, composition accounting, or noise calibration. (2) *Privacy washing:* stronger stated guarantees could be used to justify collecting more data or increasing retention without sufficient governance. (3) *Systems impact:* buffering introduces delay/backlog; in real deployments this may reduce responsiveness, create incentives to bypass privacy mechanisms, or shift risk to other system components. (4) *Residual risk:* differential privacy bounds expected leakage but does not eliminate it; careful parameter selection and monitoring remain necessary.

**Uncertainties.** Downstream impact depends on how the recipe is integrated into end-to-end pipelines (client sampling, clipping, aggregation, accounting) and on operational constraints that may differ across deployments.

**Initiatives.** We encourage (i) clear documentation of assumptions and privacy parameters, (ii) deployment checklists/tests that certify the required modular conditions (fresh noise, accounting, and threat model alignment), and (iii) empirical evaluation of privacy–utility–latency tradeoffs to discourage unsafe workarounds.

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

# Appendix

## A. Detailed Preliminaries

We formalize the setting as outlined in the introduction. We first describe the concept of an *abstract* interactive mechanism, from which we will derive other forms of interactive mechanisms later.

**Abstract Interactive Mechanism.** We use $\mathcal{U}$ to denote the collection of all valid messages, where $\{\bot, \mathsf{halt}\} \subseteq \mathcal{U}$. We use $\mathcal{R}$ to denote the collection of random seeds. In this work, we focus on the case that both $\mathcal{U}$ and $\mathcal{R}$ are finite.

An interactive mechanism $\mathcal{M}$ is specified by a distribution $R_{\mathcal{M}}$ on $\mathcal{R}$ and a transition function $\mathcal{M} : \mathcal{R} \times \mathcal{U}^* \to \mathcal{U}$, where $\mathcal{U}^* := \cup_{i \in \mathbb{N}} \mathcal{U}^i$, with the convention that $\mathcal{U}^0$ is the identity under Cartesian product, i.e., $\mathcal{R} \times \mathcal{U}^0 = \mathcal{R}$. Note that we sometimes overload the notation such that $\mathcal{M}$ denotes both the mechanism and the corresponding transition function. Here is the functionality of an interactive mechanism.

---

1. The mechanism $\mathcal{M}$ samples a (secret) random seed $r \in \mathcal{R}$ according to the distribution $R_{\mathcal{M}}$.

2. At time step $t = 0$, the mechanism outputs $\mathcal{M}(r) \in \mathcal{U}$. If the mechanism is supposed to wait for the first input message, we use the convention that $\mathcal{M}(r) = \bot$.

3. At time step $t \geq 1$, suppose a message $x_t \in \mathcal{U}$ (which may be chosen depending on the history before time $t$) is sent to the mechanism. We use $x[1..t] := (x_1, x_2, \ldots, x_t) \in \mathcal{U}^t$ to denote the messages received by the mechanism from the previous time steps up to $t$.

   Then, at this time step, the mechanism outputs $\mathcal{M}(r; x[1..t]) \in \mathcal{U}$.

4. If, for some $t$, $\mathcal{M}(r; x[1..t]) = \mathsf{halt}$, then the mechanism halts at time step $t$.

---

**Instantiation: Mapping the Abstract Interaction to Our Transcript.** The abstract interaction formalism above subsumes the continual-learning setting of Section 2. Concretely, we instantiate the mechanism's input messages as the user-side stream $x_t := f_t$ (or, when modeling a subcomponent, $x_t$ may represent an emitted bin $B_t$ or a per-bin update $g_t$), and we instantiate the mechanism's outputs as the released observations $y_t := (r_t, w^{(t)})$, so that the induced transcript is $\mathsf{tr}_{\leq T} = ((r_t, w^{(t)}))_{t=0}^T$. The adversary's view function $\nu_{\mathcal{A}}$ reveals exactly the observable portion of each release—the timing bit and model snapshot—matching the trajectory-and-timing threat model in Section 2.

**Bounded Termination.** In addition to finite $\mathcal{R}$ and $\mathcal{U}$, we consider mechanisms that always terminate in a bounded number of steps. Specifically, for each mechanism $\mathcal{M}$, there exists $T_{\mathcal{M}} \in \mathbb{Z}$ such that $\mathcal{M}$ always terminates in at most $T_{\mathcal{M}}$ steps. This is without loss of generality, because any privacy notion defined for mechanisms with bounded termination can be extended naturally to unbounded mechanisms. We simply require that, for any $T \in \mathbb{Z}_+$, the truncated mechanism obtained by running the (unbounded) mechanism for $T$ steps will satisfy the privacy notion defined for bounded mechanisms. This matches the "for every horizon $T$" convention in Section 2.

*Mechanism View.* The view observed by a mechanism $\mathcal{M}$ up to time step $t$ consists of its generated random seed $r$ and its received input messages $x[1..t]$ up to time $t$.

**Remark A.1.** Since $\mathcal{M}$ can recover its own output sequence from its random seed and input sequence, it suffices to include the latter two in its view.

**Similarity Between Mechanisms.** To formalize privacy notions later, we need a way to quantify the similarity between two mechanisms. Since we consider randomized mechanisms, we will use power functions to compare how different two distributions are. It is known that power functions are general enough to capture all divergences satisfying the data processing inequality. Recall that in this work, we focus on finite sample spaces.

**Definition A.2** (Data Processing Inequality). A divergence measure is a function D that takes two distributions $P$ and $Q$ on the same space such that $\mathsf{D}(P\|Q) \geq 0$, where equality holds *iff* the distributions $P = Q$ are identical.

A divergence D satisfies the *data processing inequality* if, for any stochastic transformation (or channel) $T$ that maps the original space to another space, the following inequality holds:

$$D(T(P)\|T(Q)) \leq D(P\|Q),$$

where $T(P)$ and $T(Q)$ are the resulting distributions after applying the transformation $T$ to $P$ and $Q$, respectively.

**Definition A.3** (Power Function as Fractional Knapsack Problem ([Kadane, 1968](#))). Suppose $P$ and $Q$ are distributions on the same finite sample space $\Omega$, i.e., $P$ and $Q$ are vectors in $\mathbb{R}_{\geq 0}^{\Omega}$ whose coordinates sum to 1. The power function $\mathsf{Pow}(P\|Q) : [0,1] \to [0,1]$ can be defined in terms of the *fractional knapsack problem*.

Given a collection $\Omega$ of items, suppose $\omega \in \Omega$ has weight $P(\omega)$ and value $Q(\omega)$. Then, given $\alpha \in [0,1]$, $\mathsf{Pow}(P\|Q)(\alpha)$ is the maximum value attained with weight capacity constraint $\alpha$, where items may be taken fractionally.

*Intuition.* When two distributions $P$ and $Q$ are the same, it is clear that given any capacity constraint $\alpha$, the maximum reward is also $\alpha$; this means the power function is exactly the identity function. However, if the two distributions are very different, this means that there are items whose reward-to-weight ratios are large; hence, in this case, the power function can initially grow faster than the identity function.

**Partial Order on Power Functions.** Pointwise comparison naturally induces a partial order on power functions. We denote $f_1 \leq f_2$ if for all $\alpha \in [0,1]$, $f_1(\alpha) \leq f_2(\alpha)$, where a larger $f_2$ indicates that the two distributions are more different.

**Fact A.4** (Properties of Power Functions). *For any two distributions $P$ and $Q$ on the same sample space, $\mathsf{Pow}(P\|Q) \geq \mathsf{Id}$, where $\mathsf{Id} : [0,1] \to [0,1]$ is the identity function, and equality holds iff the distributions $P = Q$ are identical.*

*Moreover, power functions satisfy the* data processing inequality, *i.e., for any stochastic transformation (or channel) $T$ that maps the original space to another space, the following inequality holds:*

$$\mathsf{Pow}(T(P)\|T(Q)) \leq \mathsf{Pow}(P\|Q).$$

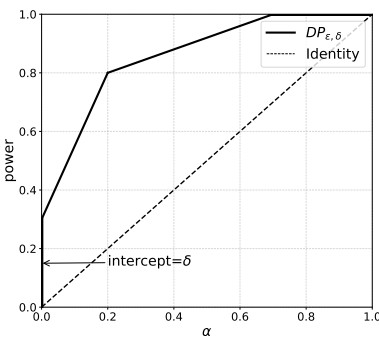

*Figure 4.* Power function $\mathsf{DP}_{\varepsilon,\delta}$ for $\varepsilon = 1$ and $\delta = 0.3$.

**Differential Privacy in Terms of Power Function.** Given $\varepsilon \geq 0$ and $0 \leq \delta \leq 1$, the power function $\mathsf{DP}_{\varepsilon,\delta} : [0,1] \to [0,1]$ can be described as follows (see Figure [4](#)):

1. In the $xy$-plane, starting at $(0,\delta)$, the function increases linearly with slope $e^{\varepsilon}$ until it touches the line $y = 1 - x$ at $\left(\frac{1-\delta}{1+e^{\varepsilon}}, \frac{e^{\varepsilon}+\delta}{1+e^{\varepsilon}}\right)$.

2. After touching the line $y = 1 - x$, the slope changes to $e^{-\varepsilon}$, until the line segment reaches $(1-\delta, 1)$, where the curve remains horizontally till the end $(1,1)$ is reached.

In other words,

$$\mathsf{DP}_{\varepsilon,\delta}(\alpha) = \begin{cases} \delta + e^{\varepsilon} \cdot \alpha, & \text{for } 0 \leq \alpha \leq \frac{1-\delta}{1+e^{\varepsilon}}; \\ 1 - e^{-\varepsilon}(1-\delta) + e^{-\varepsilon}\alpha, & \text{for } \frac{1-\delta}{1+e^{\varepsilon}} < \alpha \leq 1 - \delta; \\ 1, & \text{for } 1 - \delta < \alpha \leq 1. \end{cases}$$

Below are some facts on power functions that we will use, which are usually equivalently stated in the literature in terms of the tradeoff function $\mathsf{T}(P\|Q)(\alpha) := 1 - \mathsf{Pow}(P\|Q)(\alpha)$.

**Fact A.5** (DP in Terms of Power Function (Dong et al., 2022)). *Given two distributions $P$ and $Q$ on the same sample space $\Omega$, $\mathsf{Pow}(P\|Q) \leq \mathsf{DP}_{\varepsilon,\delta}$ is equivalent to the statement that for all $S \subseteq \Omega$, $P(S) \leq e^\varepsilon \cdot Q(S) + \delta$ and $Q(S) \leq e^\varepsilon \cdot P(S) + \delta$.*

**Fact A.6.** *(Triangle Inequality for Power Functions)[Theorem 2.14 in (Dong et al., 2022)] Suppose $X, Y, Z$ are distributions on the same sample space such that $\mathsf{Pow}(X\|Y) \leq g_1$ and $\mathsf{Pow}(Y\|Z) \leq g_2$. Then, $\mathsf{Pow}(X\|Z) \leq g_2 \circ g_1$, where the composition is defined as $(g_2 \circ g_1)(x) := g_2(g_1(x))$. In other words, $\mathsf{Pow}(X\|Z) \leq \mathsf{Pow}(Y\|Z) \circ \mathsf{Pow}(X\|Y)$.*

**Example.** Consider $g = \mathsf{DP}_{\varepsilon,\delta}$. Then, $g \circ g \leq \mathsf{DP}_{2\varepsilon,\delta(1+e^\varepsilon)}$. As we shall see, this is relevant to the scenario of *multi-hop* neighbors – *aka group* DP in the literature.

**Definition A.7** (Supremum of Power Functions). Given a collection $\mathcal{S}$ of power functions, its supremum $\sup(\mathcal{S}) = \sup_{f \in \mathcal{S}} f$ is the least power function $\widehat{f}$ such that for all $\mathfrak{f} \in \mathcal{S}$, $f \leq \widehat{f}$.

Formally, for each $\alpha \in [0, 1]$, $\widehat{f}(\alpha) :=$

$$\sup\{\lambda \cdot f_1(\alpha_1) + (1 - \lambda) \cdot f_2(\alpha_2) \mid f_1, f_2 \in \mathcal{S}; \alpha_1, \alpha_2, \lambda \in [0, 1] : \alpha = \lambda\alpha_1 + (1 - \lambda)\alpha_2\}.$$

**Definition A.8** (Tensor Product of Power Functions). Suppose $f$ and $g$ are power functions that are attained by the corresponding distributions in $\mathsf{Pow}(P_1\|P_2) = f$ and $g = \mathsf{Pow}(Q_1\|Q_2)$. Then, their corresponding tensor product $f \otimes g = \mathsf{Pow}(P_1 \times Q_1 \| P_2 \times Q_2)$ is defined in terms the corresponding product distributions.

**Fact A.9** (Joint Convexity (Dong et al., 2022)). *Suppose $\Lambda$ is an indexing set and $f$ is a power function such that for each $\lambda \in \Lambda$, a pair $(P(\lambda), Q(\lambda))$ of distributions satisfy $\mathsf{Pow}(P(\lambda)\|Q(\lambda)) \leq f$.*

*Then, for any distribution $L$ on $\Lambda$, $\mathsf{Pow}(P(L)\|Q(L)) \leq f$.*

**Fact A.10** (Composition Rule of Power Functions (Dong et al., 2022)). *Suppose $P$ an $P'$ are distributions on the sample space $\Omega$. For each $\omega \in \Omega$, $Q(\omega)$ and $Q'(\omega)$ are distributions on sample space $\Omega'$. If $\mathsf{Pow}(P\|P') \leq f$ and $\mathsf{Pow}(Q(\omega)\|Q'(\omega)) \leq g$ for all $\omega \in \Omega$, then $\mathsf{Pow}((P, Q(P))\|(P', Q'(P'))) \leq f \otimes g$.*

*Distinguishing between Interactive Mechanisms.* Suppose we wish to compare the behavior of two mechanisms $\mathcal{M}_0$ and $\mathcal{M}_1$, to which we only have oracle accesses. An adversary $\mathcal{A} : \mathcal{R} \times \mathcal{U}^* \to \mathcal{U}$ is also an interactive mechanism. The adversary has some view function $\nu_\mathcal{A} : \mathcal{U} \to \mathcal{U}$ that specifies what information it can gather from the output of the mechanism. However, later when we instantiate from the abstract mechanism, we will use the structure of the message space to clarify what the adversary can observe.

We next define the interaction $\mathcal{A} \leftrightarrow \mathcal{M}$ between the two mechanisms. The adversary $\mathcal{A}$ repeatedly sends messages to $\mathcal{M}$. At each time step, the adversary determines the next message to be sent to $\mathcal{M}$, based on the history of prior interactions. This process repeats iteratively, until the mechanism $\mathcal{M}$ outputs halt in some time step. Formally, it is captured by the following description.

---

1. The mechanisms $\mathcal{A}$ and $\mathcal{M}$ sample the random seeds $r_\mathcal{A}$ and $r_\mathcal{M}$, respectively.

2. At time step $t = 0$, $\mathcal{M}$ outputs $y_0 \leftarrow \mathcal{M}(r_\mathcal{M}) \in \mathcal{U}$.

   $\mathcal{A}$ observes $v_0 := \nu_\mathcal{A}(y_0)$.

3. At time step $t \geq 1$,

   (a) $\mathcal{A}$ generates $x_t \leftarrow \mathcal{A}(r_\mathcal{A}; v[0..t-1])$ and sends to $\mathcal{M}$.
   (b) $\mathcal{M}$ responds $y_t \leftarrow \mathcal{M}(r_\mathcal{M}; x[1..t])$;
       $\mathcal{A}$ observes $v_t := \nu_\mathcal{A}(y_t)$.

   The process terminates as soon as either $x_t$ or $y_t$ is halt; otherwise, the process continues for the next time step.

---

**Definition A.11** (Adversary View). Suppose the interaction $\mathcal{A} \leftrightarrow \mathcal{M}$ terminates at time step $t$ (which may be a random quantity). Then, the adversary view is defined as:

$\mathsf{view}(\mathcal{A} \leftrightarrow \mathcal{M}) := (r_\mathcal{A}; v[0..t])$.

*Terminology ("non-adaptive").* In this paper, when we say "non-adaptive" for a continual mechanism's privacy analysis, we mean the mechanism's input stream is fixed in advance (equivalently, the environment/adversary that provides inputs does not condition on past outputs).

**Definition A.12** (Non-Adaptive Adversary). An adversary transition function $\mathcal{A} : \mathcal{R} \times \mathcal{U}^* \to \mathcal{U}$ is non-adaptive, if there exists a transition function of the form $\widehat{\mathcal{A}} : \mathcal{R} \times \mathbb{Z} \to \mathcal{U}$ such that for any length-$t$ sequence $v[0..t-1]$, $\mathcal{A}(r; v[0..t-1]) = \widehat{A}(r; t)$.

We can next define a distance notion between two mechanisms based on power functions. Intuitively, it quantifies the difficulty of distinguishing between two mechanisms.

**Definition A.13** (Power Function between Mechanisms). Given two interactive mechanisms $\mathcal{M}_0$ and $\mathcal{M}_1$, we overload the notation and denote $\mathsf{Pow}\,(\mathcal{M}_0 \| \mathcal{M}_1) \leq f$, for some power function $f$, if for all adversaries $\mathcal{A}$, $\mathsf{Pow}\,(\mathsf{view}(\mathcal{A} \leftrightarrow \mathcal{M}_0) \| \mathsf{view}(\mathcal{A} \leftrightarrow \mathcal{M}_1)) \leq f$.

If we have the weaker requirement that the inequality holds for just non-adaptive adversaries, we denote: $\mathsf{Pow}^{\mathrm{NA}}\,(\mathcal{M}_0 \| \mathcal{M}_1) \leq f$.

**Simulation of Two Close Interactive Mechanisms with Close Random Seeds.** The following result from (Vadhan & Zhang, 2023) states that if two mechanisms are close in the sense of Definition A.13, then it is possible to simulate either one of them exactly with the same mechanism transition function that initially takes a random seed generated from one of the two appropriate distributions. Because the proof uses induction and Cartesian products of sample spaces, only the special case of finite sample and message spaces and mechanisms with bounded termination is considered. It is plausible that the result could be generalized to unbounded spaces, but that would likely require the Axiom of Choice, which often falls outside the scope of the computer science community.

This is a very useful lemma in proving privacy composition results, because instead of interacting with two potentially different mechanisms in every time step, we can view the adversary as interacting with the same mechanism transition function but with initial random seeds drawn from two different distributions. One useful analogy is that after a sensitive database is anonymized with appropriate noise, then it can be queried unlimited number of times without privacy degradation because of data post-processing.

**Lemma A.14** (Exact Simulation of Two Close Interactive Mechanisms (Vadhan & Zhang, 2023), Theorem 1.5). *Suppose $\mathcal{M}_0$ and $\mathcal{M}_1 : \mathcal{R} \times \mathcal{U}^* \to \mathcal{U}$ are two interactive mechanisms with bounded termination, and $f$ is a power function such that $\mathsf{Pow}(\mathcal{M}_0 \| \mathcal{M}_1) \leq f$ in the sense of Definition A.13.*

*Then, there exist two distributions $N_0$ and $N_1$ on some sample space $\Omega$ and a transition function $\mathcal{P} : \Omega \times \mathcal{U}^* \to \mathcal{U}$ such that $\mathsf{Pow}(N_0 \| N_1) \leq f$, and, for all adversaries $\mathcal{A}$, the following holds for both $b \in \{0, 1\}$: $\mathsf{view}(\mathcal{A} \leftrightarrow \mathcal{M}_b) \equiv \mathsf{view}(\mathcal{A} \leftrightarrow \mathcal{P}(N_b))$,*

*where $\mathcal{P}(N)$ is the interactive mechanism that uses the specified transition function $\mathcal{P}$ with a random seed drawn from the distribution $N$.*

## B. Modular Composition of Interactive Mechanisms with Hidden Outputs

**Interactive Mechanism with Hidden Output.** We next consider a special form of interactive mechanism with a transition function of the form $\mathcal{M} : \mathcal{R} \times \mathcal{X}^* \to \mathcal{V} \times \mathcal{Y}$. Here, we use $\mathcal{X}$ to denote the input space. In each time step, the mechanism produces $(v, y) \in \mathcal{V} \times \mathcal{Y}$, where the adversary can observe $v$ from the view space $\mathcal{V}$ and the output $y$ from the output space $\mathcal{Y}$ is potentially hidden from the adversary, which may be passed to another interactive mechanism (that could however possibly leak information about $y$). If an adversary only interacts with $\mathcal{M}$, then it can be described by a transition function with the form $\mathcal{A} : \mathcal{R} \times \mathcal{V}^* \to \mathcal{X}$.

**Neighboring Input Sequences.** Similar to the conventional differential privacy, we need a neighboring notion for input sequences such that intuitively the behaviors of a mechanism on two neighboring input sequences should be similar.

**Assumption B.1** (Prefix-Closed Neighboring Relation). We assume there is a neighboring relation $\sim_{\mathcal{X}^*}$ (or just $\sim$ for succinctness) that satisfies the property that if two sequences $x[1..t] \sim x'[1..t']$ are neighboring, then the following holds:

(i) They have the same length, i.e., $t = t'$.
(ii) All their corresponding prefixes are neighboring, i.e., for all $1 \leq \tau < t$, $x[1..\tau] \sim x'[1..\tau]$.

*Privacy for Interactive Mechanisms.* Since the inputs for an interactive mechanism in later time steps can be influenced by its earlier behavior, one cannot formally define its privacy just based on neighboring input sequences. In the literature, *adaptive* differential privacy (Jain et al., 2023) is based on a construction that we name as *paired simulation*.

**Definition B.2** (Paired Simulation). Suppose an interactive mechanism is given by the transition function $\mathcal{M} : \mathcal{R} \times \mathcal{X}^* \to \mathcal{V} \times \mathcal{Y}$ and the random seed distribution $R_{\mathcal{M}}$, where a neighboring relation is defined on $\mathcal{X}^*$.

Then, the paired simulation consists of a canonical pair of interactive mechanisms $(\mathcal{M}_0^{\mathsf{pair}}, \mathcal{M}_1^{\mathsf{pair}})$ such that for each $b \in \{0, 1\}$, the transition function has the form $\mathcal{M}_b^{\mathsf{pair}} : \mathcal{R} \times (\mathcal{X}^2)^* \to \mathcal{V} \times \mathcal{Y}$. Moreover, in the paired simulation, an adversary with transition function $\mathcal{A} : \mathcal{R} \times \mathcal{V}^* \to \mathcal{X}^2$ interacts with $\mathcal{M}_b^{\mathsf{pair}}$ as follows.

---

The interaction between an adversary $\mathcal{A}$ and $\mathcal{M}_b^{\mathsf{pair}}$:

1. $\mathcal{A}$ and $\mathcal{M}$ sample the hidden random seeds $r_{\mathcal{A}}$ and $r_{\mathcal{M}}$ from their own distributions, respectively.

2. At time step $t = 0$, the simulation calls $(v_0, y_0) \leftarrow \mathcal{M}(r_{\mathcal{M}})$. If $\mathcal{M}$ is supposed to wait for the first input message, $(v_0, y_0) = (\bot, \bot)$.

3. At time step $t \geq 1$, $\mathcal{A}$ constructs a pair $(x_t^{(0)}, x_t^{(1)}) \leftarrow \mathcal{A}(r_{\mathcal{A}}; v[0..t-1]) \in \mathcal{X}^2$ and sends to the simulation.

   If the sequences $x^{(0)}[1..t]$ and $x^{(1)}[1..t]$ generated so far are not neighboring, the whole simulation terminates.

   Otherwise, $(v_t, y_t) \leftarrow \mathcal{M}(r_{\mathcal{M}}; x^{(b)}[1..t])$, and $\mathcal{A}$ observes $v_t$.

   If either $\mathcal{A}$ or $\mathcal{M}$ decides to halt, then the whole simulation terminates.

   The adversarial view is denoted as $\mathsf{view}(\mathcal{A} \leftrightarrow \mathcal{M}_b^{\mathsf{pair}})$. If the simulation does not terminate before time step $t$, then the view up to time $t$ includes the random seed $r_{\mathcal{A}}$ and $v[0..t]$.

---

**Definition B.3** (Differentially Private (DP) View). Given an interactive mechanism $\mathcal{M}$, suppose $(\mathcal{M}_0^{\mathsf{pair}}, \mathcal{M}_1^{\mathsf{pair}})$ is the paired simulation as in Definition B.2. For a power function $f$, the mechanism $\mathcal{M}$ is adaptively $f$-differentially private (DP) if

$\mathsf{Pow}(\mathcal{M}_0^{\mathsf{pair}} \| \mathcal{M}_1^{\mathsf{pair}}) \leq f$, for all adaptive adversaries in the sense of Definition A.13. Similarly, non-adaptively $f$-DP is defined by $\mathsf{Pow}^{\mathsf{NA}}(\mathcal{M}_0^{\mathsf{pair}} \| \mathcal{M}_1^{\mathsf{pair}}) \leq f$.

**Multi-Hop Neighboring Notion.** In the literature, group DP or multi-hop neighbors can be generalized to sequences readily.

**Definition B.4** ($k$-Hop Neighboring Sequences). For $k = 1$, 1-hop neighboring is the same as that in Assumption B.1. For $k \geq 2$, two sequences $x$ and $x'$ are $k$-hop neighboring if there exist $x_1, \dots, x_{k-1}$ such that for $0 \leq i < k$, $x_i$ and $x_{i+1}$ are 1-hop neighboring, where $x_0 = x$ and $x_k = x'$.

We also generalize Definition B.2 to multi-hop neighboring simulation.

**Definition B.5** ($k$-Hop Neighboring Simulation). Generalizing the description in Definition B.2, in each time step $t$, the adversary generates a tuple $(x_t^{(0)}, x_t^{(1)}, \dots, x_t^{(k)})$ such that for each $0 \leq i < k$, $x^{(i)}[1..t]$ and $x^{(i+1)}[1..t]$ are neighboring sequences.

Moreover, for $b \in [0..k]$, $\mathcal{M}_b^{\mathsf{hop}}$ is a simulation of $\mathcal{M}$ that takes the $b$-th component $x_t^{(b)}$ at each step $t$.

**Fact B.6** (Adaptive DP for Multi-Hop Neighboring Sequences). *Suppose $\mathcal{M}$ is adaptively $f$-DP as in Definition B.3. Then, in a $k$-hop neighboring simulation as in Definition B.5, we have:* $\mathsf{Pow}(\mathcal{M}_0^{\mathsf{hop}} \| \mathcal{M}_k^{\mathsf{hop}}) \leq f^{\circ k}$,

*where $f^{\circ 1} = f$ and $f^{\circ(i+1)} = f^{\circ i} \circ f$.*

*Proof.* This follows from Definition B.3 and Fact A.6. $\square$

**Modular Composition of Interactive Mechanisms.** As mentioned above, the hidden output of one mechanism may be fed as the input of another mechanism. To this end, we define a specific form of composition that we call *modular*

*composition*. The adversary only directly supplies an input to the "head" mechanism of the list, where each intermediate mechanism receives its input from the (hidden) output from the previous mechanism.

---

*Modular composition* is an interactive mechanism with a transition function of the form $(\mathcal{M} \to \mathcal{N}) : \mathcal{R}_1 \times \mathcal{R}_2 \times \mathcal{X}^* \to \mathcal{V} \times \mathcal{W} \times \mathcal{Z}$ described as follows.

1. The mechanisms $\mathcal{A}$, $\mathcal{M}$ and $\mathcal{N}$ sample the hidden random seeds $r_\mathcal{A}$, $r_\mathcal{M}$, $r_\mathcal{N}$ from their own distributions, respectively.

2. At time step $t = 0$, call the subroutine $(v_0, y_0) \leftarrow \mathcal{M}(r_\mathcal{M})$ and $(w_0, z_0) \leftarrow \mathcal{N}(r_\mathcal{N})$.

   The adversary observes $(v_0, w_0)$.

3. At time step $t \geq 1$,

   (a) the adversary generates $x_t \leftarrow \mathcal{A}(r_\mathcal{A}; v[0..t-1], w[0..t-1])$, and sends it to the head mechanism of the modular composition $(\mathcal{M} \to \mathcal{N})$.
   (b) call the subroutine $(v_t, y_t) \leftarrow \mathcal{M}(r_\mathcal{M}; x[1..t])$ and pass the hidden output $y_t$ to the next mechanism $\mathcal{N}$;
   (c) call the subroutine $(w_t, z_t) \leftarrow \mathcal{N}(r_\mathcal{N}; y[1..t])$;
   (d) the adversary observes $(v_t, w_t)$.

   The whole process terminates as soon as one of $\mathcal{A}$, $\mathcal{M}$ and $\mathcal{N}$ terminates. Otherwise, the process continues for the next time step.

   If the process terminates at time step $t$, the view of the adversary consists of its random seed $r_\mathcal{A}$, $v[0..t]$ and $w[0..t]$.

---

**Definition B.7** (Modular Composition)**.** Suppose two interactive mechanisms are given by the transition functions $\mathcal{M} : \mathcal{R}_1 \times \mathcal{X}^* \to \mathcal{V} \times \mathcal{Y}$ and $\mathcal{N} : \mathcal{R}_2 \times \mathcal{Y}^* \to \mathcal{W} \times \mathcal{Z}$, with random seed distributions $R_\mathcal{M}$ and $R_\mathcal{N}$, respectively. Then, an adversary with transition function $\mathcal{A} : \mathcal{R} \times (\mathcal{V} \times \mathcal{W})^* \to \mathcal{X}$ interacts with the modular composition $\mathcal{M} \to \mathcal{N}$ as above.

*Neighbor-Preserving Paired Simulation.* To consider composition of differentially oblivious mechanisms (Chan et al., 2022), the framework neighbor-preserving differential obliviousness (NPDO) (Zhou et al., 2023; 2024) has been introduced to formalize the condition that if neighboring inputs are submitted to a mechanism, then the corresponding outputs would somehow be neighboring for a subsequent mechanism. To generalize this to interactive mechanisms, we will need to introduce neighbor-preserving paired simulation. In addition to neighboring input sequences in $\mathcal{X}^*$ for mechanism $\mathcal{N}$, we assume that there is a similar notion of neighboring sequences in $\mathcal{Y}^*$ (satisfying Assumption B.1) for mechanism $\mathcal{N}$.

**Definition B.8** (Neighbor-Preserving Paired Simulation (NPP))**.** Given an interactive mechanism with transition function $\mathcal{M} : \mathcal{R} \times \mathcal{X}^* \to \mathcal{V} \times \mathcal{Y}$ and random seed distribution $R_\mathcal{M}$, a neighbor-preserving paired simulation (which might not be unique) is a pair $(\mathcal{M}_0^{\mathsf{npp}}, \mathcal{M}_1^{\mathsf{npp}})$ of interactive mechanisms such that the following holds.

1. There exists possibly another collection $\mathcal{R}'$ of random seeds and an appropriate distribution $R'_\mathcal{M}$ such that for each $b \in \{0, 1\}$, the corresponding mechanism has a transition function of the form $\mathcal{M}_b^{\mathsf{npp}} : \mathcal{R} \times \mathcal{R}' \times (\mathcal{X}^2)^* \to \mathcal{V} \times \mathcal{Y}^2$, where the random seed distribution on $\mathcal{R}$ still follows $R_\mathcal{M}$, and the random seed from $\mathcal{R}'$ is sampled from $R'_\mathcal{M}$.

   In other words, in addition to a random seed in $\mathcal{R}$, an extra random seed in $\mathcal{R}'$ may be taken. Just like paired simulation in Definition B.2, a sequence of input pairs are also taken. However, the transition function returns a tuple in $\mathcal{V} \times \mathcal{Y}^2$, i.e., in addition to $v \in \mathcal{V}$, a pair $(y_0, y_1) \in \mathcal{Y}^2$ is returned.

2. For each $b \in \{0, 1\}$, $r \in \mathcal{R}$, $r' \in \mathcal{R}'$, $x^{(0)}[1..t]$ and $x^{(1)}[1..t] \in \mathcal{X}^*$,

   if $x^{(0)}[1..t]$ and $x^{(1)}[1..t]$ are not neighboring sequences, then $\mathcal{M}_b^{\mathsf{npp}}(r, r'; x^{(0)}[1..t], x^{(1)}[1..t]) = \perp$, and the whole simulation terminates at time step $t$.

   Otherwise, the pair $(y^{(0)}[1..t], y^{(1)}[1..t])$ of output sequences produced in the first $t$ time steps are neighboring in $\mathcal{Y}^*$.

3. If for some $(v, y_0, y_1) \in \mathcal{V} \times \mathcal{Y}^2$, $\mathcal{M}_b^{\mathsf{npp}}(r, r'; x^{(0)}[1..t], x^{(1)}[1..t]) = (v, y_0, y_1)$, then $\mathcal{M}(r; x^{(b)}[1..t]) = (v, y_b)$.

   In other words, we require consistency between $\mathcal{M}_b^{\mathsf{npp}}$ and $\mathcal{M}_b^{\mathsf{pair}}$ (from Definition B.2). Specifically, the projection of $\mathcal{M}_b^{\mathsf{npp}}$'s output onto $\mathcal{M}_b^{\mathsf{pair}}$'s range must match the behavior of $\mathcal{M}_b^{\mathsf{pair}}$.

However, unlike paired simulation $\mathcal{M}_b^{\text{pair}}$ whose behavior is totally determined by the given $\mathcal{M}$, in the tuple $(v, y_0, y_1)$ returned above by $\mathcal{M}_b^{\text{npp}}$, the value $y_{\bar{b}}$ (where $b \neq \bar{b}$) can depend on all the input parameters in a manner that is not specified by $\mathcal{M}$.

4. **Augmented Adversary View.** When an adversary interacts with $\mathcal{M}_b^{\text{npp}}$, the adversarial view consists of its randomness and the sequence in $(\mathcal{V} \times \mathcal{Y}^2)^*$ produced until termination.

**Definition B.9** (NPDP Interactive Mechanisms). Given a power function $f$, an interactive mechanism $\mathcal{M}$ is said to be adaptively $f$-neighbor-preserving differentially private (NPDP), if there exists a neighbor-preserving paired simulation $(\mathcal{M}_0^{\text{npp}}, \mathcal{M}_1^{\text{npp}})$ such that $\text{Pow}(\mathcal{M}_0^{\text{npp}} \| \mathcal{M}_1^{\text{npp}}) \leq f$, in the sense of Definition A.13.

**Remark B.10** (Construction of Neighbor-Preserving Paired Simulation). Observe that if one wishes to prove that a mechanism is NPDP as in Definition B.9, there is potentially some freedom to construct an appropriate NPP simulation.

On the contrary, for non-interactive mechanisms (that only run for one time step), the recent result on universally closest distribution refinements (Chan & Xue, 2025) implies that a "best possible" NPP simulation may be used.

Unfortunately, this does not seem to generalize to interactive mechanisms. The high level reason is that there may be no closest distribution refinements on two random sequences of length $t+1$ such that its prefix projection would give a closest distribution refinement for the corresponding prefixes of length $t$. In other words, if one constructs the "best possible" NPP up to time step $t$, it may not result in the best NPP if the process enters step $t+1$.

## C. Modular Composition Theorem for Adaptively Differentially Private Interactive Mechanisms

We next describe a modular composition theorem for interactive mechanisms. At a high level, the theorem states that if the first stage is *neighbor-preserving* (NPDP) with privacy profile $f$, and the second stage is (adaptive) DP with profile $g$ on the intermediate stream, then the end-to-end interactive pipeline is (adaptive) DP with profile $f \otimes g$.

**How this theorem relates to existing composition frameworks.** Our theorem can be viewed as a direct consequence of two existing ideas, once NPDP is interpreted through *paired simulation*.

First, it is shown (Zhou et al., 2024) that neighbor-preserving notions (including differential obliviousness/NPDO) admit an equivalent characterization in terms of a paired simulation that outputs a *refinement pair*. Informally, in the standard secret-bit distinguishing game, an adversary provides at each time step a neighboring pair $(x_t^{(0)}, x_t^{(1)})$, and a secret bit $b \in \{0, 1\}$ selects which component is the "real" input. A mechanism is DP if the adversary's views under $b = 0$ and $b = 1$ are close. The refinement-pair view resolves the usual obstacle in modular composition: even if the real world uses $x^{(b)}$ and produces a real intermediate output $y^{(b)}$, one also needs a synthetic neighbor $\bar{y}^{(1-b)}$ so that the next stage can be analyzed in a paired simulation. The NPDP guarantee precisely asserts the existence of such a paired simulator that, for each $b$, produces a pair $(y^{(0)}, y^{(1)})$ whose $b$-marginal matches the real execution, and such that the two resulting pair-distributions (for $b = 0$ vs. $b = 1$) are close.

Second, a general framework for adaptive privacy of continual mechanisms (Henzinger et al., 2026) is developed to prove concurrent composition theorems under interactive adversaries. In particular, once a mechanism is expressed in the standard secret-bit game (and the controller only interacts with downstream submechanisms through their privatized outputs), privacy of the overall system follows from standard interactive composition and post-processing. Under the paired-simulation perspective, our two-stage pipeline $\mathcal{M} \to \mathcal{N}$ fits this template: the NPDP stage can be viewed as a mechanism that, on secret bit $b$, outputs a paired intermediate stream $(y_t^{(0)}, y_t^{(1)})$ (together with visible outputs $v_t$), and the DP stage $\mathcal{N}$ is then run on $y^{(b)}$. Thus, one may reduce modular composition to ordinary interactive composition: compose the lifted NPDP mechanism (as DP-on-the-bit with profile $f$) with the second-stage DP mechanism (profile $g$), and finally apply post-processing to remove the extra synthetic transcript.

While this reduction gives a useful conceptual guide and could be formalized by instantiating the conditions of (Henzinger et al., 2026) together with the refinement-pair characterization of (Zhou et al., 2024), we include below a self-contained proof tailored to our setting and our power-function accounting. In particular, a more general concurrent setting (multiple submechanisms with interleaving calls) is studied in (Henzinger et al., 2026), whereas we only need the serial modular composition used by our buffering–aggregation pipeline. Providing a direct proof also makes explicit how the privacy profile tensor $f \otimes g$ arises in our notation, and it cleanly interfaces with the certification arguments in other sections.

**Theorem C.1.** *Suppose an interactive mechanism with transition function $\mathcal{M} : \mathcal{R}_1 \times \mathcal{X}^* \to \mathcal{V} \times \mathcal{Y}$ is adaptively $f$-NPDP (as in Definition B.9), and another one with $\mathcal{N} : \mathcal{R}_2 \times \mathcal{Y}^* \to \mathcal{W} \times \mathcal{Z}$ is adaptively $g$-DP (as in Definition B.3). Then, the modular composition (as in Definition B.7) $\mathcal{M} \to \mathcal{N} : \mathcal{R}_1 \times \mathcal{R}_2 \times \mathcal{X}^* \to \mathcal{V} \times \mathcal{W} \times \mathcal{Z}$ is adaptively $f \otimes g$-DP.*

---

The augmented adversary $\widehat{\mathcal{A}}$ interacts with $\widehat{\mathcal{L}}_b$, for each $b \in \{0, 1\}$, as follows.

1. The mechanisms $\mathcal{A}$, $(\mathcal{M}_0^{\mathsf{npp}}, \mathcal{M}_1^{\mathsf{npp}})$ and $(\mathcal{N}_0^{\mathsf{pair}}, \mathcal{N}_1^{\mathsf{pair}})$, sample the random seeds $r_{\mathcal{A}}$, $(r_{\mathcal{M}}, r'_{\mathcal{M}})$, $r_{\mathcal{N}}$ from the appropriate distributions, respectively.

2. At time step $t = 0$, call the subroutines $(v_0, y_0^{(0)}, y_0^{(1)}) \leftarrow \mathcal{M}_b^{\mathsf{npp}}(r_{\mathcal{M}}, r'_{\mathcal{M}})$ and $(w_0, z_0) \leftarrow \mathcal{N}_b^{\mathsf{pair}}(r_{\mathcal{N}})$.
   $\widehat{\mathcal{A}}$ observes $(v_0, w_0)$.

3. At time step $t \geq 1$,

   (a) $\widehat{\mathcal{A}}$ generates $(x_t^{(0)}, x_t^{(1)}) \leftarrow \mathcal{A}(r_{\mathcal{A}}; v[0..t-1], w[0..t-1])$, and sends it to $\widehat{\mathcal{L}}_b$.
   (b) $(v_t, y_t^{(0)}, y_t^{(1)}) \leftarrow \mathcal{M}_b^{\mathsf{npp}}(r_{\mathcal{M}}, r'_{\mathcal{M}}; x^{(0)}[1..t], x^{(1)}[1..t])$ and pass the hidden output pair $(y_t^{(0)}, y_t^{(1)})$ to the next mechanism $\mathcal{N}_b^{\mathsf{pair}}$;
   (c) $(w_t, z_t) \leftarrow \mathcal{N}_b^{\mathsf{pair}}(r_{\mathcal{N}}; y^{(0)}[1..t], y^{(1)}[1..t])$;
   (d) $\widehat{\mathcal{A}}$ observes $(v_t, w_t)$.

The whole process terminates as soon as one of $\mathcal{A}$, $\mathcal{M}_b^{\mathsf{npp}}$ and $\mathcal{N}_b^{\mathsf{pair}}$ terminates. Otherwise, the process continues for the next time step.

If the process terminates at time step $t$, the view of the adversary consists of its random seed $r_{\mathcal{A}}$, $v[0..t]$ and $w[0..t]$.

---

*Proof.* We denote the resulting interactive mechanism from the modular composition by $\mathcal{L} := \mathcal{M} \to \mathcal{N}$. In view of Theorem B.3, we consider the paired simulation $(\mathcal{L}_0^{\mathsf{pair}}, \mathcal{L}_1^{\mathsf{pair}})$ as described in Theorem B.2. Specifically, our goal is to show that for any adversary $\mathcal{A}$, the two distributions $\mathsf{view}(\mathcal{A} \leftrightarrow \mathcal{L}_0^{\mathsf{pair}})$ and $\mathsf{view}(\mathcal{A} \leftrightarrow \mathcal{L}_1^{\mathsf{pair}})$ are "close" in the sense quantified by the power function $f \otimes g$.

Our proof strategy is to construct an augmented pair $(\widehat{\mathcal{L}}_0, \widehat{\mathcal{L}}_1)$ of interactive mechanisms and an augmented adversary $\widehat{\mathcal{A}}$ whose views have the same distributions as those from $(\mathcal{L}_0^{\mathsf{pair}}, \mathcal{L}_1^{\mathsf{pair}})$ and $\mathcal{A}$.

We next use Fact A.14 to show that instead of considering the pair $(\mathcal{L}_0^{\mathsf{pair}}, \mathcal{L}_1^{\mathsf{pair}})$ with the same random seed distribution, we can equivalently simulate the pair with some common transition function, but using two different random seed distributions. By the data processing inequality, it suffices to analyze the alternative simulation and the power function of the two random seed distributions. Because $\mathcal{M}$ is $f$-NPDP, let $(\mathcal{M}_0^{\mathsf{npp}}, \mathcal{M}_1^{\mathsf{npp}})$ be the corresponding NPP simulation as guaranteed by Theorem B.9 such that $\mathsf{Pow}(\mathcal{M}_0^{\mathsf{npp}} \| \mathcal{M}_1^{\mathsf{npp}}) \leq f$. By Fact A.14, there exist some interactive mechanism $\mathcal{P}$ and two random seed distributions $F_0$ and $F_1$ such that $\mathsf{Pow}(F_0 \| F_1) \leq f$ and for each $b \in \{0, 1\}$, $\mathcal{P}(F_b)$ is an exact simulation of $\mathcal{M}_b^{\mathsf{npp}}$ (with the appropriate random seed distribution $R_{\mathcal{M}} \times R'_{\mathcal{M}}$) for any adversary. Specifically, for each $b$, $\mathcal{M}_b^{\mathsf{npp}}(R_{\mathcal{M}} \times R'_{\mathcal{M}})$ and $\mathcal{P}(F_b)$ are equivalent.

Similarly, since $\mathcal{N}$ is $g$-DP, the paired simulation $(\mathcal{N}_0^{\mathsf{pair}}, \mathcal{N}_1^{\mathsf{pair}})$ satisfies $\mathsf{Pow}(\mathcal{N}_0^{\mathsf{pair}} \| \mathcal{N}_1^{\mathsf{pair}}) \leq g$. Again, by Fact A.14, there exist some interactive mechanism $\mathcal{Q}$ and two random seed distributions $G_0$ and $G_1$ such that $\mathsf{Pow}(G_0 \| G_1) \leq g$ and for each $b \in \{0, 1\}$, $\mathcal{Q}(G_b)$ is an exact simulation of $\mathcal{N}_b^{\mathsf{pair}}$ (with the appropriate random seed distribution $R_{\mathcal{N}}$) for any adversary. Specifically, for each $b$, $\mathcal{N}_b^{\mathsf{pair}}(R_{\mathcal{N}})$ and $\mathcal{Q}(G_b)$ are equivalent.

Given an adversary $\mathcal{A}$ that interacts with $\mathcal{L}_b^{\mathsf{pair}}$ for $b \in \{0, 1\}$, we describe how an augmented adversary $\widehat{\mathcal{A}}$ interacts with $\widehat{\mathcal{L}}_b$ in the above colored box. Because we follow exactly the same structure as in Definition B.7, after comparing the two descriptions line-by-line, it follows that $\mathsf{view}(\mathcal{A} \leftrightarrow \mathcal{L}_b^{\mathsf{pair}})$ has the same distribution as $\mathsf{view}(\widehat{\mathcal{A}} \leftrightarrow \widehat{\mathcal{L}}_b)$.

As aforementioned, we use Fact A.14, for each $b \in \{0, 1\}$, we can equivalently replace $\mathcal{M}_b^{\mathsf{npp}}(R_{\mathcal{M}}, R'_{\mathcal{M}})$ with $\mathcal{P}(F_b)$, and $\mathcal{N}_b^{\mathsf{pair}}(R_{\mathcal{N}})$ with $\mathcal{Q}(G_b)$ in the above description.

Hence, the first inequality below follows from the data processing inequality:

$$\mathsf{Pow}\big(\mathsf{view}(\mathcal{A} \leftrightarrow \mathcal{L}_0^{\mathsf{pair}}) \,\|\, \mathsf{view}(\mathcal{A} \leftrightarrow \mathcal{L}_1^{\mathsf{pair}})\big) = \mathsf{Pow}\big(\mathsf{view}(\widehat{\mathcal{A}} \leftrightarrow \widehat{\mathcal{L}}_0) \,\|\, \mathsf{view}(\widehat{\mathcal{A}} \leftrightarrow \widehat{\mathcal{L}}_1)\big)$$
$$\leq \mathsf{Pow}(F_0 \times G_0 \,\|\, F_1 \times G_1)$$
$$\leq f \otimes g,$$

where the second inequality follows from the data processing inequality and the exact simulations, and the last inequality follows from Fact A.10. □

## D. RandBin: Achieving Privacy for Edit-Style Neighboring Inputs

In the literature, neighboring streams are typically defined in the Hamming sense. Specifically, given a collection $\mathcal{O}$ of objects, an (infinite) input stream is $x : \mathbb{Z}_{\geq 1} \to \mathcal{O}$. In our applications, each object in $\mathcal{O}$ captures the characteristics of a user.

*Hamming-Style Neighboring* Two streams $x \sim_H x'$ are neighboring in the Hamming sense if there exists some $t \in \mathbb{Z}$ such that $\sigma(\tau) = \sigma'(\tau)$ for all $\tau \neq t$.

*Edit-Style Neighboring.* We are interested in the neighboring notion when an element may be inserted or deleted from a stream.

**Definition D.1** (Neighboring in the Edit Sense). Given a stream $x$ and $t \in \mathbb{Z}$, we use $\mathsf{Delete}(x; t)$ to denote the stream $x'$ such that for all $\tau < t$, $x'[\tau] = x[\tau]$ and for all $\tau \geq t$, $x'[\tau] = x[\tau + 1]$; in other words, the element at time $t$ is deleted in stream $x$ to produce $x'$.

Two streams $x \sim_{\mathsf{E}} x'$ are neighboring in the edit sense if one stream can be obtained from the other by such a deletion operation.

Two finite sequences of the same length are neighboring (in the edit sense) if they are prefixes of such neighboring streams.

### D.1. Motivating Example: Private Prefix-Sum for Edit-Style Neighboring Streams

The input for the prefix-sum problem is a stream $\sigma : \mathbb{Z}_{>0} \to \mathbb{Z}$ of integers, and the (accurate) output is a vector $c$ of prefix-sums, i.e., for $t \geq 1$, $c[t] = \sum_{i=1}^{t} \sigma[i]$. The well-known private prefix-sum mechanism (Dwork et al., 2010a; Chan et al., 2011) based on the binary tree structure is designed for neighboring streams in the Hamming sense. At the step $t_0$ at which the two streams differ, the difference must be bounded, for instance $|\sigma(t_0) - \sigma'(t_0)| \leq 1$.

Hamming-style neighboring captures the scenario when a user wants some protection on its data contents. However, whether a user has participated in the process may be regarded as sensitive information. This is captured by a neighboring notion in the edit sense. However, deleting one element from a stream can change the element for every step. For instance, if we delete the first element from the stream $\sigma = (1, 0, 1, 0, \dots)$, the resulting stream $\sigma' = (0, 1, 0, 1, \dots)$ will have every position flipped! This is as far away from being Hamming-neighboring as one can get, and indeed the DP mechanism based on binary trees is not designed to provide good privacy guarantees for these two streams. Specifically, for all odd time steps $t$, the adversary can observe the noisy $\sigma[t] + N_t$, where $N_t$ is independently generated noise centered at 0. Therefore, by averaging over these numbers for odd time steps, the adversary can distinguish between $\sigma$ and $\sigma'$ almost for certain.

Instead of re-designing the DP prefix-sum mechanism to cater for edit-style neighboring streams, our goal is to design a pre-processing mechanism RandBin that somehow transforms edit-style neighboring streams into Hamming-style neighboring streams. Then, using the modular composition framework, we pass the processed stream to the tree-based DP prefix-sum mechanism.

**Random Buffer to the Rescue.** Edit-style neighboring notion has been considered in the context of oblivious data structures (Chan et al., 2022; Zhou et al., 2023), and the high-level idea is to use a *buffering* technique to introduce random delays into the streams in the hope of eventually aligning the elements from the two edit-style neighboring streams. We give a simplified view of the procedure that processes a stream $\sigma$ of elements. It maintains a buffer queue that is initially empty. At each step $t$, the element $\sigma[t]$ enters the end of the queue. The procedure then makes two random decisions: (i) whether to remove any element from the head of the queue; (ii) if yes, then remove a random number of elements from the head of the queue and return their sum in the output stream.

Consider applying this random procedure to two edit-style neighboring streams, where we use alphabets to label bits at different position:

$\sigma = [a, b, c, d, e, f] = [1, 0, 1, 0, 1, 0]$ and $\sigma' = [b, c, d, e, f, g] = [0, 1, 0, 1, 0, 1]$.

Below are possible outcomes of the random batching process for each stream, where we sum up the bits in each non-empty batch.

$$\sigma : \qquad [\bot, \bot, \{a, b\}, \bot, \bot, \{c, d, e\}] \rightarrow [\bot, \bot, 1, \bot, \bot, 2]$$
$$\sigma' : \qquad [\bot, \bot, \{b\}, \bot, \bot, \{c, d, e\}] \rightarrow [\bot, \bot, 0, \bot, \bot, 2]$$

Note that at the end of step 6, the queue contains $\{f\}$ for the case of $\sigma$ and contains $\{f, g\}$ for the case of $\sigma'$. In this particular lucky scenario, we see that the RandBin procedure has converted edit-style neighboring streams to Hamming-style neighboring streams. However, observe that these two outcomes from the two scenarios do not occur with exactly the same probability, but similar probabilities if the RandBin procedure is designed carefully. Moreover, for easy illustration, we give an oversimplified description of Hamming-style neighboring streams. We next give the formal and technical details.

### D.2. Technical Description of RandBin

We use the following Hamming-style neighboring notion on bin sequences, which are suitable for private mechanisms that are designed expecting Hamming-style neighboring streams.

**Definition D.2** (Neighboring Bin Sequences). We use $\mathcal{B}$ to denote the collection of *concrete* bins, where a concrete bin contains an array of objects with some specified maximum array length. In our usage, a concrete bin is non-empty and is typically *opaque* in the sense that the adversary cannot observe its contents.

A bin sequence of length $T$ is denoted by $B : [1..T] \rightarrow \mathcal{B} \cup \{\bot\}$, where $\bot$ means that no concrete bin is returned in that time step. Adjacent concrete bins with indices $i < j$ in a sequence $B$ mean that for $i < t < j$, $B[t] = \bot$.

Two bin sequences $B \sim_\mathsf{B} B'$ of the same length $T$ are neighboring if the following holds.

1. For all $t \in [1..T]$, $B[t] = \bot$ *iff* $B'[t] = \bot$.

2. Except for at most 2 indices $t$ in $[1..T]$, $B[t] = B'[t]$.

   (a) If $B$ and $B'$ differ in exactly one index, the corresponding index refers to the last concrete bin in both sequences. Moreover, the two bin contents must have the following form.
   There exist elements $w$ and $u$ and (possibly empty) element arrays $\sigma_1$ and $\sigma_2$ such that the two differing bins from $B$ and $B'$ contain the arrays:
   $[\sigma_1, w, \sigma_2]$ and $[\sigma_1, \sigma_2, u]$.

   (b) If $B$ and $B'$ differ in exactly two indices $i < j$, then the two indices correspond to adjacent concrete bins in both sequences. Moreover, there exist elements $w$ and $u$ and element arrays $\sigma_1$, $\sigma_2$ and $\sigma_3$ such that two adjacent concrete bins from each bin sequence are illustrated as follows:
   - $[\sigma_1, w, \sigma_2], \bot, \ldots, \bot, [u, \sigma_3]$
   - $[\sigma_1, \sigma_2, u], \bot, \ldots, \bot, [\sigma_3]$

Observe that if each of two neighboring bin sequences are concatenated, two resulting item sequences will have lengths differ by at most 1, in which case deleting one element from the longer item sequence results in the shorter item sequence.

We need the following primitives to describe the RandBin mechanism.

**Definition D.3** (Symmetric Geometric Distribution). Let $\alpha > 1$. The symmetric geometric distribution $\mathrm{Geom}(\alpha)$ takes integer values such that the probability mass function at $k$ is $\frac{\alpha - 1}{\alpha + 1} \cdot \alpha^{-|k|}$.

It is known that using $\mathrm{Geom}(e^\varepsilon)$ as additive noise can mask two integers differing by 1 with $(\varepsilon, 0)$-DP. However, if we want the noise to have bounded support, then we truncate the noise and get only $(\varepsilon, \delta)$-DP.

**Definition D.4** (Truncated Geometric Distribution). Let $\varepsilon > 0$, $\delta \in (0,1)$. Let $U$ be the smallest even positive integer such that $\mathbf{Pr}\left[|\mathrm{Geom}(e^\varepsilon)| \geq \frac{U}{2}\right] \leq \delta$, where $U = O(\frac{1}{\varepsilon} \log \frac{1}{\delta})$. The truncated geometric distribution $G(\varepsilon, \delta)$ has support $[-\frac{U}{2}, \frac{U}{2}]$, and is obtained by first sampling $r$ from $\mathrm{Geom}(e^\varepsilon)$ and then truncating $r$ within $[-\frac{U}{2}, \frac{U}{2}]$, i.e., return $\min(\max(-\frac{U}{2}, r), \frac{U}{2})$.

**Differentially Private Streaming Prefix Sum with Always Error Bounds.** We use the $(\varepsilon, \delta)$-DP interactive mechanism from Theorem F.1, denoted as $\mathsf{StreamSum}_{\varepsilon,\delta}$. Observe that $\mathsf{StreamSum}_{\varepsilon,\delta}$ is not necessarily private against adaptive adversaries. Even though we will prove that RandBin is adaptively NPDP, we do not need the adaptive property from $\mathsf{StreamSum}_{\varepsilon,\delta}$, but rather that the additive error from all time steps $t$ be bounded by some function $E(t)$ with probability 1.

**Mechanism Intuition.** In Algorithm 3, we reinterpret RandBin (which was introduced in (Chan et al., 2022; Zhou et al., 2023)) as an interactive mechanism. On a high level, the mechanism maintains an internal buffer that is a subarray. In each time step $t$, the mechanism takes an element $x[t]$ from the input stream and appends it to the end of the buffer. The mechanism decides according to some rules if a new bin should be created from a prefix of the buffer contents. If yes, a prefix from the buffer is removed to form a new bin that is returned in this step; otherwise, nothing is returned in this step and we denote the output by $\perp$.

---

**Algorithm 3** Interactive RandBin Mechanism

1: **Input:** $\varepsilon > 0$, $\delta \in (0, 1)$; a stream $x : \mathbb{Z}_{\geq 1} \to \mathcal{O}$ of items
2: **Output:** A bin sequence $\mathbb{Z}_{\geq 1} \to \mathcal{B} \cup \{\perp\}$; an adversary can only observe whether a concrete bin or $\perp$ is output in each step
3: Denote $\varepsilon_1 = \varepsilon_2 = \frac{\varepsilon}{2}$ and $\delta_1 = \delta_2 = \frac{\delta}{2}$
4: Let $G(\varepsilon_1, \delta_1)$ be the truncated geometric distribution, and let $U := O\left(\frac{1}{\varepsilon_1} \log\left(\frac{1}{\delta_1}\right)\right)$ be the value defined in Definition Theorem D.4
5: Denote the distribution $\mathcal{G} := \frac{3U}{2} + G(\varepsilon_1, \delta_1)$, whose mean is $\frac{3U}{2}$ and whose support is $[U..2U]$
6: Initialize an instance of the **interactive mechanism** $\mathsf{StreamSum}_{\varepsilon_2,\delta_2}$, as in Theorem F.1, that takes elements in $[U..2U]$
7: Moreover, with probability 1, the additive error for the length-$i$ prefix sum is **strictly** less than

$$E[i] := O\left(\frac{1}{\varepsilon_2} \cdot \log i \cdot \left(\log i + \log \frac{1}{\delta_2}\right)\right)$$

8: Initialize an empty FIFO queue $\mathsf{Buf} \leftarrow \emptyset$
9: Set $\mathsf{Index} \leftarrow 0$, $C[0] \leftarrow 0$, and $S[0] \leftarrow 0$
10: **for** each time step $t \geq 1$ at which element $x[t]$ is received **do**
11:     Append $x[t]$ to the end of the queue $\mathsf{Buf}$
12:     **if** $t < S[\mathsf{Index}] + E[\mathsf{Index}] + 2U$ **then**
13:         Output $\perp$ in step $t$
14:     **else**
15:         Remove the first $C[\mathsf{Index}]$ elements from the queue $\mathsf{Buf}$ to form $\mathsf{Bin}[\mathsf{Index}]$
16:                                                   $\triangleright$ $\mathsf{Bin}[\mathsf{Index}] \neq \perp$ *is a concrete bin* $\Leftrightarrow C[\mathsf{Index}] > 0 \Leftrightarrow \mathsf{Index} > 0$
17:         Output $\mathsf{Bin}[\mathsf{Index}]$ in step $t$
18:         Update $\mathsf{Index} \leftarrow \mathsf{Index} + 1$
19:         Sample $C[\mathsf{Index}]$ independently from $\mathcal{G}$
20:         $S[\mathsf{Index}] \leftarrow \mathsf{StreamSum}_{\varepsilon_2,\delta_2}(C[\mathsf{Index}])$
21:     **end if**
22: **end for**

---

**Theorem D.5** (Utility Guarantee of RandBin). *At the end of every step $t$, with probability 1, the concatenation of the concrete bins that have been output so far is a prefix of the input stream with length at least $t - O(\frac{1}{\varepsilon} \cdot \log t \cdot (\log t + \log \frac{1}{\delta}))$.*

*Proof.* Observe that because the additive error of $\mathsf{StreamSum}_{\varepsilon,\delta}$ holds with probability 1, the $\mathsf{Buf}$ will never run into underflow in line 15. Actually, to ensure no underflow, it suffices to have $t < S[\mathsf{Index}] + E[\mathsf{Index}]$ in line 12; we shall see that the extra term $2U$ is crucial to argue that RandBin is private against adaptive adversaries.

Note that at the end of time step $t$, we have the invariant that $t < S[\text{Index}] + E[\text{Index}] + 2U$; this holds because $S[\text{Index} + 1] - S[\text{Index}] \geq U \geq 1$ and $E[\text{Index} + 1] \geq E[\text{Index}]$.

Moreover, at this moment, the number of elements in the queue Buf is

$$t - \sum_{j=1}^{\text{Index}-1} C[j] < (S[\text{Index}] + E[\text{Index}] + 2U) - (S[\text{Index} - 1] - E[\text{Index} - 1])$$

$$\leq 4U + E[\text{Index}] + E[\text{Index} - 1] = O(\frac{1}{\varepsilon} \cdot \log t \cdot (\log t + \log \frac{1}{\delta})),$$

using the crude bound $\text{Index} \leq t$. □

### D.3. Adaptive Privacy Analysis

**Privacy of** RandBin**.** Observe that in each step, the view of the adversary just consists of one bit of information: whether a concrete bin or $\perp$ is returned. Moreover, the distribution of the view is always the same, regardless of what the input stream is. However, as noted in (Zhou et al., 2023), when a mechanism has hidden outputs that will be passed to another mechanism, one needs to consider the joint distribution of the view and the output, which is why NPDO was defined, which we have extended to interactive mechanisms in Definition B.9.

**Construction of Neighbor-Preserving Paired (NPP) Simulation for** RandBin**.** The proofs in (Zhou et al., 2023) already contain an implicit description of how an NPP simulation for RandBin is constructed, but we will make it explicit here. Recall that the adversary will generate two input sequences $(x^{(0)}, x^{(1)})$ in an NPP simulation in Definition B.8. For $b \in \{0, 1\}$, the mechanism $\mathcal{M}_b^{\text{npp}}$ will do two things in each step $t$.

1. Simulate the behavior of the original mechanism $\mathcal{M}$ on the sequence $x^{(b)}[1..t]$ to produce the corresponding view $v[t]$ and output $y^{(b)}[t]$.

2. Generate another output $y^{(\bar{b})}[t]$ such that the output sequences $y^{(b)}[1..t]$ and $y^{(\bar{b})}[1..t]$ are neighboring.

   The counter-intuitive point is that $y^{(\bar{b})}[1..t]$ is not necessarily the simulation of the original $\mathcal{M}$ on the other input sequence $x^{(\bar{b})}$. In fact, it is possible that $y^{(\bar{b})}[1..t]$ may not even be in the support of $\mathcal{M}$ on $x^{(\bar{b})}$.

   In general, extra randomness other than that used in the original $\mathcal{M}$ may be needed to generate $y^{(\bar{b})}[t]$. However, for RandBin, no extra randomness is needed.

*Technical Assumption on Input Pair.* Note that in Definition B.8, the adversary needs to ensure that at every time step $t$, the pair $(x^{(0)}[1..t], x^{(1)}[1..t])$ of input sequences generated so far are neighboring (in the edit sense).

For instance, for $t = 4$, the pair $([0, 1, 0, 1], [1, 0, 1, 0])$ of sequences are neighboring in the edit sense, but the issue is that we do not know which stream has an element deleted. If next input pair at $t = 5$ is $(0, a)$, then we know that in the two streams $(x^{(0)}, x^{(1)})$, the first element from $x^{(0)}$ is deleted to produce $x^{(1)}$; on the other hand, if the next input pair at $t = 5$ is $(a, 1)$, then the opposite is true.

However, in the construction of $\text{RandBin}_b^{\text{npp}}$, we would need to know whether an element is deleted from $x^{(0)}$ or $x^{(1)}$. Observe that this issue will not arise if a stream contains only distinct elements. In practice, this is a valid assumption because an item is typically tagged with unique metadata.

Hence, in the proof, we assume that if the adversary first generates $x^{(0)}[t] \neq x^{(1)}[t]$ at step $t$, then when we receive a pair at the beginning of step $t + 1$, we will know from which stream an element is deleted. This will not be too late, because the additive error for the prefix-sum $S[\text{Index}]$ is at most $E[\text{Index}]$, we can ensure that the Buf queue will never be empty, i.e., an element arriving at step $t$ will never be returned in a concrete bin in the same step $t$.

**Definition D.6** (Description of $(\text{RandBin}_0^{\text{npp}}, \text{RandBin}_1^{\text{npp}})$)**.** We focus on the description of $\text{RandBin}_0^{\text{npp}}$ (because the behavior of $\text{RandBin}_1^{\text{npp}}$ is symmetric); recall that the adversary generates a pair of time sequences $(x^{(0)}, x^{(1)})$.

The first component simulates RandBin from Algorithm 3 on the sequence $x^{(0)}$; in every step $t$, it will output $y^{(0)}[t]$ that is either a concrete bin or $\perp$.

We describe the second component more carefully, which is how $y^{(1)}[t]$ is constructed in each step $t$. Note that since the two output sequences must be neighboring as in Definition D.2, we will make sure $y^{(0)}[t] = \bot$ iff $y^{(1)}[t] = \bot$. Moreover, we need to ensure that at most two adjacent concrete bins are different in a specific way. We consider two cases.

1. Suppose an element $x^{(0)}[t] = w$ is deleted from $x^{(0)}$ to produce the stream $x^{(1)}$. In the output sequence $y^{(0)}$, we focus on two adjacent concrete bins that contains the deleted item $w$ and the next concrete bin:

   $[\sigma_1, w, \sigma_2], \bot, \ldots, \bot, [u, \sigma_3],$

   where $\sigma_1, \sigma_2$ and $\sigma_3$ are (potentially empty) subarrays.

   Then, in the output sequence $y^{(1)}$, those two concrete bins are modified to:

   $[\sigma_1, \sigma_2, u], \bot, \ldots, \bot, [\sigma_3].$

   Note that the bin $y^{(0)}[\tau] = [\sigma_1, w, \sigma_2]$ is returned in the output sequence $y^{(0)}$ in some step $\tau > t$; observe that at the end of step $\tau$, in the remaining Buf queue, $u$ should be the first element. Moreover, $u$ should have already appeared in the input pair together with the last element of $\sigma$. Hence, $\mathsf{RandBin}_0^{\mathsf{npp}}$ is able to produce $y^{(1)}[\tau] = [\sigma_1, \sigma_2, u]$.

   Moreover, note that if $[u, \sigma_3]$ contains exactly $U$ elements, then $[\sigma_3]$ will not be in the support of simulating the actual RandBin.

2. Suppose an element $x^{(1)}[t] = w$ is deleted from $x^{(1)}$ to produce the stream $x^{(0)}$. Again, we focus on two adjacent concrete bins from the output sequence $y^{(0)}$:

   $[\sigma_1, \sigma_2, u], \bot, \ldots, \bot, [\sigma_3],$

   where $[\sigma_1, \sigma_2, u]$ is the bin that contains the element $x^{(0)}[t]$, which is also the first element in the subarray $[\sigma_2, u]$. Note that in the case that $\sigma_2$ is empty, then $u = x^{(0)}[t]$.

   Then, in the output sequence $y^{(1)}$, those two concrete bins are modified to:

   $[\sigma_1, w, \sigma_2], \bot, \ldots, \bot, [u, \sigma_3]$

   Note that if $[\sigma_3]$ contains $2U$ elements, then $[u, \sigma_3]$ will not appear in the support of simulating the actual RandBin.

From construction, the two output bin sequences $y^{(0)}$ and $y^{(1)}$ are neighboring as in Definition D.2.

With the formal terminology in place, we can formally restate the result in (Zhou et al., 2023).

**Fact D.7** (RandBin is NPDP Against Non-Adaptive Adversary). *The interactive mechanism in Algorithm 3 is non-adaptively $(\varepsilon, \delta)$-NPDP. Specifically, in terms of power function, the NPP simulation $(\mathsf{RandBin}_0^{\mathsf{npp}}, \mathsf{RandBin}_1^{\mathsf{npp}})$ in Definition D.6 satisfies:* $\mathsf{Pow}^{\mathsf{NA}}(\mathsf{RandBin}_0^{\mathsf{npp}} \| \mathsf{RandBin}_1^{\mathsf{npp}}) \leq \mathsf{DP}_{\varepsilon, \delta}.$

**Lemma D.8.** *Suppose for some power function $f$, the interactive mechanism $\mathsf{RandBin}$ in Algorithm 3 is non-adaptively $f$-NPDP (with respect to input neighboring notion in Definition D.1 and output neighboring notion in Definition D.2). Then, $\mathsf{RandBin}$ is also adaptively $f$-NPDP; specifically, the NPP simulation $(\mathsf{RandBin}_0^{\mathsf{npp}}, \mathsf{RandBin}_1^{\mathsf{npp}})$ in Definition D.6 satisfies:*

$\mathsf{Pow}(\mathsf{RandBin}_0^{\mathsf{npp}} \| \mathsf{RandBin}_1^{\mathsf{npp}}) \leq f.$

*Proof.* By joint convexity in Fact A.9, we assume a deterministic adversary $\mathcal{A}$ that interacts with $(\mathsf{RandBin}_0^{\mathsf{npp}}, \mathsf{RandBin}_1^{\mathsf{npp}})$. Moreover, without loss of generality, to simplify the notation, we assume that there is a canonical sequence $x(t) := t$.

Then, for some initial steps $t$, the default action of the adversary is to generate a pair $(x_t^{(0)}, x_t^{(1)})$ satisfying $x_t^{(0)} = x_t^{(1)} = t$ until some critical step $t_c$ in which the adversary decides (based on its observation) to deviate from the default behavior. It can either:

(i) delete from $x^{(0)}$, i.e., for $t \geq t_c$, $x_t^{(0)} = t + 1$ and $x_t^{(1)} = t$, or

(ii) delete from $x^{(1)}$, i.e., for $t \geq t_c$, $x_t^{(0)} = t$ and $x_t^{(1)} = t + 1$.

Observe that before step $t_c$, the two input sequences $x^{(0)}$ and $x^{(1)}$ are identical. Hence, the single decision made by the adversary is at which time step $t_c$ it wants to deviate from the default behavior, and it does not really matter whether it deletes from $x^{(0)}$ or $x^{(1)}$.

*Augmented View.* We assume a more powerful adversary that can observe, in addition to its normal view (which consists of the contents of both bins $y^{(0)}[t]$ and $y^{(1)}[t]$), the noisy prefix-sums $S[\cdot]$ once they are created. We note that the proof in (Zhou et al., 2023) also allows the (non-adaptive) adversary to have this augmented view. However, it is important when the adversary learns about these noisy prefix-sums.

*Default Distribution.* Suppose the canonical sequence $x$ is used in both of $(x^{(0)}, x^{(1)})$. Then, the behaviors of both interactive mechanisms $(\mathsf{RandBin}_0^{\mathsf{npp}}, \mathsf{RandBin}_1^{\mathsf{npp}})$ are identical. We use $\Omega$ to denote the space of views, and $\mathcal{P}$ to denote the corresponding distribution of views. Each $\omega \in \Omega$ consists of the view and output of every time step $t \geq 1$.

*Decision of Adversary $\mathcal{A}$.* Because we have assumed that $\mathcal{A}$ is deterministic, for each $\omega \in \Omega$, either (i) $\mathcal{A}$ will never deviate from the default behavior, or (ii) $\mathcal{A}$ will deviate at the beginning of some step $t$, in which case the resulting view will be different from $\omega$ starting from step $t$ onwards. We use $\Omega_t \subseteq \Omega$ to denote the collection of view in which the adversary deviates at the beginning of step $t$.

Consider some $\omega \in \Omega_t$. As explained above, this means that the adversary decides to deviate at the beginning of step $t$, after observing the information in $\omega$ up to step $t - 1$. This means it will generate a pair $(x^{(0)}[t], x^{(1)}[t])$ of different inputs, and as a result the distributions of outputs for $(\mathsf{RandBin}_0^{\mathsf{npp}}, \mathsf{RandBin}_1^{\mathsf{npp}})$ from step $t$ onwards may be different. For such $\omega \in \Omega_t$, we use $g_\omega := \mathsf{Pow}(V_\omega^{(0)} \| V_\omega^{(1)})$ to denote the power function between the corresponding two view distributions $V_\omega^{(0)}$ and $V_\omega^{(1)}$ starting from step $t$.

**Key Insight.** We next show that for $\omega \in \Omega_t$, the decision of $\mathcal{A}$ to deviate the beginning of step $t$ is equivalent to a randomized non-adaptive adversary that interacts with $(\mathsf{RandBin}_0^{\mathsf{npp}}, \mathsf{RandBin}_1^{\mathsf{npp}})$. Suppose $i$ is the index of concrete bins such that $S[i-1] + E[i-1] + 2U < t \leq S[i] + E[i] + 2U$. First, analyze what information the adversary has gathered till the beginning of step $t$:

1. At the end of step $S[i-1] + E[i-1] + 2U$, the adversary learns the exact count $C[i-1]$ (and also $C[i-2], \ldots, C[1]$), because so far the two output sequences are identical. Moreover, it also learns the noisy prefix-sum $S[i]$ that is an estimate of $\sum_{j=1}^{i} C[j]$.

2. At the beginning of step $t$, it knows the exact number of elements in the Buf queue, which is larger than $2U$. Hence, the input pair of elements generated at time $t$ will definitely not go into the concrete bin with index $i$. This is why we need the term $2U$ in line 12.

   The **crucial observation** is that at this moment, the number $C[i+1]$ of elements in the $(i+1)$-st concrete bin has not been sampled yet.

3. Hence, if the adversary $\mathcal{A}$ decides to deviate at the beginning of step $t$, it will know that the element at step $t$ will be the $N_\omega := (t - \sum_{j=1}^{i} C[j])$-th element in the Buf queue.

   The adversary does not know $C[i]$, but based on the knowledge of $S[i], S[i-1], \ldots, S[1]$ and $C[i-1], \ldots, C[1]$, it knows the conditional distribution of $C[i]$, and hence, can derive the distribution of $N_\omega$.

*Conclusion.* For $\omega \in \Omega_t$, the adversary $\mathcal{A}$ deciding to deviate at the beginning of step $t$ will be equivalent to another instance where a randomized non-adaptive adversary decides upfront to deviate at some step according to the distribution $N_\omega$. From the hypothesis, we conclude that the power function satisfies $g_\omega = \mathsf{Pow}(V_\omega^{(0)} \| V_\omega^{(1)}) \leq f$.

Then, the whole proof concludes because:

$$\mathsf{Pow}(\mathsf{view}(\mathcal{A} \leftrightarrow \mathsf{RandBin}_0^{\mathsf{npp}}) \| \mathsf{view}(\mathcal{A} \leftrightarrow \mathsf{RandBin}_1^{\mathsf{npp}})) = \mathsf{Pow}((\mathcal{P}, V_\mathcal{P}^{(0)}) \| (\mathcal{P}, V_\mathcal{P}^{(1)}))$$
$$\leq \mathsf{Pow}(\mathcal{P} \| \mathcal{P}) \otimes f = f,$$

where the inequality follows from the composition rule of power functions in Fact A.10.

$\square$

Fact D.7 and Lemma D.8 immediately give the following corollary.

**Corollary D.9.** *The interactive mechanism* $\mathsf{RandBin}$ *in Algorithm 3 is adaptively* $(\varepsilon, \delta)$-*NPDP with input neighboring notion* $\sim_{\mathsf{E}}$ *and output neighboring notion* $\sim_{\mathsf{B}}$.

# E. Adaptive Differential Privacy for Independently Decomposable Mechanisms

We show that if an interactive mechanism has a special structure, then any non-adaptive privacy result can be automatically extended to adaptive adversaries.

**Definition E.1** (Independently Decomposable Mechanisms). An (abstract) interactive mechanism with transition function $\mathcal{M} : \mathcal{R} \times \mathcal{U}^* \to \mathcal{U}$ is independently decomposable if the following properties hold.

- Each random seed $r \in \mathcal{R}$ takes the form $r = (\omega_1, \omega_2, \ldots) = \Omega^{\mathbb{N}}$. Moreover, the mechanism samples $\omega_t$ independently from some distribution $R_t$ in time step $t$.
- For each $t \geq 1$, there exists a deterministic function $\mathcal{M}_t$ such that the output $y_t \leftarrow \mathcal{M}_t(\omega_t; x[1..t]) \in \mathcal{U}$ depends only on $x[1..t]$ and the randomness $\omega_t$ sampled in step $t$.

The main result is that for a power function $f$, an independently decomposable interactive mechanism $\mathcal{M}$ is adaptively $f$-DP *iff* it is non-adaptively $f$-DP. Clearly, the "only-if" direction is trivial, and we prove the "if" direction as follows.

**Theorem E.2.** *Consider Hamming-style neighboring input sequences in $\mathcal{U}$, i.e., two input sequences are neighboring if they differ in at most one position. For a power function $f$, suppose an independently decomposable interactive mechanism $\mathcal{M} : \mathcal{R} \times \mathcal{U}^* \to \mathcal{U}$ is non-adaptively $f$-DP. Then, the mechanism $\mathcal{M}$ is also adaptively $f$-DP.*

*Proof.* In view of the joint convexity property in Fact A.9, it suffices to consider an adaptive adversary that is deterministic. Moreover, we may assume that the adversary initiates the first message, because any initial message generated by the mechanism may be assumed to be deterministic (by Fact A.9 again) and assimilated by the adversary's initial decision. We prove the result by induction on the number of rounds $T$ run on the mechanism $\mathcal{M}$.

The base case $T = 1$ is trivial, because there is no distinction between adaptive and non-adaptive adversaries. Consider the case when the interactive mechanism that runs for $T > 1$ steps. To simplify the notation, we assume, without loss of generality, that the adaptive adversary can observe the entire output of $\mathcal{M}$ in $\mathcal{U}$ in each time step. Recall from Definition B.2 of paired simulation that we consider a deterministic adaptive adversary $\mathcal{A} : \mathcal{U}^* \to \mathcal{U}^2$ that interacts with the pair $(\mathcal{M}_0^{\mathsf{pair}}, \mathcal{M}_1^{\mathsf{pair}})$. Our goal is to show that if $\mathcal{M}$ is non-adaptively $f$-DP, then we have:

$$\mathsf{Pow}(\mathsf{view}(\mathcal{A} \leftrightarrow \mathcal{M}_0^{\mathsf{pair}}) \| \mathsf{view}(\mathcal{A} \leftrightarrow \mathcal{M}_1^{\mathsf{pair}})) \leq f.$$

Since we consider Hamming-style neighboring sequences, except in one time step, the adversary typically generates a pair $(x^{(0)}[t], x^{(1)}[t])$ such that $x^{(0)}[t] = x^{(1)}[t]$. However, in at most one time step $t^*$ known as the *challenge step*, the adversary may generate a pair such that $x^{(0)}[t] \neq x^{(1)}[t]$

Since $\mathcal{A}$ is deterministic, let $(x^{(0)}[1], x^{(1)}[1])$ be the pair of inputs picked by $\mathcal{A}$ in the first step. Suppose $R_1$ is the distribution of the randomness used by $\mathcal{M}$ in the first time step. For $b \in \{0, 1\}$, denote the distribution $U^{(b)} = \mathcal{M}(R_1; x^{(b)}[1])$ of output generated by $\mathcal{M}_b^{\mathsf{pair}}$ in the first time step. Let $f_1 := \mathsf{Pow}(U^{(0)} \| U^{(1)})$. There are two separate cases.

**Case 1:** Step 1 is the challenging step, i.e., $x^{(0)}[1] \neq x^{(1)}[1]$. From the case $T = 1$, since the mechanism is non-adaptively $f$-DP, we have $f_1 \leq f$. However, since the adversary must produce a pair of identical inputs in all subsequent time steps, the corresponding views of $\mathcal{M}_0^{\mathsf{pair}}$ and $\mathcal{M}_1^{\mathsf{pair}}$ for $t \geq 2$ are data post-processing of $U^{(0)}$ and $U^{(1)}$ by the same (randomized) process.

Hence, we have $\mathsf{Pow}(\mathsf{view}(\mathcal{A} \leftrightarrow \mathcal{M}_0^{\mathsf{pair}}) \| \mathsf{view}(\mathcal{A} \leftrightarrow \mathcal{M}_1^{\mathsf{pair}})) \leq f$.

**Case 2:** Step 1 is a typical step, i.e., $x^{(0)}[1] = x^{(1)}[1]$. Since $U^{(0)}$ and $U^{(1)}$ have the same distribution, their power function $f_1 = \mathsf{Id}$ is the identity function.

Suppose $\Lambda \subseteq (\mathcal{U}^2)^{T-1}$ denotes the collection of pairs from time step 2 to $T$ will form two (Hamming-style) neighboring sequences of length $T - 1$.

Because $\mathcal{M}$ is independently decomposable, given $\lambda$ (and also $(x^{(0)}[1], x^{(1)}[1])$), we can recover exactly the distribution of outputs from step 2 to step $T$ of $(\mathcal{M}_0^{\mathsf{pair}}, \mathcal{M}_1^{\mathsf{pair}})$ without knowing the output from step 1.

Writing $\lambda = (x^{(0)}[2..T], x^{(1)}[2..T])$, for $b \in \{0, 1\}$, let $V_\lambda^{(b)}$ denote the distribution of the outputs from step 2 to $T$ when $x^{(b)}[1..T]$ is sent to $\mathcal{M}$, where the randomness of $V_\lambda^{(b)}$ is derived from the independent randomness $R_2 \times \cdots \times R_T$ sampled from step 2 to $T$. We define the power function: $g_\lambda := \mathsf{Pow}(V_\lambda^{(0)} \| V_\lambda^{(1)})$.

Consider the non-adaptive adversary that picks $(x^{(0)}[1], x^{(1)}[1])$ in the first step and picks $\lambda \in \Lambda$ in step 2 to $T$. Because $\mathcal{M}$ is non-adaptively $f$-DP and independently decomposable,

$$f_1 \otimes g_\lambda = g_\lambda \leq f.$$

Because this inequality holds for all $\lambda \in \Lambda$, we have:

$$g := \sup_{\lambda \in \Lambda} g_\lambda \leq f.$$

Let $u \in \mathcal{U}$ denote some output from the support of $U^{(0)}$ and $U^{(1)}$. Conditioning on the output of the first step being $u$, we follow the deterministic strategy of the given $\mathcal{A}$, and use $\mathcal{A}_u$ to denote the corresponding (adaptive) adversary that runs from step 2 to step $T$, which can be viewed as interacting with a paired simulation $(\mathcal{M}'_0, \mathcal{M}'_1)$ for $T-1$ steps. Note that $(\mathcal{M}'_0, \mathcal{M}'_1)$ can be derived from the given $\mathcal{M}$ and the pair $(x^{(0)}[1], x^{(1)}[1])$ from the first step.

For $b \in \{0, 1\}$, denote the distribution:

$W^{(b)}(u) := \mathsf{view}(\mathcal{A}_u \leftrightarrow \mathcal{M}'_b)$. Define $h_u := \mathsf{Pow}(W^{(0)}(u) \| W^{(1)}(u))$.

By the definition of $g$, the interactive mechanism $\mathcal{M}'$ is non-adaptively $g$-DP. Hence, by the induction hypothesis, we have:

$h_u \leq g$, for all $u$ in the support of $U^{(0)}$ and $U^{(1)}$.

Finally, observe that for $b \in \{0, 1\}$, $\mathsf{view}(\mathcal{A} \leftarrow \mathcal{M}_b^{\mathsf{pair}})$ has exactly the same distribution as $(U^{(b)}, W^{(b)}(U^{(b)}))$. By the composition rule of power functions in Fact A.10, we have:

$\mathsf{Pow}(\mathsf{view}(\mathcal{A} \leftarrow \mathcal{M}_0^{\mathsf{pair}}) \| \mathsf{view}(\mathcal{A} \leftarrow \mathcal{M}_1^{\mathsf{pair}})) \leq f_1 \otimes (\sup_u h_u) = \sup_u h_u \leq g \leq f$, as required to finish the inductive step.

$\square$

# F. Streaming Prefix-Sum Mechanism

There have been numerous works on differentially private prefix-sum algorithms since the *binary tree mechanism* (Chan et al., 2010; 2011) was proposed. We state the properties of the variant that are needed for our applications and briefly outline how they can be achieved from existing works.

**Problem Setting.** For each time step $t \geq 1$, an input stream has some integer $x_t \in \mathcal{X} = [a..b]$ from some known range that is passed to the interactive mechanism $\mathsf{StreamSum}$ that returns a value $S_t$ that is supposed to be an estimation of the sum $\sum_{\tau=1}^t x_\tau$. The *additive error* is $|S_t - \sum_{\tau=1}^t x_\tau|$. The adversary observes $S_t$.

**Neighboring Notion.** Two input streams are *neighboring* if they differ in at most one time step, and the corresponding two values differ by 1.

As aforementioned, a mechanism with unbounded termination satisfies $(\varepsilon, \delta)$-DP if for every $T > 0$, the truncated mechanism run for $T$ steps is $(\varepsilon, \delta)$-DP.

**Theorem F.1** (Stream Sum with Bounded Error and Consistency). *For any $\varepsilon > 0, \delta \in (0, 1)$ and an interval $[a..b] \subseteq \mathbb{Z}$ of integers, there exists an $(\varepsilon, \delta)$-differentially private interactive mechanism $\mathsf{StreamSum}_{\varepsilon, \delta}$, such that given a stream $\{x_t\}$ of integers in $[a..b]$, the algorithm outputs a number $S_t$ in each time step $t$ such that the following hold.*

- ***Bounded Error:*** *With probability 1, at each time step $t$, the output has additive error at most $E(t) := O\left(\frac{1}{\varepsilon} \cdot \log t \cdot (\log t + \log \frac{1}{\delta})\right)$.*

- ***Consistency:*** *With probability 1, for each time step $t$, the increment $S_t - S_{t-1} \in [a..b]$ is in the above range.*

As in (Chan et al., 2022), we start with a pure $\varepsilon$-DP prefix sum algorithm that has high probability guarantee on the additive error.

**Fact F.2** (Pure DP Stream Sum (Chan et al., 2011)). *For any $\varepsilon > 0$, there exists an $(\varepsilon, 0)$-differentially private interactive mechanism $\mathsf{HybridStreamSum}_\varepsilon$ that takes an input stream $\{x_t\}$ of integers and output $S_t$ at each time step $t$ with the following properties.*

- ***Bounded Error:*** *Fix any $0 < \delta < 1$. With probability $1 - \delta$, for every time step $t$, the output at time $t$ has an additive*

*error of at most $O(\frac{1}{\varepsilon} \log t(\log t + \log \frac{1}{\delta}))$.*

- **Structure:** *In each time step $t$, the mechanism actually allows the adversary to observe noisy counts of the form $\sum_{i \in I} x_i$ masked with independent noise, where the index interval has the form $I = [c..t]$. The output $S_t$ can be constructed deterministically from the noisy counts released so far.*

*Proof of Theorem F.1.* Fix $\varepsilon$ and $\delta$. Starting from the mechanism $\mathsf{HybridStreamSum}_\varepsilon$ in Fact F.2, we modify it step by step to achieve the desired properties.

1. *Always Error Bound.* If the output $S_t$ is too large, i.e., $S_t > \sum_{i=1}^t x_i + E(t)$, return $\sum_{i=1}^t x_i + E(t)$; if the output is too small, i.e., $S_t < \sum_{i=1}^t x_i - E(t)$, return $\sum_{i=1}^t x_i - E(t)$.

   The argument in Theorem 3.2 of (Chan et al., 2022) shows this can turn a high probability additive error bound into an additive error bound with probability 1, at the cost of getting $(\varepsilon, \delta)$-DP.

2. *Consistency.* The argument in Lemma 5.2 of (Chan et al., 2011) shows that a further truncation operation can achieve consistency without increasing the additive error bound.

□

**Corollary F.3.** *The interactive mechanism $\mathsf{HybridStreamSum}_\varepsilon$ in Fact F.2 is adaptively $(\varepsilon, 0)$-DP, and can be made to be consistent.*

*Proof.* The structure property in Fact F.2 implies that noisy counts obey the independent decomposability property in Definition E.1. Adaptability follows from Theorem E.2.

Consistency can be achieved, because the modification can be done deterministically from the released noisy counts only.

□

**Remark F.4.** It is not clear whether the "Always Error Bound" can be achieved under the independent decomposability framework, because the truncation needs to refer back to the private inputs, in addition to the released noisy counts.

### F.1. Prefix Sums via Lower-Triangular Matrix Factorizations

We briefly restate the prefix-sum workload in the language of the *matrix mechanism*. Fix a horizon $T$ and write the stream as a vector $x \in \mathbb{R}^T$. The prefix sums are the linear workload

$$M_{\mathsf{count}} x \in \mathbb{R}^T, \qquad M_{\mathsf{count}}[t, i] = \mathbf{1}[t \geq i],$$

i.e., $(M_{\mathsf{count}} x)_t = \sum_{i=1}^t x_i$. We use the event-level neighboring notion: $x \sim x'$ iff $x - x' = \pm \Delta e_i$ for some coordinate $i \in [T]$, where $\Delta > 0$ is the sensitivity unit (in the above we take $\Delta = 1$).

**Privacy Notions: Renyi DP (RDP) and zCDP.** Let $P, Q$ be distributions with densities $p, q$ w.r.t. a common base measure. For $\alpha > 1$, the (order-$\alpha$) Renyi divergence is

$$D_\alpha(P \| Q) := \frac{1}{\alpha - 1} \log \int p(y)^\alpha q(y)^{1-\alpha} \, dy.$$

An interactive mechanism $\mathcal{M}$ is $(\alpha, \varepsilon)$-*RDP* if for all neighboring $x \sim x'$,

$$D_\alpha\big(\mathcal{M}(x) \,\|\, \mathcal{M}(x')\big) \leq \varepsilon.$$

It is $\rho$-*zCDP* if for all $\alpha > 1$ and all neighboring $x \sim x'$,

$$D_\alpha\big(\mathcal{M}(x) \,\|\, \mathcal{M}(x')\big) \leq \rho \alpha.$$

Equivalently, $\rho$-zCDP means $(\alpha, \rho\alpha)$-RDP holds simultaneously for every $\alpha > 1$.

**Converting RDP to $(\varepsilon, \delta)$-DP.** If $\mathcal{M}$ satisfies $(\alpha, \varepsilon_\alpha)$-RDP for some $\alpha > 1$, then for every $\delta \in (0, 1)$ it satisfies $(\varepsilon, \delta)$-DP with

$$\varepsilon = \varepsilon_\alpha + \frac{\log(1/\delta)}{\alpha - 1}.$$

One may optimize over $\alpha > 1$ to get the tightest $(\varepsilon, \delta)$ bound:

$$\varepsilon(\delta) = \inf_{\alpha > 1} \left\{ \varepsilon_\alpha + \frac{\log(1/\delta)}{\alpha - 1} \right\}.$$

**Lower-Triangular Factorization Mechanism.** Let $M_{\text{count}} = LR$ be a (possibly rectangular) factorization where $R \in \mathbb{R}^{m \times T}$ and $L \in \mathbb{R}^{T \times m}$ are *lower-triangular in time* (i.e., causal: the $t$-th output depends only on $x[1..t]$ through $R$ and $L$). Define the column-sensitivity of $R$ by

$$s(R) := \max_{i \in [T]} \|Re_i\|_2 = \|R\|_{1 \to 2}.$$

Consider the Gaussian matrix mechanism

$$\mathcal{M}_{L,R,\sigma}(x) := L(Rx + z), \qquad z \sim \mathcal{N}(0, \sigma^2 I_m) \text{ with independent coordinates.}$$

Since $L(\cdot)$ is post-processing, privacy is governed entirely by the Gaussian release of $Rx$.

**Lemma F.5** (Exact zCDP/RDP Constants for $L(Rx + z)$). *For neighboring $x \sim x'$ with $x - x' = \pm\Delta e_i$, we have $\|R(x - x')\|_2 \leq \Delta\, s(R)$. Hence the mechanism $\mathcal{M}_{L,R,\sigma}$ satisfies:*

- **zCDP:** $\rho$-zCDP with

$$\rho = \frac{\Delta^2\, s(R)^2}{2\sigma^2}.$$

- **RDP:** *for every order $\alpha > 1$, $(\alpha, \varepsilon_\alpha)$-RDP with*

$$\varepsilon_\alpha = \alpha\rho = \frac{\alpha\,\Delta^2\, s(R)^2}{2\sigma^2}.$$

**Binary Tree Mechanism as a Factorization.** Assume first $T = 2^h$ for an integer $h \geq 0$ (otherwise pad to the next power of two, which only changes constants by replacing $h$ with $\lceil \log_2 T \rceil$). Index the nodes of the complete binary tree over $[T]$ by dyadic intervals; let $q \in \mathbb{R}^m$ be the vector of all dyadic-interval sums, so that $q = R_{\text{tree}}x$ where $R_{\text{tree}}$ is a $0/1$ incidence matrix (leaf $i$ contributes to exactly its $h + 1$ ancestors). Each prefix $[1..t]$ has a canonical dyadic partition of size at most $h + 1$, so there is a reconstruction matrix $L_{\text{tree}}$ with $M_{\text{count}} = L_{\text{tree}} R_{\text{tree}}$.

**Proposition F.6** (Exact Privacy Constant for the Binary Tree Factorization). *For $T = 2^h$, every column of $R_{\text{tree}}$ has exactly $h + 1$ ones, hence*

$$s(R_{\text{tree}})^2 = \|R_{\text{tree}}\|_{1 \to 2}^2 = h + 1.$$

*Therefore, the Gaussian tree mechanism $L_{\text{tree}}(R_{\text{tree}}x + z)$ with $z \sim \mathcal{N}(0, \sigma^2 I_m)$ is $\rho_{\text{tree}}$-zCDP with*

$$\rho_{\text{tree}} = \frac{\Delta^2(h + 1)}{2\sigma^2},$$

*and for every $\alpha > 1$ it is $(\alpha, \varepsilon_{\alpha, \text{tree}})$-RDP with*

$$\varepsilon_{\alpha, \text{tree}} = \frac{\alpha\,\Delta^2(h + 1)}{2\sigma^2}.$$

*For general $T$, the same holds with $h := \lceil \log_2 T \rceil$.*

**A Smooth Lower-Triangular Factorization (Fichtenberger et al., 2023).** Fichtenberger et al. give an explicit *Toeplitz lower-triangular* factorization $M_{\mathsf{count}} = LR$ with $L = R$ defined by a scalar sequence $f(\cdot)$:

$$f(0) = 1, \qquad f(k) = \left(\frac{2k-1}{2k}\right)f(k-1) \ \ (k \geq 1), \qquad L[t,i] = R[t,i] = f(t-i) \ \ (t \geq i),$$

and $0$ otherwise. Let $\gamma$ be the Euler–Mascheroni constant and define

$$\Psi(T) := 1 - \frac{1-\gamma}{\pi} + \frac{\ln T}{\pi} + \frac{2}{T}.$$

**Proposition F.7** (Exact Privacy Constant for the Fichtenberger–Henzinger–Upadhyay Factorization). *Let $L, R$ be the above factorization. Then $M_{\mathsf{count}} = LR$ and*

$$s(R)^2 = \|R\|_{1\to 2}^2 = \|L\|_{2\to\infty}^2 \leq \Psi(T).$$

*Consequently, the mechanism $L(Rx + z)$ with $z \sim \mathcal{N}(0, \sigma^2 I_T)$ is $\rho_{\mathsf{FHU}}$-zCDP with*

$$\rho_{\mathsf{FHU}} = \frac{\Delta^2 \|R\|_{1\to 2}^2}{2\sigma^2} \leq \frac{\Delta^2 \Psi(T)}{2\sigma^2},$$

*and for every $\alpha > 1$ it is $(\alpha, \varepsilon_{\alpha,\mathsf{FHU}})$-RDP with*

$$\varepsilon_{\alpha,\mathsf{FHU}} = \frac{\alpha \Delta^2 \|R\|_{1\to 2}^2}{2\sigma^2} \leq \frac{\alpha \Delta^2 \Psi(T)}{2\sigma^2}.$$

**Takeaway.** Both mechanisms fit the same template $L(Rx + z)$, but yield different (exact) privacy constants through the column $\ell_2$ sensitivity $s(R)$: the binary tree gives $s(R)^2 = \Theta(\log_2 T)$, while the smooth Toeplitz factorization satisfies $s(R)^2 \leq 1 - \frac{1-\gamma}{\pi} + \frac{\ln T}{\pi} + \frac{2}{T}$.

# G. Additional Experiment Details

### G.1. Privacy Attack: Hamming-Style DP vs. Edit-Style DP in Continual Learning

**Objective.** We construct an attack scenario showing that a continual-learning mechanism based on prefix-sum aggregation can satisfy privacy under *Hamming-style adjacency* (record substitution), but fails under *edit-style adjacency* (single insertion/deletion with global shift). This demonstrates that Hamming-style DP guarantees do not automatically extend to edit-style (stream-level) privacy in continual-release settings.

**Neighboring streams (edit adjacency).** We construct two data streams $D$ and $D'$ over MNIST digits $\{0,1\}$ with fixed batch size and fixed batching. Importantly, each occurrence of "0" or "1" corresponds to a *distinct training sample* from the dataset (not repeated copies of a single example); the symbols only denote the class labels.

The streams are constructed as follows:

- **Stream $D$.** All samples have label $0$, except that the *first element of every odd-indexed batch* (batches $1, 3, 5, \ldots$) is replaced by a sample with label $1$.

- **Stream $D'$.** Obtained from $D$ by deleting the first element, shifting the entire stream left by one position, and appending a $0$-labeled sample at the end.

Thus $D$ and $D'$ differ by a *single edit operation* (delete+shift), and are neighbors under edit-style adjacency. However, the shift moves the injected "1" samples across batch boundaries, which produces *systematic differences in many minibatch compositions and gradients* across training rounds, despite the streams differing by only one edit.

**Learning mechanism.** We follow the framework of Algorithm 2 using a tree-based prefix-sum mechanism. Let $D_\star \in \{D, D'\}$ denote the *true private stream*. At round $t$, the $t$-th batch of $D_\star$ is fed into the current model $w^{(t-1)}$. Per-sample gradients are computed, clipped to norm $G$, and averaged to form the minibatch gradient $g_t^{(D_\star)} \in \mathbb{R}^d$. The update is made private using a continual prefix-sum mechanism: $S_t = \mathsf{PrivStreamSum}(g_1^{(D_\star)}, \ldots, g_t^{(D_\star)})$, and the model is updated by post-processing $w^{(t)} = w^{(0)} - \eta S_t$. All randomness is injected solely through the prefix-sum mechanism $\mathsf{PrivStreamSum}(\cdot)$; gradient computation, clipping, and averaging are deterministic.

---

**Algorithm 4** Streaming Majority-Vote Test

---

1: **Input:** Released private prefix sums $S_t = \mathsf{PrivStreamSum}(g_1^{(D_\star)}, \ldots, g_t^{(D_\star)})$; candidate streams $D, D'$
2: **Output:** Decision $\hat{D}_t$ at each round $t$
3: Initialize counters $c_D \leftarrow 0$ and $c_{D'} \leftarrow 0$
4: **for** $t = 1, \ldots, T$ **do**
5:     Observe noisy increment $\Delta_t \leftarrow S_t - S_{t-1}$
6:     Using model $w^{(t-1)}$, compute candidate gradients:
7:         Feed batch $t$ of $D$ to obtain $g_t^{(D)}$
8:         Feed batch $t$ of $D'$ to obtain $g_t^{(D')}$
9:     **if** $\|\Delta_t - g_t^{(D)}\| \leq \|\Delta_t - g_t^{(D')}\|$ **then**
10:       $c_D \leftarrow c_D + 1$
11:     **else**
12:       $c_{D'} \leftarrow c_{D'} + 1$
13:     **end if**
14:     $\hat{D}_t \leftarrow \arg\max\{c_D, c_{D'}\}$                ▷ *majority over history*
15: **end for**

---

**Adversarial test.** Assume the private stream is $D_\star \in \{D, D'\}$. At each even-indexed round $t$, the adversary observes the released model and reconstructs the noisy prefix-sum increment

$$\Delta_t := S_t - S_{t-1}.$$

Using the current model $w^{(t-1)}$, the adversary then:

1. feeds the *t-th batch of $D$* into the model to compute the clipped-and-averaged gradient $g_t^{(D)}$;

2. feeds the *t-th batch of $D'$* into the model to compute the clipped-and-averaged gradient $g_t^{(D')}$.

The adversary compares the observed noisy increment $\Delta_t$ with the two candidate gradients $g_t^{(D)}$ and $g_t^{(D')}$, and casts a vote for the closer one in norm. Votes are aggregated over time using a majority rule.

**Accuracy metric.** For each round $t$, the test outputs a majority decision $\hat{D}_t$ using all evidence up to round $t$. The per-round accuracy is defined as $\mathrm{Acc}(t) = \mathbf{Pr}\left[\hat{D}_t = D_{\mathrm{true}}\right]$, and is estimated by averaging over multiple independent runs. This produces an accuracy curve as a function of training round.

**Results.** Figure 5 shows the accuracy curves for different noise levels in the tree-aggregation mechanism:

• For small noise ($\sigma \in [0.1, 1]$), the accuracy rapidly converges to 1, meaning the adversary almost surely distinguishes the two streams.

• For larger noise ($\sigma \in [2, 4]$), the accuracy decreases but remains *significantly above random guessing* (0.5).

**Conclusion.** This experiment provides a concrete separation: a continual prefix-sum mechanism that is secure under Hamming-style adjacency can leak information under edit-style adjacency. The failure is not due to local noise miscalibration, but to *temporal misalignment*: a single edit induces correlated shifts across many rounds, which can be aggregated by a streaming adversary. This shows that Hamming-style DP does not imply edit-style DP for continual-release learning systems.

### G.2. Experiments Investigating Properties of RandBin

**Batch-size effects.** The mean batch-size is $\frac{3U}{2} = \Theta(\frac{1}{\varepsilon} \log \frac{1}{\delta})$. Larger $\varepsilon$ and $\delta$ lead to smaller batch sizes and more frequent updates; see Figure 6 for typical values of $(\varepsilon, \delta)$.

**Backlog validation.** We validate that the backlog of pending users remains small in practice. We add a normalized-backlog plot, namely $Q_t/t$, versus time for the same settings as the raw-backlog plot in Figure 7. This directly shows that while the

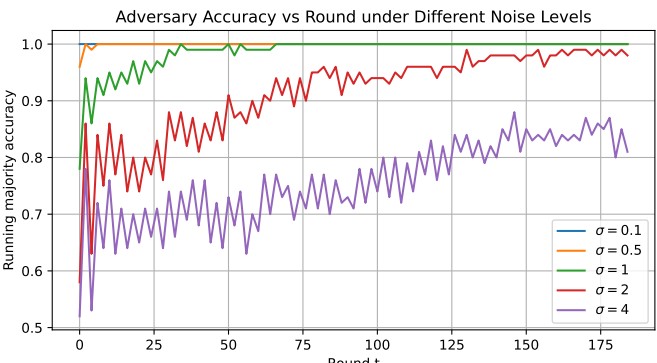

*Figure 5.* Adversary accuracy over rounds for different noise levels $\sigma$.

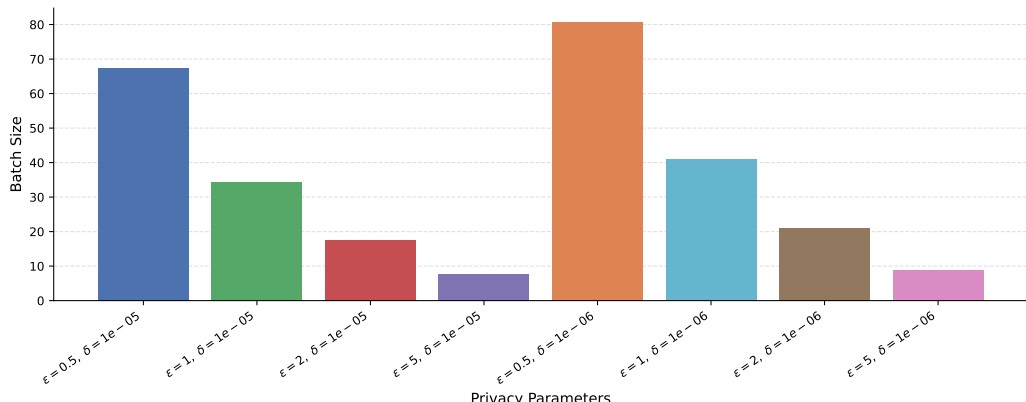

*Figure 6.* Mean batch sizes under different privacy parameters.

raw backlog can grow with the horizon, the fraction of pending arrivals becomes negligible over time. This is the practically relevant view in light of Lemma 4.2, because it shows that the fraction of delayed events becomes asymptotically negligible even when the raw backlog is horizon-dependent.

### G.3. Deep-Network Experiments for CIFAR10 and MNIST

We provide additional experimental results for CIFAR-10 and MNIST datasets in Figure 8, complementing the EMNIST results in Figure 3.

### G.4. Experiments for Application of Our Framework to ADMM

**ADMM synthetic experiment overview.** We include an ADMM experiment as a representative, more general optimizer that still fits our gradient-based update model. We instantiate Algorithm 2 with RandBin and PrivStreamSum under the single-edit stream definition, using the linearized ADMM formulation of Chan et al. (2024). Below we provide the background updates and the synthetic setup and results.

**ADMM Background and Private Updates** We follow the description from (Chan et al., 2024). Let $n$, $m$, $\ell$ be positive integers, $F : \mathbb{R}^n \to \mathbb{R}$ and $g : \mathbb{R}^\ell \to \mathbb{R}$ be convex, $A \in \mathbb{R}^{n \times m}$ and $B \in \mathbb{R}^{\ell \times m}$ be matrices, and $c \in \mathbb{R}^m$ a vector. ADMM solves $\min_{x,y} F(x) + g(y)$ subject to $Ax + By = c$ with dual $\lambda \in \mathbb{R}^m$ and penalty $\beta > 0$.

**Streaming setting.** We access $F$ through a distribution $\mathcal{D}$ of functions with $\mathbf{E}_{f \leftarrow \mathcal{D}}[f(x)] = F(x)$. A stream of users $(f_t : t \geq 1)$ arrives, each $f_t$ sampled independently from $\mathcal{D}$.

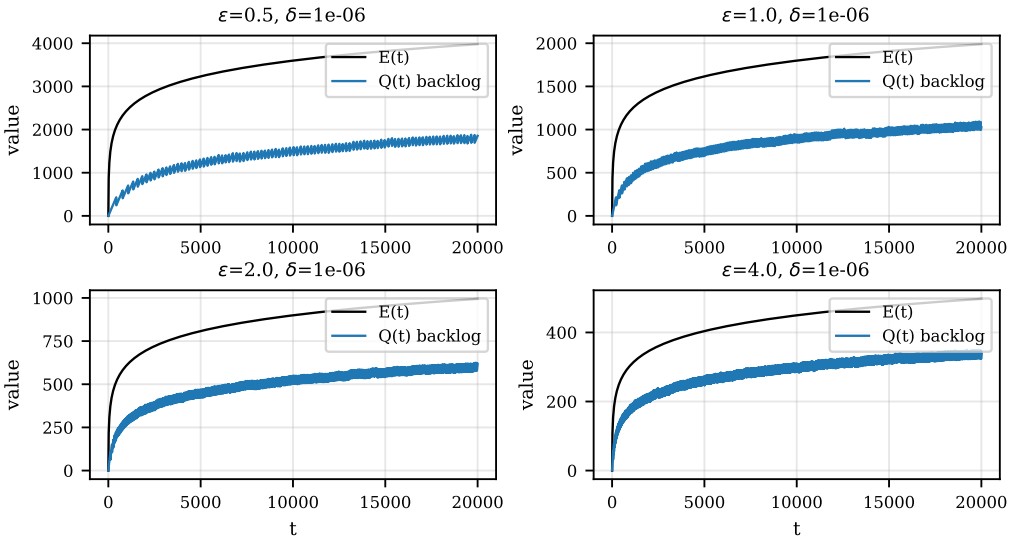

(a) Raw backlog $Q_t$ over time.

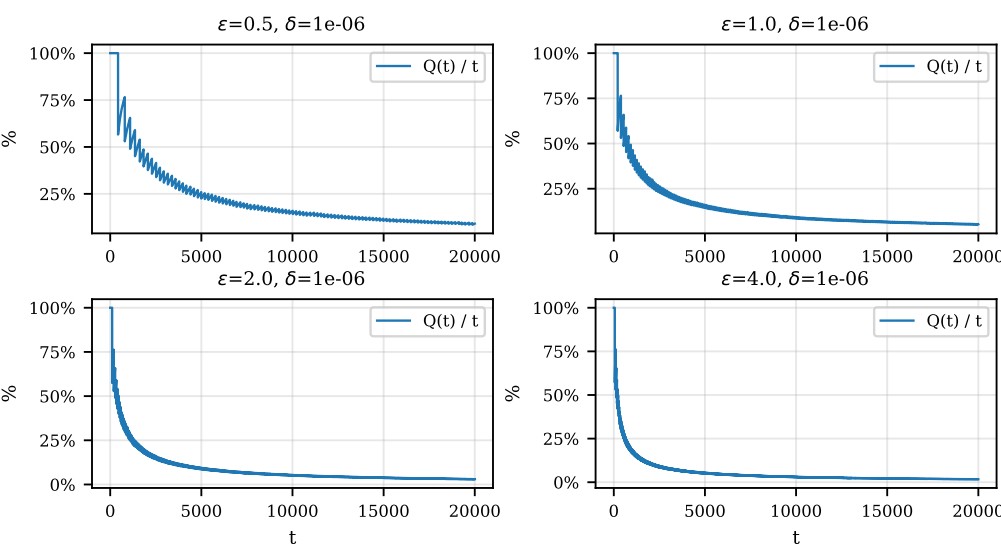

(b) Normalized backlog $Q_t/t$ for the same runs.

*Figure 7.* Privacy-implied latency under different privacy parameters. While the raw backlog may increase with the horizon, the pending fraction decreases over time and remains small at the end of training.

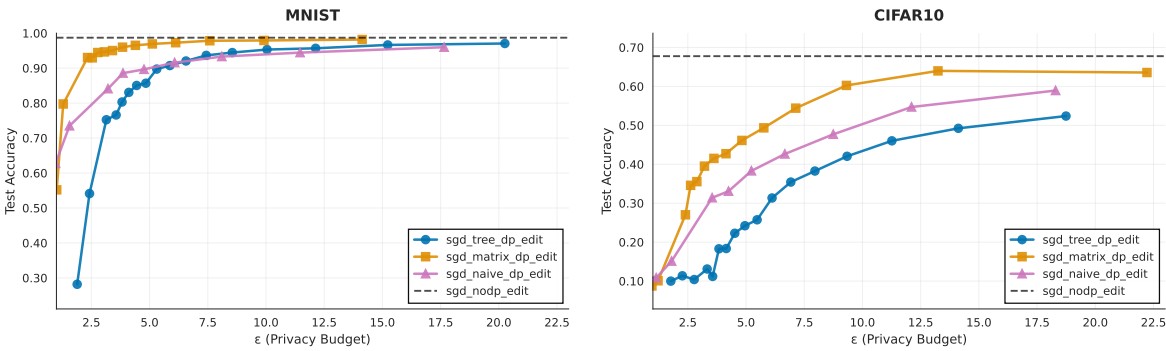

*Figure 8.* Accuracy–privacy tradeoff on MNIST and CIFAR-10 with single-edit privacy. All algorithms use the RandBin wrapper (`sgd_naive_dp_edit`, `sgd_tree_dp_edit`, and `sgd_matrix_dp_edit`).

---

**Algorithm 5** One ADMM Iteration

---

1: **Input:** Previous $(x_{t-1}, \lambda_{t-1}) \in \mathbb{R}^n \times \mathbb{R}^m$ and function $f_t : \mathbb{R}^n \to \mathbb{R}$
2: **Output:** $(x_t, \lambda_t)$
3: $y_{t-1} \leftarrow \mathcal{G}(x_{t-1}, \lambda_{t-1})$
4: $\lambda_t \leftarrow \lambda_{t-1} - \beta(Ax_{t-1} + By_{t-1} - c)$
5: $x_t \leftarrow \mathcal{F}^{f_t}(x_{t-1}, y_{t-1}, \lambda_t)$        ▷ *oracle access to* $\nabla f_t(\cdot)$
6: **return** $(x_t, \lambda_t)$

---

**One ADMM iteration.** Assume each $f$ is $\frac{1}{\eta}$-smooth. For a fixed $f$ and linearization point $\widehat{x}$, define

$$\mathcal{L}^f_{\widehat{x}}(x, y, \lambda) := f(\widehat{x}) + \langle \nabla f(\widehat{x}), x - \widehat{x} \rangle + \mathcal{H}(x, y, \lambda) + \frac{1}{2\eta} \|x - \widehat{x}\|^2$$

and

$$\mathcal{H}(x, y, \lambda) := g(y) - \langle \lambda, Ax + By - c \rangle + \frac{\beta}{2} \|Ax + By - c\|^2.$$

Let $\mathcal{G}(x, \lambda) := \arg\min_y \mathcal{H}(x, y, \lambda)$ and

$$\mathcal{F}^f(\widehat{x}, y, \lambda) := \arg\min_x \mathcal{L}^f_{\widehat{x}}(x, y, \lambda) = M^{-1}\{\widehat{x} - \eta \cdot [\nabla f(\widehat{x}) + E(y, \lambda)]\},$$

where $M = \mathbb{I} + \eta\beta A^\top A$ and $E(y, \lambda) = A^\top(\beta(By - c) - \lambda)$. As shown in (Chan et al., 2024), it is sufficient to pass only $(x, \lambda)$ between successive iterations.

**Privacy setting.** We consider an adversary that observes only $(x, \lambda)$ at each step; $y$ is recoverable from these variables.

**Iteration invariant.** We maintain $M^i x_i = x_0 + S^{(i)}$ for each batch index $i \geq 1$. The private procedure is detailed in Algorithm 6.

**Synthetic Experiment: Setup and Additional Results** We run experiments on the generalized LASSO (elastic net) problem (Zou & Hastie, 2005). Given a dataset $U = \{(a_i, b_i) \in \mathbb{R}^n \times \mathbb{R}\}_{i \in [N]}$ and parameters $c_1, c_2$, each user $i$ is associated with $f_i(x) = (\langle a_i, x \rangle - b_i)^2$. For $x, y \in \mathbb{R}^n$, denote $F(x) = \frac{1}{N} \sum_{i \in [N]} f_i(x)$ and $g(y) = c_1\|y\|_1 + c_2\|y\|_2^2$, yielding $\min_{x,y} F(x) + g(y)$ subject to $x = y$.

**Data generation.** We follow (Goldstein et al., 2014) and generate synthetic data as:

1. For each $a_i \in \mathbb{R}^n$, set

$$a_{ij} = \begin{cases} 50z_{ij}, & \text{if } j \in [\lfloor n/5 \rfloor, \lfloor 2n/5 \rfloor], \\ z_{ij}, & \text{otherwise} \end{cases}$$

where $z_{ij} \sim \mathcal{N}(0, 1)$ independently.

---

**Algorithm 6** Private ADMM for Edit-Style Neighboring User Streams

---

1: **Input:** Differential privacy parameters $(\varepsilon, \delta)$; a user stream $\{f_t : \mathbb{R}^n \to \mathbb{R} \mid t \geq 1\}$; $M := \mathbb{I} + \eta\beta A^\top A$ and $E(y, \lambda) := A^\top(\beta(By - c) - \lambda)$; an initial point $(x_0, \lambda_0)$
2: **Output:** The adversary observes some pair $(x_i, \lambda_i)$ for each step $t \geq 1$, where $i$ is the batch index
3: Instantiate private interactive mechanisms RandBin, for edit-style neighboring user streams, with bin sizes in $[U..2U]$
4: Instantiate adaptive PrivStreamSum, for Hamming-style neighboring vector streams, with overall $(\varepsilon, \delta)$-DP guarantee, where $U = O\left(\frac{1}{\varepsilon}\log\frac{1}{\delta}\right)$
5: $i \leftarrow 0$  ▷ *batch index*
6: **for** each time step $t \geq 1$ at which user $f_t$ arrives **do**
7: $\quad B_t \leftarrow \mathsf{RandBin}(f_t)$  ▷ *potential batch of released users*
8: $\quad$ **if** $B_t = \perp$ **then**
9: $\qquad$ Output $(x_i, \lambda_i)$
10: $\quad$ **else**
11: $\qquad i \leftarrow i + 1$
12: $\qquad y_{i-1} \leftarrow \mathcal{G}(x_{i-1}, \lambda_{i-1})$
13: $\qquad \lambda_i \leftarrow \lambda_{i-1} - \beta(Ax_{i-1} + By_{i-1} - c)$
14: $\qquad d^{(i)} \leftarrow -\frac{\eta}{U} \cdot M^{i-1} \sum_{f \in B_t}\left(\nabla f(x_{i-1}) + E(y_{i-1}, \lambda_i)\right)$  ▷ *mini-batch gradient update*
15: $\qquad S^{(i)} \leftarrow \mathsf{PrivStreamSum}(d^{(i)})$  ▷ *private prefix sum of vectors so far*
16: $\qquad x_i \leftarrow M^{-i}(x_0 + S^{(i)})$
17: $\qquad$ Output $(x_i, \lambda_i)$
18: $\quad$ **end if**
19: **end for**

---

2. Let $x' \in \mathbb{R}^n$ have $x'_j = 3$ for $j \in [\lfloor n/5 \rfloor, \lfloor 2n/5 \rfloor]$ and 0 otherwise. Set $b_i = \langle a_i, x' \rangle + \xi_i$ where $\xi_i \sim \mathcal{N}(0, \sigma_b^2)$ independently.

**Parameter sweeps.** We fix $n = 64$, $N = 10000$, $\beta = 0.15$, and $\eta = 0.05$. We set $\sigma = 0.01, 0.1, 1$ and consider $\varepsilon \in \{0.5, 1, 2, 5\}$, $\delta \in \{10^{-5}, 10^{-6}\}$. We plot objective values against both the number of users and the number of returned batches.

**Random vs. fixed batch sizes.** We compare randomized batch sizes in $[U..2U]$ with a fixed batch size $\frac{3U}{2}$. Figure 9 shows nearly identical behavior; small privacy parameters can delay the first update due to larger $U$.

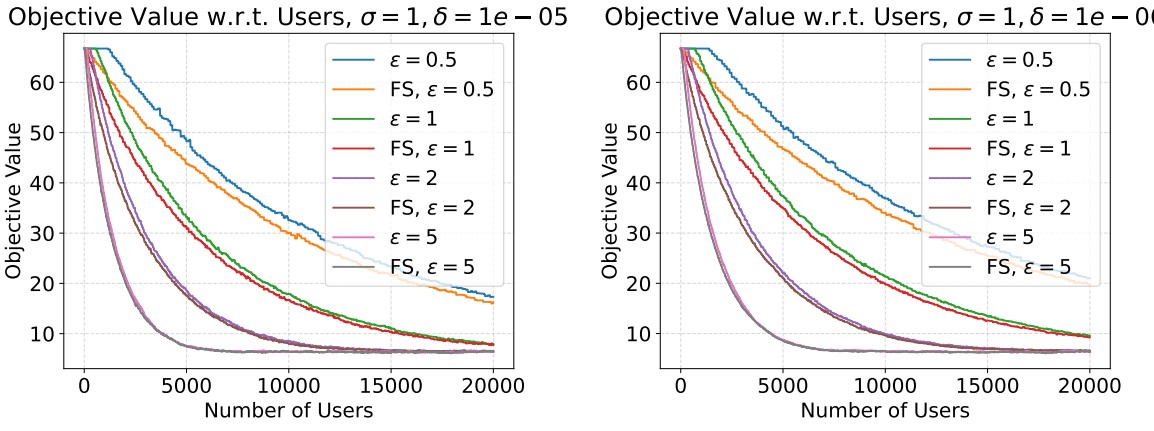

*Figure 9.* Each subfigure compares random versus fixed batch sizes for four values of $\epsilon$ as users arrive.

**Batch-based views.** Figure 10 plots objective values against the number of returned batches.

**User-based views.** We provide sweeps across $(\varepsilon, \delta)$ per noise level in Figure 11. The same qualitative trends persist: tighter privacy (smaller $\varepsilon, \delta$) yields larger batches and slower updates, while batch-indexed views collapse more tightly

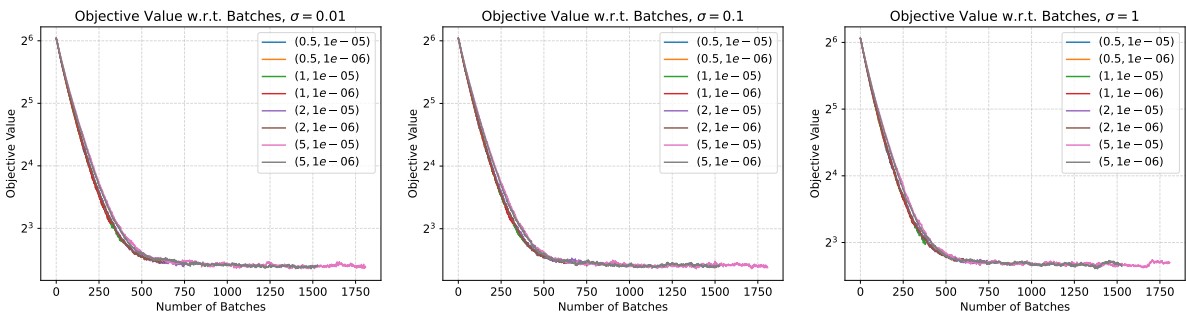

*Figure 10.* Each subfigure compares different $\varepsilon$ as batches are returned.

across parameter choices.

- Smaller privacy parameters lead to larger batch sizes and less frequent model updates, so objective values converge slower. Towards the end, larger batches yield smoother curves due to more accurate gradient estimates.
- Combining with the batch-based views, convergence is mostly driven by the number of model updates. Smaller privacy parameters can balance larger aggregation noise with more accurate gradient estimates from larger batches, resulting in smoother curves.

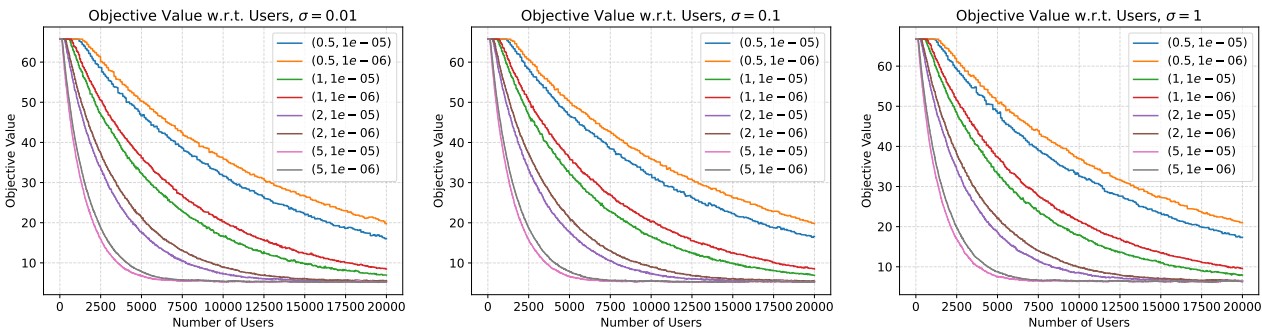

*Figure 11.* Each subfigure compares different $\varepsilon$ and $\delta$ as users arrive.

# H. Other Related Work

As the most relevant related works have already been covered in the introduction, we will elaborate further on some related aspects.

**Choices of Privacy Measure.** The central idea in differential privacy (Dwork, 2006) is that if $V_0$ and $V_1$ are two distributions of outputs produced by a mechanism from two neighboring inputs, then those two distributions should be close. The classical $(\varepsilon, \delta)$-DP notion uses two parameters to quantify closeness, where a smaller value in each parameter means that the two distributions are closer. However, since distributions are inherently complex objects, some information on the two distributions will be inevitably lost when they are compared using just two parameters. When the same Gaussian noise is used to mask two different vectors, researchers have discovered that the Rényi divergence (Mironov, 2017) can capture the variance of the Gaussian distribution perfectly, and hence, can quantify the closeness of two such distributions (with the same variance but different means).

In general, any useful way to quantify privacy guarantees must satisfy the property that if the output satisfies certain privacy requirement, then any further processing of the output cannot violate that specific requirement. This can be formally formulated by requiring that the divergence – used for measuring how different two distributions are – must satisfy the *data processing inequality*; it is worth noting that some common distance notion such as the $\ell_2$-norm does not satisfy this property.

Instead of just using a few parameters to capture the closeness of two distributions, *tradeoff functions* (Dong et al., 2022)

have been proposed to define differential privacy, because a tradeoff function can capture all the essential information about how two distributions differ in the sense that any divergence satisfying the data processing inequality can be recovered from the tradeoff function. Indeed, tradeoff functions offer a powerful tool to describe the composition of private mechanisms.

However, one notational inconvenience is that a larger tradeoff (measured by pointwise comparison) means that the two distributions are closer, which has the opposite interpretation from other divergence parameters such as $\varepsilon$ and $\delta$. In fact, an *ad hoc* concept of *generalized probability distance* has been defined in (Vadhan & Zhang, 2023) to reverse the direction of the inequalities such that it will be consistent with the notion of distance. On the other hand, a simpler way to achieve this notation consistency is to replace a tradeoff function with its complement that is known as a *power function*, which naturally preserves all the equivalent mathematical properties. As we shall see in Definition A.3, a power function also has an intuitive description using the fractional knapsack problem.

As illustrated in (Vadhan & Zhang, 2023; Zhou et al., 2024), if one uses such a powerful tool to define any new notion of differential privacy, then a single composition theorem (such as our Theorem C.1) will be sufficient to recover any composition result from the classical notion to the new notion of privacy. Therefore, it would not be necessary to reconstruct individual advanced composition theorems (Dwork et al., 2010b; Kairouz et al., 2015).

**Hamming vs Edit Neighboring Notions.** In (Birrell et al., 2024), two neighboring notions are considered for **static** databases:

- *Hamming-style.* Two databases have the same number of elements, and they differ in at most one element.
- *Edit-style.* One element from one database is deleted to form the other database.

They considered fixed-size mini-batches, which may be sampled in two ways: with or without replacement. Since they considered sampling from static databases, the difference between Hamming- vs edit-style neighboring static databases would not have such a stark contrast as streams, as deleting the first element of a stream can cause it to change in every position.

**Other Recent Works on Concurrent Composition.** A more general notion of concurrent composition is considered in (Haney et al., 2023), where the privacy parameters of mechanisms can be adaptively chosen. However, as in (Vadhan & Zhang, 2023), each interactive mechanism is associated with a single static database on which its neighboring relation is defined; the adversary may interact adaptively with the mechanism, but neighboring inputs differ only in this underlying database, not in a dynamic stream of updates.

Concurrent composition for mechanisms with adaptively chosen privacy parameters are also considered in (Henzinger et al., 2026), but for neighboring dynamic databases. They also give a formulation based on adaptive DP (Denisov et al., 2022), expressed via a left-or-right style distinguishing game with a verification function, which is essentially the same as our *paired simulation* in Definition B.2.

Note that both works consider composition where each mechanism has a single neighboring relation on its dataset (static or dynamic). In order to capture modular composition in which neighboring notions are defined separately on both the input and the output of a mechanism—and to reason about neighbor-preserving transformations that change the neighboring structure—we need a more refined notion of *neighbor-preserving* paired simulation, given in Definition B.8.

## I. Discussion, Limitations, and Takeaways

**Interpretation: Privacy Implies Latency.** In our setting, the buffering level $U$ is *not* a tunable systems knob: it is fixed (up to constants) by the privacy allocation $(\varepsilon_b, \delta_b)$ via $U = \Theta(\varepsilon_b^{-1} \log(1/\delta_b))$. Consequently, the privacy target $(\varepsilon, \delta)$ induces a concrete batching and latency regime: larger privacy (smaller $\varepsilon$ and $\delta$) forces larger bins and larger buffering backlog, which in turn increases delay. Our experiments therefore report utility versus $(\varepsilon, \delta)$ together with the empirical delay distributions implied by the corresponding $U(\varepsilon_b, \delta_b)$.

**Relation to Convergent Privacy-Loss Analyses.** Recent analyses of Noisy-SGD show that privacy loss can converge or saturate over long horizons (Altschuler & Talwar, 2022; Chien & Li, 2025). These results are complementary to our setting: they primarily analyze the privacy loss of optimizer dynamics, whereas Lemma 4.2 concerns the backlog and queueing delay induced by the edit-to-Hamming buffering wrapper. Extending such optimizer-specific analyses to jointly capture delayed incorporation and buffering-induced latency is an interesting direction for future work.

**Pure-DP Scope.** A positive $\delta$ in our $(\varepsilon, \delta)$-DP construction is not merely a cosmetic convenience. The probability-one delay bound relies on bounded-support buffering noise and on an always-valid prefix-sum error bound; in our implementation, both are obtained in the approximate-DP regime. For a RandBin-style shift-masking first boundary, pure DP would require unbounded-support noise, so the same bounded-support, probability-one finite delay guarantee is impossible for this proof strategy.

This should not be read as a blanket impossibility theorem for all possible pure-DP edit-private continual mechanisms. If bounded support and probability-one finite delay are dropped, pure DP is not ruled out a priori. However, the present NPDP/NPP proof architecture faces an additional underflow obstacle: with unbounded bin-size or scheduler noise, the scheduler may request an emission before enough real items are available. Simple fixes such as padding or aborting do not preserve the current neighboring-bin alignment argument, since one execution may need to pad or abort exactly where the neighboring execution emits an ordinary bin. Thus our current pure-DP barrier is a limitation of the RandBin-style bounded-delay construction and proof architecture, rather than a general impossibility claim.

**Limitations.** First, our end-to-end guarantee is stated for *single-edit* neighboring user streams (one insertion/deletion). Handling multiple edits can be obtained via standard composition/group-privacy reasoning, but doing so necessarily weakens the privacy parameters and correspondingly increases the induced latency through $U(\varepsilon_b, \delta_b)$. Second, the certification theorem is intentionally conservative: it applies to independently decomposable (prefix-causal, fresh-randomness) continual-release primitives. Mechanisms that reuse randomness across rounds or depend on future inputs fall outside Checklist 1 and require separate, interaction-aware analysis. Third, we focus on a streaming SGD instantiation; other optimizers fit the same template, but we do not claim the pipeline resolves broader issues such as client clustering or personalization in federated settings.

**Practical Mapping: Events, Releases, and Rounds.** Delay is naturally measured in *events*: $D(i)$ counts subsequent arrivals until $f_i$ is binned. If one prefers a "round" interpretation, the relevant unit is the emitted-bin index $k$ (i.e., the number of releases), since the model is updated only at emission times; the event-level delay distribution then induces a distribution over the number of releases an event waits.

**Takeaway.** The main contribution is an *auditable* continual-learning recipe: randomized buffering reduces single-edit to a Hamming-style interface with explicit backlog guarantees, certification identifies when standard continual DP primitives remain private under adaptive interaction, and modular composition yields an end-to-end DP guarantee that is validated empirically together with its privacy-implied latency.

