# OpenReview forum: "Continual Learning With Participation Privacy: An Auditable Buffering-Aggregation Recipe"
_ICML.cc/2026/Conference — ICML 2026 regular_

### Official Review · Reviewer_uA2A · 2026-03-07

**Soundness:** 3
**Presentation:** 2
**Significance:** 2
**Originality:** 2
**Overall Recommendation:** 5
**Confidence:** 1

**Summary:**

The submission studies continual learning systems that release intermediate model snapshots. The core privacy target is participation privacy on a stream. Two streams are neighbors if one can be obtained from the other by a single deletion or insertion of an element, and the rest of the elements get shifted. This breaks continual learning analysis build for Hamming neighbors (we change only one aligned element between the sequences).

The proposed recipe is as follows:
1) To handle single-edit adjacency, the paper introduces randomized buffering, called RandBin. This is a wrapper that collects incoming user events into random-sized bins before passing them to a DP mechanism. The goal is to prevent one insertion/deletion from misaligning all batches. RandBin emits events of size [U, 2U] where U is governed by privacy epsilon and delta. The events introduce delay and backlog in a controlled way.
2) Tree prefix sums privately aggregates the per-bin updates.
3) For a broad class of independently decomposable mechanisms, Hamming neighbor DP proof lifts to this setting.

The authors instantiate the recipe with streaming DP-SGD and empirically evaluate privacy, utility and latency tradeoffs.

**Compliance With Llm Reviewing Policy:**

Affirmed.

**Key Questions For Authors:**

1. Is randomized buffering necessary for this proof technique to go through? Are there any other proof techniques that would achieve similar results?

2. Regarding Lemma 4.2. Do you believe the backlog bound or the privacy-implied delay bound are asymptomatically tight or can be improved? Are the log factors are proof artifacts or fundamentally necessary?

3. How restrictive is independent decomposability condition? Are there DP-SGD variants (momentum, adaptive optimizers) that fail this condition?

**Limitations:**

yes

**Strengths And Weaknesses:**

Soundness: clear end to end privacy guarantees. The main guarantee (Theorem 3.1) explicitly splits privacy budgets into buffering (eps_b, d_b) and aggregation (eps_a, d_b). It explicitly sets buffer size to U = O(1/eps_b * log(1/d_b)). It requires downstream continual primitive to be adaptively DP on Hamming update streams. Randbin wrapper interface is spelled out at the right level of abstraction. Weaknesses: main proofs are deferred to the appendices and rely on fairly involved proofs, which might be non-trivial for many readers to follow end-to-end.

Presentation: High level narrative is well structured. The certification theorem has a proof sketch provided. Appendices provide examples of the key concepts. It would be helpful if some of these examples would be moved to the main body. Weaknesses: the paper is for audiences familiar with DP. For other audiences it might be hard to digest. It might be helpful to have a glossary or table of introduces concepts.

Significance: the paper targets a real gap: privacy for released trajectories with edit neighboring relations, where naive continual learning analysis does not transfer. The authors do empirical evaluation  and provide an evidence that the wrapper cost can be small in practice. Weaknesses: participation privacy unit is one edit and the guarantee degrades with k edits and correspondingly increases the required budgets.

Originality: Novel idea of using randomized buffering specifically as an edit to Hamming distance interface. This enables the use of standard continual learning primitives. Weakness: the individual components (tree prefix sums for instance, randomized buffering) exist in literature. The novelty lies in modular packaging, which may seem incremental for experts of interactive DP.

---

> ### Author Rebuttal · Authors · 2026-03-29
>
> [Presentation Improvement.] We are glad the high-level narrative and certification-theorem sketch were clear. We agree that accessibility can be improved for readers not already familiar with continual DP. In the revision, we will move a representative motivating example from the appendix to the main text, and add a compact glossary / notation table for the main concepts and symbols. These are presentation improvements rather than technical gaps, and we expect them to make the paper substantially easier to digest.
>
>
> [Q1]
> Randomized buffering is not needed for the certification theorem per se; that theorem is a reusable adaptivity lift for independently decomposable continual mechanisms. In our proof, buffering serves a different purpose: it converts single-edit adjacency, where deterministic batching can fail because one insertion or deletion may shift many later batch boundaries, into a bounded Hamming-style perturbation that standard continual-DP primitives can handle. We therefore do not claim that randomized buffering is the only possible route. However, for this modular proof strategy, some edit-shielding interface, or else a direct native proof under edit adjacency, appears necessary, and achieving comparable utility without such an interface seems technically difficult. Formally ruling out alternatives would likely require a sharper notion that also excludes methods whose internals are effectively equivalent to randomized buffering despite superficial differences; we view this as an interesting future research direction. Our contribution is to cleanly isolate the two reusable ingredients: an upstream edit-to-Hamming wrapper and a downstream certification theorem.
>
>
> [Q2]
> We do not currently claim a matching minimax lower bound for every NPDP wrapper. However, the $\log t$ dependence in Lemma 4.2 is not merely loose algebra from our proof. The lemma is obtained by instantiating RandBin's emission scheduler with a continual private prefix-sum mechanism and then translating that mechanism's additive error into backlog. Concretely, if $M_t$ denotes the cumulative number of emitted items by time $t$, then on a unit-rate stream the backlog is exactly $Q_t=t-M_t$. Thus, for this scheduler family, a horizon-uniform backlog bound would immediately yield a horizon-uniform continually released private counter, and more generally any $o(\log t)$ backlog bound would imply an $o(\log t)$ continual-counting error.
>
> Known lower bounds for continual private counting make such an improvement implausible for this proof route: classical pure-DP continual counting already has an Ω(logT) lower bound [Dwork et al. STOC 2010], and recent approximate-DP online lower bounds likewise show logarithmic dependence on the horizon in the sparse regime [Cohen et al. COLT 2024].
>
> For this reason, we believe the horizon dependence is intrinsic to the current prefix-sum-based scheduler, even if we do not yet prove a matching lower bound for the full NPDP-wrapper abstraction. What remains open is whether the exact form
> $O(\epsilon_b^{-1}\log t,(\log t+\log(1/\delta_b)))$
> is minimax-tight, or whether one of the logarithmic factors can be improved by a sharper continual counter or scheduler.
>
>
> [Q3]
> This is closely related to [Q2] of reviewer [heKh] about how restrictive the independently decomposable condition is. As noted there, independent decomposability is a sufficient black-box condition for Theorem 4.3, not a necessary condition for adaptive privacy. Regarding DP-SGD variants, we would be cautious about making blanket claims: methods with momentum or adaptive optimizer state may fall outside the theorem's scope because they introduce extra state or cross-round dependence, but this is a limitation of the certification theorem, not evidence that such methods are not adaptively private. Whether a specific variant satisfies the condition requires a separate audit of its implementation and state evolution.

---

> > ### Author Rebuttal · Reviewer_uA2A · 2026-04-01
> >
> > I thank the authors for addressing my questions. I increased my score to 5 (Accept).

---

### Official Review · Reviewer_N7Cu · 2026-03-08

**Soundness:** 3
**Presentation:** 3
**Significance:** 3
**Originality:** 3
**Overall Recommendation:** 5
**Confidence:** 4

**Summary:**

This paper studies differential privacy under continual observation under a new "adjacency" measure: two streams are considered adjacent if one can add/delete* a single event in one stream to get the other. This is different from the well-studied and popular change-one formulation: where two streams are adjacent if they differ on one event and all remaining events are aligned in index.

This new privacy defintion is aimed to capture the "participation" privacy requirement: an individual may wish their participation into the loop to be privatized, whereas in the traditional model it assumed the individuals' participation as public knowledge and only aimed to protect their feedback to the learner.

In the new model, since adding/deleting one event shifts all future udpates, the algorithm can no longer respond as soon as an update has arrived. Instead, the paper proposes to buffer a (randomized) number of events and process them all in update. By randomizing the size of batches a little bit, the participation privacy can be preserved.

The paper also discusses extension of this new framework to handle adaptive feedback loops (where the future inputs may depend on the previous transcripts), and presented a general lifting theorems to "lift" popular "static" privacy proofs to handle this adaptive setting.

**Compliance With Llm Reviewing Policy:**

Affirmed.

**Final Justification:**

I am supportive of this paper. It has positionsed itself with the literature well (after rebuttal), carries a solid conceptual message with interesting techincal contributions.

**Key Questions For Authors:**

Can the authors compare with the said reference? My current evaluation is that I believe in the novelty in the concept but I am less sure about the technical significance of the submission.

**Limitations:**

Yes

**Strengths And Weaknesses:**

Strengths:

* This paper studies a (IMHO) fundamental issue on differrential privacy under continual observation.
* The writing is clear and easy to follow.
* The proposed algorithm is intuitive and natural.

Weaknesses:

* I believe the approximate DP relaxation is necessary for the model, though I don't have a rigorous proof currently in mind. It would be good if the authors can comment on this.
* It seems the paper takes heavy inspiration from the related development in differentially olivious algorithms), and may have overlooked some related development in the DP side: specfically in https://arxiv.org/pdf/2211.06387 a very similar challenge arises in their privacy proof. And a similar geometric buffering algorithm + simulaton privacy proof were offered there. I wonder how the authors would compare the technical contribution with the referenced paper.
* Another related reference is (https://arxiv.org/pdf/2403.00028, appendix B). There, the goal was to get a "continual learner" with utility guarantee independent of $T$ (the number of "days"). Instead, the utility should only depend on the number of "interesting events" (updates) across the $T$ days.  Say there are $k\ll T$ such updates. The algorithm cannot react as soon as one new update arrives (the issue is, again, similar to the "participation privacy" consideration described in the current submission. It was then necessary to buffer the updates into groups to hide the *existence* of the updates.


===== UPDATES =====

March 31st: I have read the rebuttal and increased the score to 5.

---

> ### Author Rebuttal · Authors · 2026-03-29
>
> [W1]
> We agree that a positive $\delta$ in $(\epsilon,\delta)$-DP is not merely a convenience in our current construction. The almost-sure (probability-1) delay bound relies on bounded-support buffering noise and on an always-valid prefix-sum error bound; in our appendix, both are obtained only in the approximate-DP regime. More fundamentally, for any RandBin-style shift-masking first boundary, pure DP forces unbounded support, so a finite delay bound with probability 1 is impossible for that class of mechanisms.
>
> [W2]
> We agree that the Reorder-Slice-Compute (RSC) paradigm of Cohen et al. [STOC 2023] is closely related at the level of the random-partition idea: both RSC and our randomized streaming buffering use geometric-type random boundaries to address the same one-edit shift/domino phenomenon and to enable synchronization between neighboring executions. At the same time, the earliest example of this geometric-buffering lineage that we are aware of is Chan et al. (SODA 2019; JACM 2022), which our current related-work discussion already cites as the obliviousness-side precursor. Our setting differs in one important respect: unlike Cohen et al., our continual streaming model exposes release timing itself to the adversary, so privacy must also account for whether a step outputs a real block or the dummy symbol $\bot$. By contrast, Cohen et al. has an ordered notion of progress through slice rounds, and implicitly through cumulative consumed prefix length, but it does not natively include a finer-grained time axis corresponding to individual elements or explicit per-step no-release events. On the other hand, in the relevant buffering subroutine of Chan et al., the input is processed via a sequential scan into a working buffer, so the scan position can be viewed as a proxy for logical time. We will revise the related-work discussion to mention Cohen et al. explicitly, clarify this close conceptual connection, and state more clearly that our additional technical burden is handling observable timing and dummy-release behavior in the interactive continual setting.
>
>
> [W3]
> We agree that Appendix B of [Cohen et al. COLT 2024] is a relevant conceptual precedent and should be cited in the related-work discussion. In that construction, the mechanism directly observes the online bit stream, including every positive event, but suppresses the transcript's immediate positive reaction: even after seeing a 1, it may continue outputting $\bot$, and $\top$ outputs are allowed only after sufficiently many prior positive inputs. As a result, utility depends on the number of positive or "interesting" events rather than the horizon $T$.
>
> That said, the connection is only close in spirit, not a technical equivalence. First, Appendix B [COLT 2024] is not an explicit randomized buffering-into-bins construction of the kind used here; it is closer to delayed thresholding on a directly observed online bit stream. Second, the privacy setting is materially different: their Appendix B works in a JDP-based Mirror setting, whereas our paper studies full trajectory-level $(\varepsilon,\delta)$-DP under adaptive interaction.
>
> Most importantly, the source of delay is different. In Cohen et al., the mechanism directly sees each positive update and delays the transcript's reaction to those observed positives, so multiple positive events may need to accumulate before the output changes. In our setting, the online mechanism never directly observes "the inserted event" or "the deleted event" as such. A single insertion/deletion appears only in the neighboring-stream analysis, where the main challenge is the global alignment shift induced by one edit. RandBin therefore delays for a different reason: it waits for enough future arrivals (as opposed to future insertion/deletion events), under a randomized private emission scheduler, to form blocks of size in $[U,2U]$, so that a single edit is converted into only a bounded Hamming-style perturbation at the downstream continual interface, while protecting the full transcript including release timing.
>
> We will revise the related-work discussion accordingly: [COLT 2024] is an important conceptual antecedent for privacy-induced delayed reaction, but it does not address the same buffering interface, observability structure, or edit-induced alignment obstacles that arise in our setting.

---

> > ### Author Rebuttal · Reviewer_N7Cu · 2026-03-31
> >
> > Thank you for your informative resposne. I have increased my score to 5 and I think this can be a solid addition to NeurIPS.
> >
> > I would like to see the discussion about pure-DP (im)possibility, as well as the related work to be incorperated in the revision of the paper.
> >
> > One follow up question: the fact that pure-DP cannot be compatible with bounded support partly matches my intuition. if we do not insist on bounded support or almost surely delay bound, is there possibly a pure-DP scheme? Or are there any other barriers?

---

> > > ### Author Response · Authors · 2026-04-02
> > >
> > > Thank you again for the helpful follow-up. We will incorporate both points in the revision: we will add the related-work discussion and clarify the scope of the pure-DP discussion.
> > >
> > > Our current statement should be read as a barrier for the present RandBin-style proof strategy, not as a blanket impossibility theorem for all possible edit-private continual mechanisms. If bounded support and the probability-1 delay guarantee are dropped, then pure DP is not ruled out a priori. However, for the current approach there are further obstacles.
> > >
> > > The issue is that a pure-DP variant would naturally require both the bin-size randomness and the scheduler noise to have unbounded support. Once this is allowed, actual underflow events occur with positive probability: the scheduler may request a bin before enough real items are available. In the approximate-DP regime, one can isolate such events through the always-valid error bound used in our appendix. In pure DP, by contrast, these positive-probability paths must be handled exactly.
> > >
> > > For the current NPDP / NPP proof architecture, this is where the difficulty becomes substantive. The paired simulation must output not only the real execution on one stream, but also a companion output stream for the neighboring edit-stream, and those two output streams must themselves satisfy the required neighboring-bin relation. Underflow breaks this construction. In particular, simple fixes such as padding or aborting do not preserve the current NPP alignment argument: on a positive-probability path, one side may need to pad or abort exactly where the other side emits an ordinary bin, and this falls outside the local neighboring-bin structure used by our wrapper proof.
> > >
> > > So our current understanding is that pure DP is impossible for the bounded-support / almost-sure-delay version of RandBin, and that even without those requirements the present proof architecture faces additional underflow-related barriers. At the same time, we do not claim a general impossibility theorem for every conceivable pure-DP scheme. We will revise the paper to make this distinction explicit.

---

### Official Review · Reviewer_Ns96 · 2026-03-12

**Soundness:** 4
**Presentation:** 3
**Significance:** 3
**Originality:** 3
**Overall Recommendation:** 5
**Confidence:** 4

**Summary:**

The authors discover the single-edit privacy risk in the weight release of continual learning, and further conduct persuasive experiments to confirm the risk practically. To solve the issue, the authors propose RandBin and an adaptive-safety certification, which jointly guarantee the privacy loss. Finally, the experiments about privacy–utility–latency tradeoffs are reported on standard datasets in the field of private learning.

**Compliance With Llm Reviewing Policy:**

Affirmed.

**Final Justification:**

Most of my major concerns have been solved. Only one additional suggestion is for Q1: I think the author can attach the results of amortized backlog to show how the fraction of pending arrivals becomes asymptotically negligible, so it’s complementary to the seemingly unbounded backlog.

**Key Questions For Authors:**

I suggest the author to focus on Weakness 3, as it’s crucial for the privacy analysis.

**Limitations:**

Yes, which is shown in Appendix I.

**Strengths And Weaknesses:**

**Strengths**:
1) The authors are motivated by the huge discrepancy between the hamming neighbor and the edit neighbor, so they indicate different privacy risks. The motivation is strong since the edition operation is common in the field of continual learning, e.g., sometimes we need to unlearn some samples, we have to delete some, and release an adaptive weight stream, which may uncover the information of deleted samples. Besides, experiments in Appendix G.1 are easy to follow and powerful to claim the importance of single-edit privacy risk.

2) The authors provide proofs in detail, especially for one in Appendix C.

3) The paper is well organized and easy to follow.

**Weakness**:
1) I recommend that the authors justify the feasibility of the order of backlog proved in Lemma 4.2 and shown in Fig. 7, since it’s unbounded (due to the fact that a tree with depth log T is required) and the order of 10^3 in the experiment is sometimes a large value. I think further analysis can be discussed on some specific loss functions and noisy learning algorithms to derive saturated bounded error [1][2].

2) Some research [3] utilizes sliding window release to achieve saturated privacy loss. It‘s beneficial for the authors to mention and compare such a mechanism, at least declare the infeasibility of sliding window release in this paper.

3) Is it really possible to know whether an element is deleted from x0 to x1 in constructing RandBin_{b}^{npp}? If not, in some cases (e.g., elements are anonymous, or exist duplicated IDs), RandBin_{b}^{npp} could not be well defined.

[1] Chien E, Li P. Convergent privacy loss of noisy-sgd without convexity and smoothness[J]. arXiv preprint arXiv:2410.01068, 2024.
[2] Altschuler J, Talwar K. Privacy of noisy stochastic gradient descent: More iterations without more privacy loss[J]. Advances in Neural Information Processing Systems, 2022, 35: 3788-3800.
[3] Watson L, Ghosh A, Rozemberczki B, et al. Continual and Sliding Window Release for Private Empirical Risk Minimization[J]. arXiv preprint arXiv:2203.03594, 2022.

---

> ### Author Rebuttal · Authors · 2026-03-29
>
> [Q1]
> We would like to clarify that the cited works [Chien et al. 2024, Altschuler et al, Neurips 2022] primarily analyze convergent privacy loss (DP guarantees) for Noisy-SGD, rather than utility/accuracy in our buffered continual-release setting; their techniques do not directly address the analysis of backlog, queueing delay, or delayed incorporation induced by an edit-to-Hamming wrapper.  We therefore view them as complementary motivation for future optimizer-specific analysis, rather than as a direct alternative to Lemma 4.2.
>
> For Lemma 4.2, the relevant quantity is not only the raw backlog $Q_t$, but also its normalized size. Our bound is
> $Q_t = O(\frac{1}{\varepsilon_b}\log t,(\log t+\log(1/\delta_b)))$,
> hence
> $Q_t/t \to 0$ as $t \to \infty$.
>
> So although the worst-case backlog is unbounded with the horizon, the fraction of pending arrivals becomes asymptotically negligible. In Figure 7, even the largest observed backlog is about $10^3$ at horizon $2 \times 10^4$, i.e. roughly $5%$. Since backlog corresponds to delayed rather than discarded events, the practical implication is that by that horizon the system has already incorporated about $95%$ of the stream, and the remainder can be recovered by running only modestly longer. We will revise the discussion around Lemma 4.2 / Figure 7 to emphasize this normalized interpretation, which better reflects the practical impact than the raw backlog alone.
>
> More broadly, our contribution here is an auditable systems/privacy guarantee: Lemma 4.2 makes the privacy-implied latency explicit. Deriving bounded end-to-end optimization error for particular loss classes and noisy learning algorithms is an interesting future direction, but it requires additional assumptions beyond the scope of the present paper.
>
> [Q2]
> Thank you for pointing out Watson et al. (2022). That paper is relevant as another example where privacy loss can saturate under continual release, but it studies a different objective and threat model: releasing ERM models for recent data windows, including a constant-size sliding-window variant, with bounded per-point privacy over an unbounded horizon. Our setting instead studies participation privacy under single-edit neighboring streams with adaptive feedback, where one insertion/deletion shifts all subsequent positions. In this regime, a sliding-window formulation is not a drop-in replacement: it does not by itself resolve the global alignment shift that breaks standard Hamming-style continual-DP analyses, and it also targets a different utility notion, since windowed methods intentionally forget older data whereas our framework tracks delayed but eventual incorporation via explicit backlog/delay guarantees. We therefore do not claim a formal impossibility of sliding-window approaches, but rather that they address a different problem from the edit-adjacency setting considered here. Our contribution is an auditable edit-to-Hamming shielding interface, together with a certification theorem that lets standard continual primitives be reused safely under feedback.
>
> [Q3]
> The reviewer is correct that, without an additional disambiguating assumption, $\randbin_b^{\npp}$ is not well defined for anonymous streams with duplicate values. A concrete ambiguity is the prefix pair $x^{(0)}{1:4}=abcb$ and $x^{(1)}{1:4}=acbc$. This same observed history is consistent both with $abcbc \to acbc$ by deleting the first $b$ from $x^{(0)}$, and with $acbcb \to abcb$ by deleting the first $c$ from $x^{(1)}$. Thus, before the next disambiguating symbol arrives, an online NPP simulator cannot know which stream contains the unmatched early item. Since RandBin is FIFO, the two interpretations require different alignments of later repeated $b/c$ arrivals and therefore different bin assignments. So in the anonymous/duplicate-ID setting the construction indeed breaks.
>
> This is why our appendix explicitly adds the technical assumption that, once the first mismatch appears, the next step reveals from which stream the deletion came; in practice, this is automatically satisfied when each item carries a unique ID or metadata tag. We agree this assumption should be more visible, and in the revision we will highlight it explicitly in the main text, rather than leaving it only in the appendix.

---

> > ### Author Rebuttal · Reviewer_Ns96 · 2026-04-01
> >
> > Most of my major concerns have been solved. Only one additional suggestion is for Q1: I think the author can attach the results of amortized backlog to show how the fraction of pending arrivals becomes asymptotically negligible, so it’s complementary to the seemingly unbounded backlog.

---

> > > ### Author Response · Authors · 2026-04-02
> > >
> > > Thank you for the helpful suggestion. We agree that a normalized view would better complement the raw backlog curves in Figure 7. In the revision, we will add a plot of normalized backlog, namely $Q_t/t$ (equivalently, backlog divided by the cumulative number of arrivals up to time $t$), versus time. This directly visualizes the point made in our response to Q1 that, although the absolute backlog can grow with the horizon, the pending fraction becomes asymptotically negligible. We will also report representative endpoint values in the caption/text (e.g. the largest observed backlog in Figure 7 is about $10^3$ at horizon $2\times 10^4$, which is about $5%$).

---

### Official Review · Reviewer_heKh · 2026-03-13

**Soundness:** 3
**Presentation:** 3
**Significance:** 3
**Originality:** 3
**Overall Recommendation:** 5
**Confidence:** 3

**Summary:**

This paper studies continual learning with participation privacy under single-edit neighboring streams, where a single insertion or deletion can shift all subsequent events, making standard continual DP analyses based on Hamming-neighbor inapplicable. The paper proposes a modular pipeline that combines randomized buffering (RandBin) with standard continual DP primitives, and further establishes a certification theorem showing that a broad class of continual mechanisms are adaptive-safe, thereby yielding trajectory-level $(\varepsilon,\delta)$-differential privacy guarantees with an explicit privacy–latency tradeoff.

**Compliance With Llm Reviewing Policy:**

Affirmed.

**Final Justification:**

This paper is solid and well-written. While its originality lies mainly in problem formulation and modular design rather than new components, the rebuttal effectively addressed my concerns and clarified the contributions. So, I have raised my score to a positive recommendation.

**Key Questions For Authors:**

1. How do different allocations between $(\varepsilon_b,\delta_b)$ and $(\varepsilon_a,\delta_a)$ affect latency and overall performance? I would like to see a more systematic analysis of this tradeoff.
2. How essential is the independently decomposable assumption in Theorem 4.3? In particular, can the certification result be extended to mechanisms with correlated noise or partial randomness reuse across rounds?

**Limitations:**

yes

**Strengths And Weaknesses:**

## Strengths
1. The paper is well motivated, and the main idea is intuitive. In particular, viewing RandBin as an interface that transforms single-edit neighboring user streams into a downstream update stream that is more amenable to Hamming-style continual DP analysis is conceptually clean and easy to follow.
2. The theoretical analysis is fairly complete.

## Weaknesses
1. The degree of novelty is somewhat difficult to assess, since parts of the framework appear to build on existing randomized buffering ideas and standard continual-DP components. The paper would benefit from a clearer discussion of what is fundamentally new and what is inherited from prior work.

---

> ### Author Rebuttal · Authors · 2026-03-29
>
> [W1]
> We agree that some building blocks are inherited, and we will clarify this more explicitly. Our novelty is not in tree mechanisms, randomized buffering, or composition principles in isolation. Rather, it is in using these ingredients to solve a different privacy problem: trajectory-level DP for single-edit participation streams under adaptive feedback. In our setting, one insertion/deletion shifts all later positions, so standard Hamming-neighbor continual-DP analyses and deterministic batching do not directly apply. The main new ingredients are therefore: (i) the edit-neighbor formulation for continual learning under feedback, (ii) the use of RandBin as an interface that converts single-edit streams into a bounded Hamming-style interface with explicit backlog/delay guarantees in this continual setting, and (iii) the certification theorem showing that, for independently decomposable continual mechanisms, a standard non-adaptive Hamming-DP proof lifts automatically to the adaptive setting. The end-to-end trajectory guarantee then comes from packaging these ingredients into a single auditable pipeline.
>
> [Q1]
> At the asymptotic level, any constant-fraction split is sensible, and 50/50 is the simplest symmetric default. End-to-end privacy composes additively as $(\epsilon,\delta)=(\epsilon_b+\epsilon_a,\delta_b+\delta_a)$, while buffering latency scales as $U=\Theta(\epsilon_b^{-1}\log(1/\delta_b))$. Thus, setting $\epsilon_b=c\epsilon$ and $\epsilon_a=(1-c)\epsilon$ for any fixed $c\in(0,1)$ changes guarantees only by constant factors; likewise, assigning constant fractions of $\delta$ changes $\log(1/\delta_b)$ only by an additive $O(1)$ term. So 50/50 is asymptotically as good as any other constant split, though not necessarily the best constant in a specific finite-sample regime. In practice, the best $c$ can be chosen empirically by sweeping budget splits and measuring the latency-accuracy tradeoff.
>
> [Q2]
> Theorem 4.3 is best viewed as a sufficient black-box certification condition, not a necessary condition for adaptive privacy. Its role is to give a simple structural criterion under which a non-adaptive privacy proof lifts automatically to the adaptive setting: if each round is prefix-causal and uses fresh independent randomness, then conditioning on the past transcript does not change the one-step privacy argument. This is why the theorem is intentionally conservative.
>
> That conservatism matters. If independent decomposability fails, one can no longer conclude adaptive privacy from the non-adaptive proof alone, and adaptive privacy can in fact be strictly worse. For example [Denisov et al. NeurIPS 2022], in a two-step anisotropic Gaussian mechanism, the first step reveals a random direction $u\in\mathbb{R}^2$, and the second answers $x\in\mathbb{R}^2$ with large noise along $u$ but small noise in the orthogonal direction. A non-adaptive adversary averages over $u$, but an adaptive adversary can first observe $u$ and then choose neighboring inputs in the low-variance direction, yielding a stronger distinguisher. So reused or correlated randomness can genuinely worsen adaptive privacy.
>
> At the same time, failure of Theorem 4.3's condition does not mean adaptive privacy is false; it only means the theorem no longer certifies it as a black box. Some mechanisms outside the theorem can still be analyzed directly with a mechanism-specific proof. RandBin itself is one example: its adaptive analysis requires a separate argument because it maintains a secret internal queue state and is therefore not independently decomposable in the theorem's sense.
>
> So the right takeaway is: independent decomposability is essential for Theorem 4.3 as a reusable certification tool, but it is not necessary for adaptive privacy in general. Mechanisms with correlated noise or partial randomness reuse may still be adaptively private, but they typically require exposing an appropriate independently decomposable intermediate object and treating the final output as post-processing, or else giving a custom interaction-aware proof. We chose the stronger assumption because it yields a short, checkable theorem that covers many practical continual mechanisms while clearly separating the cases that need bespoke analysis.

---

> > ### Author Rebuttal · Reviewer_heKh · 2026-04-02
> >
> > Thank you for addressing my questions. The rebuttal has deepened my understanding of your research, and I will increase my score to 5.

---

### Decision · Program_Chairs · 2026-04-30

**Decision:**

Accept (regular)

**Comment:**

This paper studies continual learning with participation privacy under single-edit neighboring streams, where a single insertion or deletion can shift all subsequent events. The paper proposes to buffer a (randomized) number of events, and by randomizing the size of batches the participation privacy can be preserved. All reviewers found the paper to be interesting and relevant to ICML, with high praises as to its presentation and theoretical rigor. As such, I see no qualms about accepting the paper.